# Parasitic plasmids are anchored to inactive regions of eukaryotic chromosomes through a nucleosome signal

Fabien Girard[1,2,3], Antoine Even[4], Agnès Thierry [1], Myriam Ruault[4], Léa Meneu [1,2], Pauline Larrous [1,2], Mickaël Garnier [4], Sandrine Adiba[5], Angela Taddei[4], Romain Koszul [1✉] & Axel Cournac [1✉]

## Abstract

**Natural plasmids are common in prokaryotes, but few have been documented in eukaryotes. The natural 2µ plasmid present in the yeast *Saccharomyces cerevisiae* is one of these best-characterized exceptions. This highly stable genetic element has coexisted with its host for millions of years, faithfully segregating at each cell division through a mechanism that remains unclear. Using proximity ligation methods (such as Hi-C, Micro-C) to map the contacts between 2µ plasmid and yeast chromosomes under dozens of different biological conditions, we found that the plasmid is tethered preferentially to regions with low transcriptional activity, often corresponding to long, inactive genes. These contacts do not depend on common chromosome-structuring factors, such as members of the structural maintenance of chromosome complexes (SMC) but depend on a nucleosome-encoded signal associated with RNA Pol II depletion. They appear stable throughout the cell cycle and can be established within minutes. This chromosome hitchhiking strategy may extend beyond the 2µ plasmid/*S. cerevisiae* pair, as suggested by the binding pattern of the natural eukaryotic plasmid Ddp5 along silent chromosome regions of the amoeba *Dictyostelium discoideum*.**

**Keywords** Eukaryotic Plasmid; Two micron plasmid; *Saccharomyces cerevisiae*; *Dictyostelium discoideum*; Nuclear organization
**Subject Categories** Chromatin, Transcription & Genomics; Genetics, Gene Therapy & Genetic Disease

## Introduction

Characterizing how mobile DNA elements, such as plasmids, viruses and transposons, are maintained in their hosts is necessary to understand their phenotypes and impact. Some can be retained for long periods in the nucleus through various mechanisms. For instance, the hepatitis B virus (HBV) can integrate into the genome and remain in a latent state for very long periods (Dias et al, 2022), while the Epstein Barr virus (EBV) of the herpesvirus family remains as an episome in the nucleus, replicating and hitchhiking on host chromosomes and vertically propagating throughout cell division (Coursey and McBride, 2019; Kim et al, 2020). In contrast, few examples of plasmids naturally present in eukaryotic nuclei have been documented (Esser et al, 2012). To our knowledge, only two families of natural plasmid present in eukaryotic nuclei have been clearly identified and characterized: the *Saccharomyces cerevisiae* 2µ plasmid (Sau et al, 2019) and derivative in other ascomycota species, and the Ddp plasmid family in the social amoeba *Dictyostelium discoideum* (Rieben et al, 1998; Shammat et al, 1998). The reason for such a small number of plasmids is not clear: either these objects have been less studied and are overlooked in sequencing data, or eukaryotes have developed effective protection and/or defence mechanisms. In either case, it is not yet clear how these molecules are maintained across generations.

The 2µ plasmid is one of the most studied examples of a selfish DNA element, i.e. a molecule that does not appear to confer any fitness advantage on its host, without imposing a significant cost (Mead et al, 1986). The 6.3 kb sequence, named after its contour length when observed with electron microscopy (Sinclair et al, 1967), is chromatinized (Nelson and Fangman, 1979; Livingston and Hahne, 1979) and replicates using the host replication machinery (Zakian et al, 1979). It is present in most *S. cerevisiae* natural isolates and laboratory strains (Peter et al, 2018) suggesting a very efficient and successful persistence mechanism. It encodes a partitioning system that includes Rep1 and Rep2, two proteins that associate with the plasmid STB repeat sequence that is essential for plasmid stability (McQuaid et al, 2019) and a specific recombinase Flp1 (Be and Wl, 1985). Several works point to a 'chromosome hitchhiking' mechanism by which the plasmid binds to the chromosomes of the host, in a Rep1-dependent manner, taking advantage of its segregation machinery during cell divisions (Sau et al, 2019). Microscopy investigations show that the 2µ plasmid colocalizes with the host chromatin, though its precise nuclear

[1]Institut Pasteur, CNRS UMR 3525, Université Paris Cité, Unité Régulation Spatiale des Génomes, 75015 Paris, France. [2]Sorbonne Université, Collège Doctoral, F-75005 Paris, France. [3]Département de Biologie, Université Paris-Saclay, ENS Paris-Saclay, 91190 Gif-sur-Yvette, France. [4]Institut Curie, PSL University, Sorbonne Université, CNRS UMR 3664, Nuclear Dynamics, Paris, France. [5]Institut de Biologie de l'Ecole Normale Supérieure, Département de Biologie, Ecole Normale Supérieure, CNRS, INSERM, PSL Research University, Paris, France. ✉E-mail: romain.koszul@pasteur.fr; axel.cournac@pasteur.fr

localisation remains unclear, with studies suggesting either a preferential position at discrete chromatin loci distinct from telomeres (Scott-Drew and Murray, 1998), at the centre of the nucleus (Heun et al, 2001), or at the nuclear periphery close to telomeres (Kumar et al, 2021).

De novo calling of DNA contacts between molecules is difficult to achieve with microscopy but can be done using capture of chromosome conformation approaches such as Hi-C (Lieberman-Aiden et al, 2009). We therefore used Hi-C and Micro-C data (Hsieh et al, 2016) to map and quantify the plasmid physical contacts with the host chromosomes. We show that the plasmid contacts a set of discrete loci that remain remarkably stable in multiple growth and mutant conditions. The set of contact hotspots corresponds to relatively long, inactive regions. These contacts can evolve within a few minutes following environmental stress and depend on the nucleosomal H4 basic patch. Strikingly, a Mb long inactive artificial bacterial chromosome was able to bind plasmid molecules from their native binding positions, illustrating how the plasmid is spontaneously attracted by inactive chromatin, whatever its origin. Overall, these results point to the existence of a segregation mechanism whereby the plasmid may recognize a signal associated with chromatin structure to bind to the inactive chromatin regions of its hosts in a reversible way.

Other Saccharomycetaceae, such as the Lachancea species *Lachancea fermentati* and *Lachancea waltii* that have diverged from Saccharomyces for over 100 My ago also display episomes homologous to the 2μ. Further in the phylogenetic tree, the amoeba *D. discoideum* contains the Ddp5 plasmid. We show that all these episomes also preferentially tether to long inactive regions. Since Amoeba belongs to Amoebozoa, a group that preceded divergence with Opisthokont that encompasses animals and fungi, these results suggest that this property may be a widespread strategy for eukaryotic plasmids to ensure their correct segregation during cell division in a way that does not disrupt host regulation.

# Results

## Chromatinization and 3D folding of the 2μ plasmid

The 2μ plasmid sequence is usually filtered and overlooked from high-throughput sequencing data, including Hi-C. We therefore revisited ten years of datasets to explore the composition and behavior of this 6.3 kb molecule. Using Micro-C, a high-resolution Hi-C derivative that quantifies DNA contacts at the nucleosomal level (Hsieh et al, 2016; Swygert et al, 2019), we generated a plasmid contact map at 200 bp resolution from cells in exponential growth (Fig. 1A; Methods). The cis-contacts map revealed four small self-interacting regions corresponding to the four plasmid genes, reminiscent of the pattern observed at the level of active chromosomal genes (Hsieh et al, 2016). The ~1 kb STB-ORI region, positioned in between the *RAF1* and *REP2* genes, appeared as a constrained region with a stronger local enrichment in short range contacts. H3 chemical cleavage data (Chereji et al, 2018) shows the regular distribution of nucleosomes along the genes, with nucleosome-free regions (NFR) at transcription start sites (TSS), and an even distribution along the open reading frames (Fig. 1B, Appendix Fig. S1A), showing that the chromatin of the plasmid is highly similar to that of its host (Nelson and Fangman, 1979;

Livingston and Hahne, 1979). RNA-seq data highlight the moderate transcriptional activity of the 4 genes and the presence of non-coding RNA as previously identified (Broach et al, 1979) (Fig. 1B). Taking into account the number of copies per cell of the 2μ plasmid, the level of transcription of the 4 genes of 2μ plasmid in standard growth condition is 2 times less than the median expression of genes for its host (Appendix Fig. S1B). Chromatin accessibility assessed by ATAC-seq confirms that the intergenic regions are the most open regions of 2μ plasmid (Fig. 1B). The cis-acting plasmid partitioning locus STB appears to be the most accessible region, which supports the notion that it acts as a gateway for known recruited host proteins (Chan et al, 2013). Note that no enrichment of the centromeric histone H3-like protein Cse4 was found on the plasmid sequence, notably at STB region (Appendix Fig. S1A). We also did not detect any enrichment of cohesin and condensin complexes at the STB region (Appendix Fig. S1A). Overall, these signals confirm that the chromatin composition and organisation of the 2μ plasmid is very similar to that of its host's chromosomes (Nelson and Fangman, 1979).

## Discrete contacts between the 2μ plasmid and host chromosomes

To directly monitor the contacts made by the 2μ with the yeast genome, we plotted the relative contact frequencies between the plasmid and 16 yeast chromosomes from exponentially growing cells (Swygert et al, 2019) (Fig. 1C, blue curve; Fig. EV1) curves can also be represented using a chromosomal heatmap diagram coloured along its linear axis according to a scale that reflects contact frequencies (Fig. 1C,D, Methods).

The 2μ plasmid contacts with the hosts chromosomes were not evenly distributed, as reflected by the curve peaks (dotted black boxes) and darker stripes along the chromosomal diagrams that represent hotspots of contacts (Fig. 1C, black triangles). A peak-calling automated approach yielded 73 hotspots along the 16 host chromosomes, with most (59/73) located within chromosome arms (Methods). 14 hotspots overlapped subtelomeric regions (defined as the 30 kb at chromosome extremities), a significant enrichment compared to random group realisations (*p*-value = 0.001, Methods). Note that this number is maybe slightly underestimated because of the difficulty to align sequencing reads in *S. cerevisiae* subtelomeric regions because of their repeated nature. In contrast with past findings drawn from imaging approaches, centromeres were on average depleted in contact with the 2μ (Fig. 1E). The contact pattern measured by Hi-C correlates very well with the Rep1 occupancy signal, as measured by ChIP-seq (Appendix Fig. S2), supporting that the plasmid tends to be positioned in the close vicinity of these discrete loci within the yeast genome.

In contrast, a plasmid carrying a yeast centromeric sequence colocalizes near the spindle pole body (SPB) with the 16 yeast centromeres and displays enrichment of contacts with these regions (Appendix Fig. S3A). On the other hand, a replicative plasmid devoid of centromere (pARS) displays relatively even contacts throughout the genome and does not show contact enrichment around the regions identified with the 2μ plasmid (Fig. 1D,E; Appendix Fig. S3B). Altogether, these data show that the 2μ makes specific, discrete contacts with dozens of loci interspersed over the entire genome, excluding pericentromeric regions.

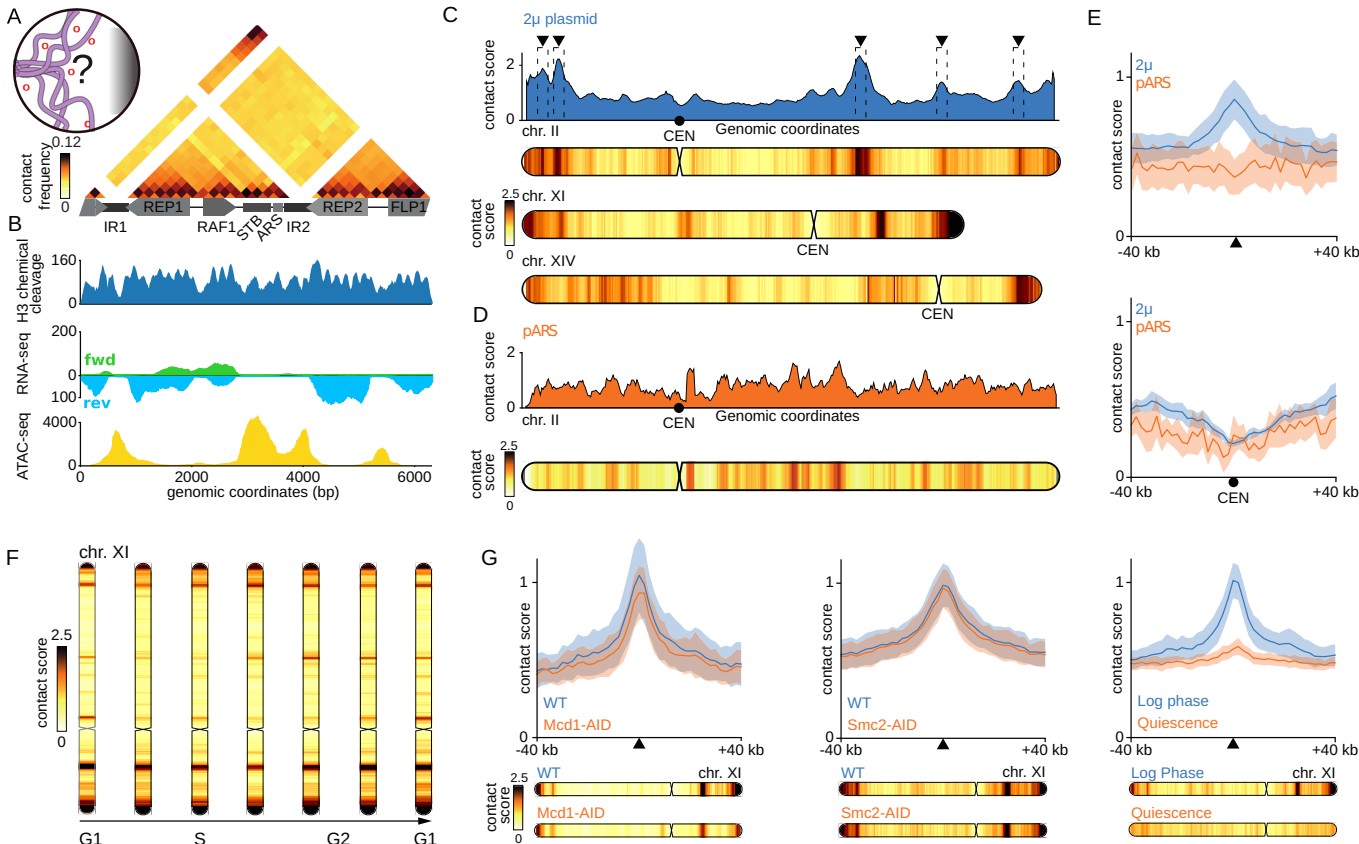

**Figure 1. Specific positioning of the natural 2μ plasmid on several genomic regions of *S. cerevisiae* chromosomes.**

A Contact map of the 2μ plasmid with bins of 200 bp based on Micro-C XL data in asynchronous cells (log phase). The 4 genes and cis-acting sequences of the 2μ plasmid genome are annotated below the map like Stability Sequence (STB) and the 2 Inverted Repeats (IR) which are non-mappable regions. B H3 chemical cleavage giving the nucleosome density, RNA-seq giving transcription level on forward and reverse strands and ATAC-seq giving chromatin accessibility along the 2μ plasmid in counts per million (CPM). C Contact profile of 2μ plasmid with several chromosomes of *S. cerevisiae* (binned at 2 kb). D Contact profile of pARS plasmid (containing no 2μ or centromere systems) for chromosome II. E Averaged contact signal of the 2μ and pARS plasmids at the positions automatically detected in WT, log phase for the 2μ plasmid (top) and at centromeres positions (below). F Contact profile of the 2μ plasmid with chr XI during the mitotic cell cycle. G Averaged contact signal of the 2μ plasmid in mutants depleted in Mcd1 (cohesin subunit), mutant depleted in Smc2 (condensin subunit) and in quiescence state.

## Rep1 and the STB sequence are necessary for binding specificity and stability

The dimer Rep1/Rep2 has been shown to associate with the plasmid STB sequence to promote partitioning during cell division and both STB and Rep1/Rep2 are required for the plasmid stability in host cell (McQuaid et al, 2019b; Mereshchuk et al, 2022). We therefore tested whether the distribution of contacts along the genome was dependent on these partners by generating Hi-C data of cells with a 2μ plasmid mutants lacking either *REP1* gene or the STB sequence (Kikuchi, 1983; McQuaid et al, 2019b) (Methods). These mutant plasmids are unstable (McQuaid et al, 2019b) so a pKan plasmid version was used which contains the *kanMX4* gene cassette that disrupts *FLP1* gene to maintain the plasmids in the cells. This instability is reflected by a low proportion of reads from 2μ in each library (Appendix Fig. S4, Appendix Table S3). In both mutants, there is no contact enrichment around the previously identified hotspots (Appendix Fig. S5A, B). On the other hand, a plasmid mutant lacking only *FLP1* gene remains enriched at the identified hotspots (Appendix Fig. S5C) which indicates that this recombinase

is not involved in contact specificity. These experiments show that the Rep1 protein and the STB sequence are essential for the establishment and/or maintenance of contacts between 2μ and specific regions of its host chromosomes.

## Plasmid-chromosome contact regions are stable under a range of conditions

The diversity of published Hi-C and Micro-C experiments and the ubiquitous nature of the 2μ plasmid in yeast strains allowed sifting through existing data to detect variations in contact patterns between different genetic backgrounds, cell cycle stages, growth, and metabolic states. Laboratory (W303 and S288C) and natural strains (BHB and Y9) (Peter et al, 2018) displaying high, or low, copy number of plasmids respectively, presented nearly identical contact hotspots profiles (Fig. EV2A) (Methods). For strain Y9, which does not naturally possess the 2μ plasmid, we used a pKan plasmid containing the kanMX4 gene cassette that disrupts the *FLP1* gene. Furthermore, most hotspots were conserved throughout the mitotic cell cycle with a few subtle changes (Fig. 1F)

(Costantino et al, 2020; Lazar-Stefanita et al, 2017) as well as during meiosis cell cycle (Appendix Fig. S6) (Muller et al, 2018; Schalbetter et al, 2019). A small general increase in contact variability was observed at the later stages of meiotic divisions (Appendix Fig. S6), which could reflect increased compaction of chromosomes or change in chromatin state at this step.

The hotspot pattern was also maintained upon degradation of chromatin-associated protein complexes including members of the structural maintenance of chromosome (SMC) family cohesin and condensin (Fig. 1G) (Bastié et al, 2022; Guérin et al, 2019) that have been proposed to be involved in 2μ plasmid stability (Kumar et al, 2021). The hotspot pattern is also present for a mutant depleted for the Smc5-Smc6 complex (Fig. EV2C) (Jeppsson et al, 2024); in cells lacking the silencing complex member Sir3 (Fig. EV2C) that is able to act as a bridging complex (Ruault et al, 2021); or in cells depleted for the DNA replication initiation factor Cdc45 that reach mitosis without replicating (Dauban et al, 2020) (Fig. EV2C). The pattern of interaction was also conserved in cells treated with nocodazole (Fig. EV2C). As previous reports pointed that nocodazole treatment impaired the recruitment of Cse4 (Hajra et al, 2006), Kip1 or microtubule-associated proteins Bim1 and Bik1 (Prajapati et al, 2017), this result suggests that those host factors are not involved in the plasmid binding to chromosomes.

The contact pattern of the 2μ appears maintained in other biological conditions, including in presence of a double-strand break (DSB) (Piazza et al, 2021), hydroxyurea (HU) (Jeppsson et al, 2022) and in different genetic mutants (Fig. EV2C). In sharp contrast, most hotspots were strongly attenuated in quiescent cells (Guidi et al, 2015; Swygert et al, 2021) (Fig. 1G), when the cells dramatically alter their transcription programme (McKnight et al, 2015) and genome organisation (Guidi et al, 2015; Swygert et al, 2021). The later observation prompted us to explore more closely the links between transcription and plasmid-chromosomal contacts.

### The 2μ preferentially tethers to inactive regions along the host's chromosomes

To explore the links between transcription and plasmid contacts (Fig. 2A,B), we first plotted the individual hotspots windows ordered by peak strength along with the corresponding transcription level and gene size annotation. We piled-up the contacts made by the 2μ plasmid, and 80 kb windows centred on the 73 peaks called on the contact profile, along with the pile-up transcription pattern of the 73 regions, revealing a strong depletion centred on the contact hotspot (Fig. 2C; Methods). This analysis also reveals strong contact enrichment with poorly transcribed regions often extending over several kbs and often corresponding to relatively long genes (e.g., >4 kb, to compare to a genome-wide gene size median of 1 kb) (Fig. 2A,B) (Methods). They include 15 of the 19 genes longer than 7 kb present in the entire genome (p-value < 0.001, Methods, Fig. EV2 for the annotations of long genes and 2μ contact signal) that are lowly expressed in these growth conditions, with the remaining four being transcribed. A statistical analysis shows more generally that the 2μ contacts are correlated with the transcription level and size of the gene (Appendix Fig. S7) (Methods). The average GC content of the hotspots sequences is slightly lower than the genome average (36.8% versus 38.2%) (Fig. 2D). No consensus was identified when processing hotspots sequences using MEME algorithm (Bailey et al, 2015) (Methods). A

magnification of contact distribution over the long inactive genes revealed a maximum enrichment in the middle of the weakly expressed gene (e.g., DYN1 gene in Fig. 2A,E). In addition, the regions contacted by the 2μ are not enriched in cis or trans contacts with each other, suggesting they do not colocalize in the nuclear space (Fig. 2F; Appendix Fig. S8A). The regions contacted by 2μ tend to make less inter-chromosome contact with each other than random groups (Fig. 2F). This may be explained by the fact that 2μ plasmid contact hotspots at centromeres are rare. The contact signal between chromosomes is strongest between centromeres due to the Rabl configuration (Duan et al, 2010), so 2μ plasmid contact sites make less contact with each other than average. Finally, the contact signal measured by Micro-C reveals an enrichment in short-range contacts along the diagonal at the hotspots positions (Appendix Fig. S8B) which could be interpreted as a singular chromatin state such as an accumulation of nucleosomes.

### Plasmid tethering is quickly reversible

We then explored the dynamics of plasmid chromosome anchoring. To do this, we induced heat shock stress, known to modify transcriptome, chromatin state and protein-genome interactions (Kim et al, 2010; Vinayachandran et al, 2018) by transferring exponentially growing cells at 25 °C to 37 °C medium (Methods). Five minutes after heat shock, changes in the contact signal of 2μ plasmid were already observed. For instance, the contacts between the 2μ plasmid and the UTP20 gene (with a size of 7.4 kb) were strongly increased (Fig. 2G), while contact enrichment at the locus of FIR1 and ZRG8 genes (with sizes of 2.6 kb and 3.2 kb) disappears (Fig. 2G). Precise kinetics with 4 time points show how contact points can appear (Fig. EV3A) or disappear (Fig. EV3B) in a matter of minutes. These results show that the plasmid can relocalize quickly to discrete regions.

### 2μ plasmid can tether to exogenous artificial inactive chromatin

To further support the relationship between chromatin inactivity and 2μ contacts, we explored the plasmid behaviour in presence of a Mb-long exogenous sequence. The plasmid positioning in strains carrying the linearized sequence of the Mycoplasma mycoides (Mmyco) chromosome as supernumerary, artificial chromosome, was investigated using Hi-C (Meneu et al, 2025; Lartigue et al, 2009) (Fig. 2H; Appendix Fig. S9B). Mmyco presents highly divergent sequence composition, as reflected by its GC content (24% to compare with yeast 38%). It is chromatinized by well-formed nucleosomes, imposes little fitness cost to the yeast, and segregates properly. Mmyco AT-rich sequence is devoid of transcription and is depleted in RNA Pol II (Meneu et al, 2025). Strikingly, contacts between the 2μ plasmid and the entire length of Mmyco's inactive 1.2 Mb sequence were 6-fold higher than the average value on wild-type chromosomes. Our past work showed that the transcriptionally inactive Mmyco chromosome adopts a globular shape at the nuclear periphery (Meneu et al, 2025). In absence of Mmyco, the plasmid appears as several foci distributed in the nucleoplasm, as previously reported (Heun et al, 2001; Velmurugan et al, 1998). In contrast, in the presence of the Mmyco, most of the 2μm signal concentrates and colocalizes with the bacterial DNA (Fig. 2I; Appendix Fig. S10). This result

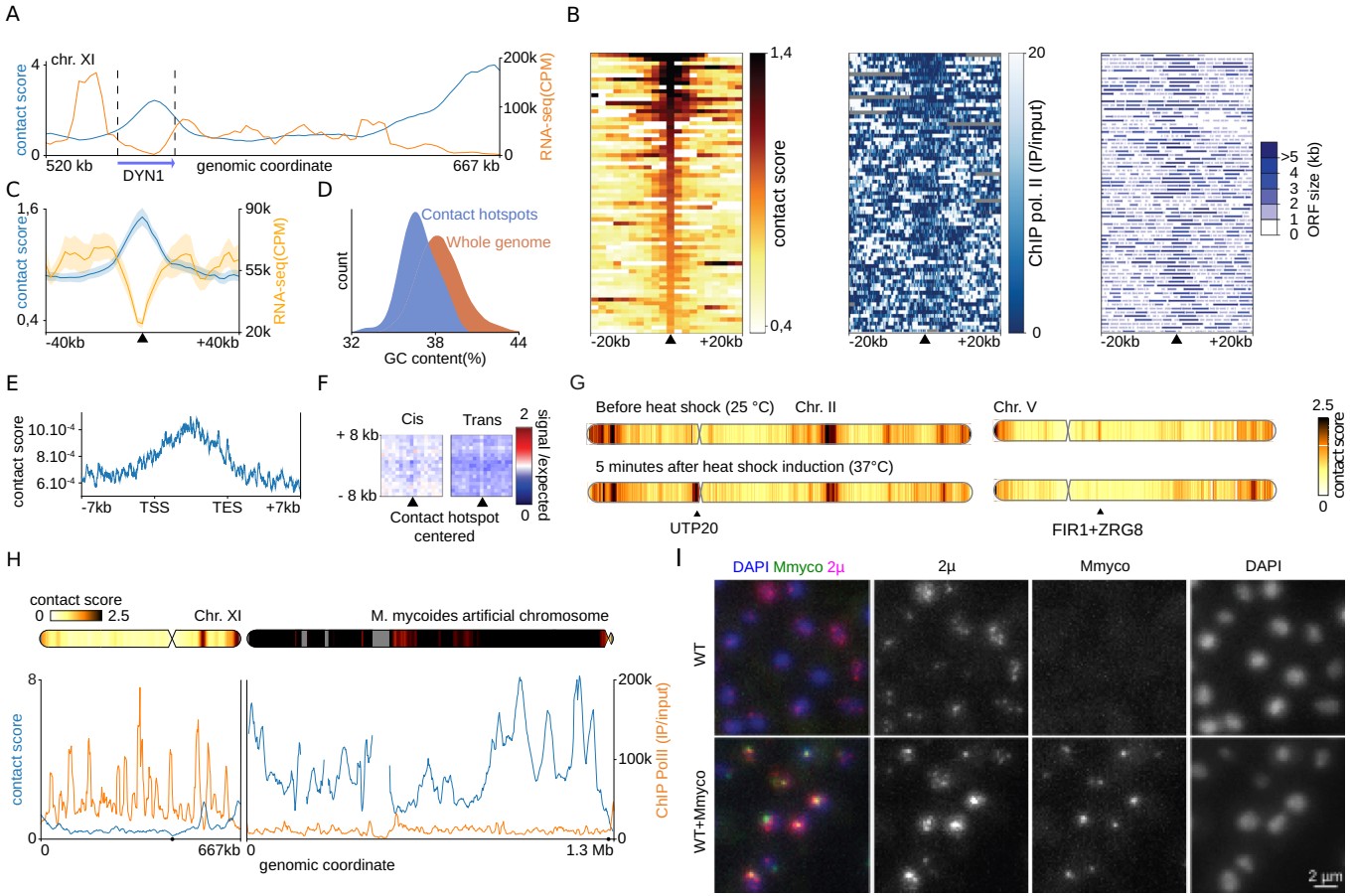

**Figure 2. The contacted regions are depleted in transcription and more frequent at genes with long sizes.**

(A) Transcription signal and contact profile of the 2 μ plasmid in a region of the Chr. XI. (B) Heat maps of 2μ plasmid contact signals, transcription level (RNA-seq in CPM) and gene structure sorted in descending order according to contact scores over the region −20 to +20 kb around the peaks of contact of 2μ plasmid. (C) Averaged contact signal of the 2 μ plasmid and transcription at the positions of contact automatically detected in WT, log phase for 2μ. (D) Distribution of GC content for the group of sequences contacted by the 2 μ plasmid and for the whole genome of *S. cerevisiae*. (E) Contact profile of 2μ plasmid along long genes (>7 kb), binned at 200 bp. (F) Mean profile heatmap between hotspots of contact with 2μ plasmid belonging to the same (left) or different chromosome (right). (G) Contact profile of the 2 μ plasmid before and 5 min after a heat shock for Chr. II and Chr. V. Examples of regions where contact intensity varies significantly are marked with a vertical arrow. (H) Contact profile of the 2μ plasmid in strains containing an additional bacterial chromosome *M. mycoides* as well as the transcription profile (ChIP Pol II/input). (I) Representative fluorescent images (Z-stack projection) of FISH (Fluorescence In Situ Hybridization) experiments using a probe specific for the *M. mycoides* genome and a probe specific for the 2μ sequence. The probes were hybridized on a wild-type strain or a strain carrying the *M. mycoides* genome fused to chromosome XVI. The scale bar is 2 μm. Source data are available online for this figure.

demonstrates that the Hi-C data reflects titration of the 2μ plasmid from their hotspots by the long inactive sequence and shows that transcriptional inactivity is one of the primary conditions of plasmid relocalization to a sequence. In addition, this result also demonstrates that the contacts quantified using Hi-C between the 2μ plasmid and genomic DNA do indeed correspond to physical relocalization of the molecules.

We also analysed contacts between the 2μ and an artificial 9 kb array consisting of 200 lacO binding sites derived from *Escherichia coli* and introduced in chromosome VII (Guérin et al, 2019). The LacO array, which has a GC content of 41%, is not transcribed and is recognized by the DNA-binding repressor LacI put under the control of an inducible promoter (Guérin et al, 2019). When LacI is not present, we observe an enrichment of contact but upon LacI binding, the array is not a contact hotspot any more (Appendix Fig. S9A). These observations further support that a long (9 kb)

inactive region from a different organism but with a similar GC content than *S. cerevisiae* can be a contact hotspot for the 2μ plasmid. It has been shown that a high level of LacI binding results in nucleosome eviction (Loïodice et al, 2021). The observation that specific contact is lost when the LacI protein is attached to the region suggests that the resulting large nucleosome-free region could be responsible for the detachment of the 2μ plasmid.

## Characterisation of hotspots of contact of the 2μ plasmid

To further characterise the composition of the regions contacted by the 2μ plasmid, we computed the average enrichment of various genomic signals (ChIP-seq, MNAse-seq, ATAC-seq) over the 73 contact hotspots. For example, we computed the histone H3 ChIP-seq average signal and observed an enrichment at the hotspots (Fig. 3A). In agreement with the transcriptional inactivity of the

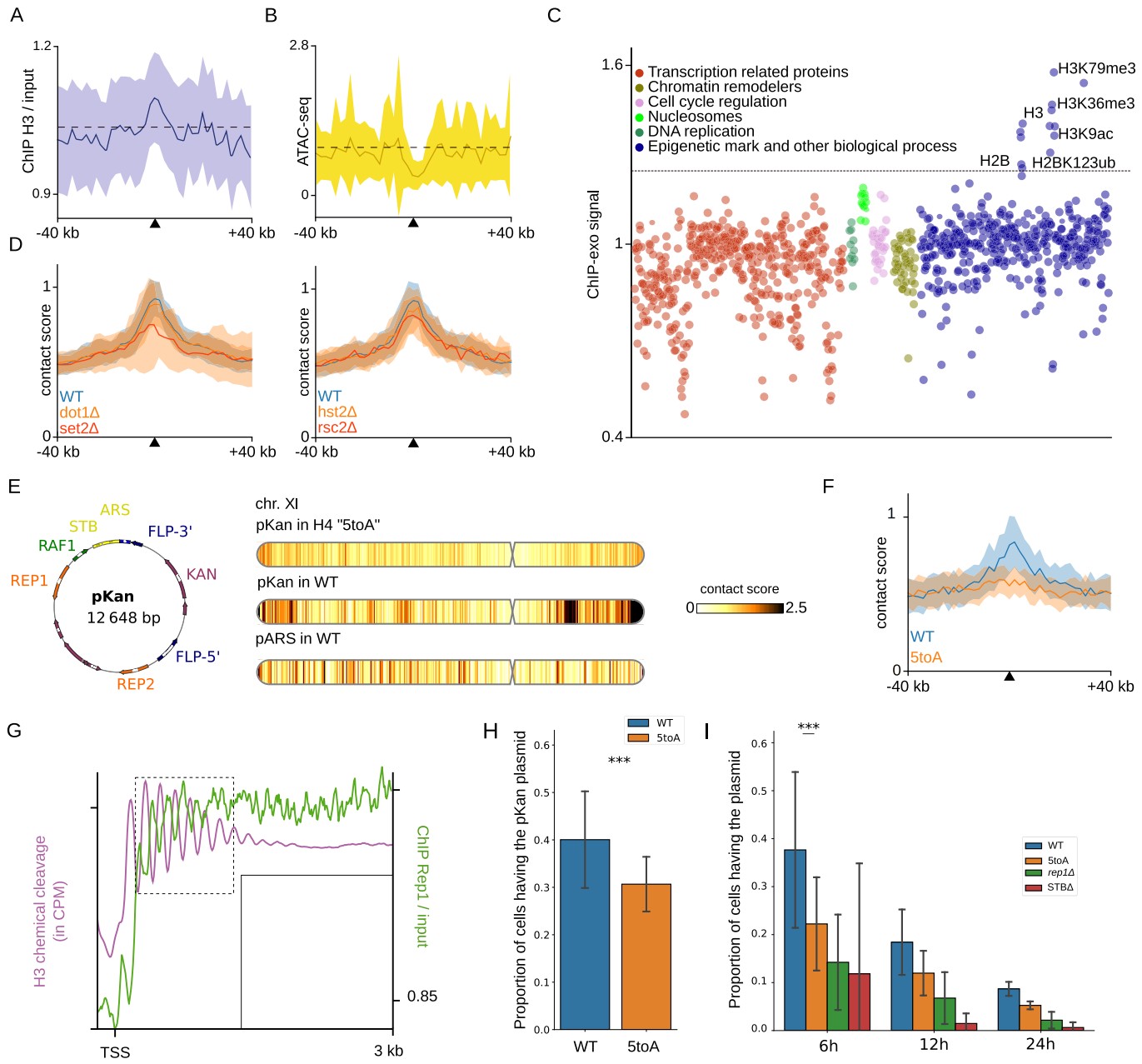

**Figure 3. Specific positioning may be associated with nucleosome signal.**

(A) Averaged signal at the hotspots of contact with 2μ plasmid for nucleosome occupancy (ChIP-seq of H3 histone) and (B) chromatin accessibility (ATAC-seq). (C) Average value of signal at the hotspots of contact for 1251 ChIP-exo libraries sorted by general categories (Rossi et al, 2021). (C) Average value of signal at the hotspots of contact for 1251 ChIP-exo libraries sorted by general categories (Rossi et al, 2021). (D) Averaged contact signal of the 2μ plasmid in mutants of epigenetic marks dot1Δ, set2Δ and mutant in chromatin remodeler rsc2Δ and in deacetylase mutant hst2Δ. (E) Contact profile of the 2μ plasmid (pKan version) with chromosomes of *S. cerevisiae* in H4 5toA mutant, WT and of the pARS plasmid in WT for chromosome XI. (F) Averaged contact signal of the 2μ plasmid in H4 5toA mutant and control. (G) Averaged signal around TSS for nucleosomes and Rep1 occupancy signals. (H) Stability measurements showing inheritance of the 2 μm-based (pKan) plasmids in a *cir⁰* yeast strain determined by plating assays for WT and 5toA mutant after overnight culture (O/N). Results represent the average (± standard deviation) from assaying 5 replicates for each condition (p-value = 4.77e−18). Asterisks indicate significant differences determined by a Chi-square test (*P < 0.05, **P < 0.005, ***P < 0.0005). (I) Stability measurements showing inheritance of the 2 μm-based (pKan) plasmids in a *cir⁰* yeast strain determined by plating assays for WT, 5toA, rep1Δ and ΔSTB mutants. Results represent the average (±s.d.) from assaying 6 replicates for each condition at 3 different time points (6 h, 12 h, 24 h) after O/N culture (with the respective p-values: 8.46e−290, 4.11e−14 and 5.57e−26). Asterisks indicate significant differences between WT and 5toA mutant determined by a Chi-square test (*P < 0.05, **P < 0.005, ***P < 0.0005). Source data are available online for this figure.

hotspots, chromatin accessibility (ATAC-seq) (Lee et al, 2018) shows that these regions are less accessible compared to the rest of the host genome (Fig. 3B). We explored many other different genomic signals, most of which do not show an enrichment signature (Appendix Fig. S11). Some, however, show a specific average signal: the GapR ChIP-seq signal, which indicates the presence of positive supercoiling (Guo et al, 2021), is depleted in the regions contacted by the 2μ (Appendix Fig. S11A). Also, unexpectedly, dinoflagellate-viral-nucleoproteins (DVNPs) expressed in yeast show enrichment in regions contacted by 2μ (Appendix Fig. S11A). These proteins are supposed to localise to histone-binding sites (Irwin et al, 2018).

To better determine the chromatin composition of the chromosomal regions contacted by the 2μ, we took advantage of a recently generated ChIP-exo dataset to screen for the deposition of ~800 different proteins or histone marks along the *S. cerevisiae* genome (Rossi et al, 2021). For each genomic signal, the deposition profiles over the 73 contact hotspots were aggregated and tested for enrichment or depletion (Methods). In agreement with the low activity of the tested regions, most of the proteins associated with active transcription (e.g., general transcription factors or proteins of the SAGA complex) were depleted (Fig. 3C; Appendix Fig. S11B). On the other hand, histones H3, H2B, and histone marks like H3K79me3 and H3K36me3, were the only signals that were enriched over the contact hotspots (Fig. 3C). We tested for the influence of both marks by characterising the plasmid contacts in absence of either Set2 or Dot1 methylase. Absence of Dot1, responsible for H3K79 methylation, did not affect plasmid hotspots of contact (Fig. 3D; Appendix Fig. S12A). In absence of Set2, which methylates H3K36, long genes are known to be derepressed (Li et al, 2007). Although contact specificity for the 2μ plasmid is still detectable in this mutant (Fig. 3D; Appendix Fig. S12A), many long genes have their contact signal with the 2μ plasmid greatly reduced while the level of transcription is slightly increased (Appendix Fig. S12B,C). Therefore, these experiments further confirm that transcription activity is linked to plasmid contact profile.

The 2μ plasmid contact profile with the genome is also independent of remodeler complexes Rsc2 or Rsc1, and the histone deacetylase protein Hst2 (Appendix Fig. S12D). Rsc2 was shown to be essential for the 2μ plasmid segregation and to overcome maternal inheritance bias (Wong et al, 2002). We observed indeed a significant drop in the proportion of reads from plasmid in this mutant (Appendix Fig. S4). However, its attachment to host chromosomes remain unchanged (Appendix Fig. S12D) suggesting that the mode of action of Rsc2 is not directly linked to plasmid attachment. The Rsc2 protein is part of the RSC chromatin remodelling complex and can affect the regulation of transcription of numerous genes. It is possible that the reported effect of Rsc2 on 2μ plasmid segregation can be explained by an indirect effect, such as impact on the expression of genes required for proper 2μ plasmid maintenance. It was also shown for non-centromeric DNA circles that Hst2 deacetylase was important for their condensation and propagation to daughter cells (Kruitwagen et al, 2018). However, we did not detect any changes in the contact profile of the *HST2* deletion mutant (Appendix Fig. S12D). Finally, the 2μ contact profile with the genome is also independent of the main chromatin remodelers, as shown by the lack of variation that follows degradation using auxin-inducible degron (AID) of *SPT6, ISW1, SWR1, FUN30, INO80, CHD1,* and *ISW2* (Jo et al, 2021) (Appendix Fig. S12D).

## Plasmid anchoring depends on the H4 basic tail

A recent study suggested that chromatin folding at the nucleosomal level is altered when five basic amino acids (aa) of histone H4 tail (basic patches aa 15 to 20) are converted into alanine in a mutant called H4 5toA (Swygert et al, 2021). This change takes place without affecting transcription of the contact hotspots identified in WT condition (Fig. EV4A). The 2μ plasmid contacts with chromosomes in the H4 5toA mutant were quantified using Hi-C (Figs. 3E and EV4B,D,E). We used an artificial version of the 2μ plasmid, called pKan, which differs from the natural version in that it contains a KAN gene and does not contain the FLP1 recombinase gene (Fig. 3E). The specific recombinase Flp1 is therefore no longer present in this type of plasmid. No more contact enrichment on hotspots was detected (Figs. 3F and EV4B), suggesting that the 2μ plasmid contact hotspots depend on the presence or composition of the H4 tail basic patch. If we compare the contact peaks detected for the pKan plasmid in WT background and the pKan in the 5toA background, the 5toA mutant has only 24 detected peaks and only 3 common peaks with the 36 peaks detected for the pKan plasmid in the WT background. Importantly, in the H4 5toA mutant, transcription is not increased on the contact hotspots compared to WT condition (Fig. EV4A). This result shows that the plasmid-chromosome contacts can be suppressed not only by transcription activation, but also only through an alteration of the nucleosome H4 tail basic patch. The same analysis was replicated in quiescent cells with the natural version of the 2μ plasmid: in that condition, the remaining contact specificity between the 2μ plasmid and the 73 hotspots is also lost (Fig. EV4D), suggesting the tail patch or more generally chromatin structure plays an important role in their maintenance. A careful analysis showed that the Rep1 ChIP-seq signal is 90° phase-shifted with nucleosome position (Fig. 3G) suggesting that Rep1 is not randomly contacting the host chromatin but is positioned in relation to the distribution of nucleosomes along the chromatin fibre.

## Loss of plasmid anchoring is associated with reduced stability

To test the stability of the plasmid in the H4 5toA mutant, we measured the proportion of cells containing the artificial pKan plasmid after overnight culture in a selective medium (Fig. 3H, Methods, as described in (McQuaid et al, 2019)). Note that only ~40% of the pKan plasmid was found in WT in the stability assay. This reduced stability of this artificial plasmid could have a different origin that remains unknown, such as the absence of the specific recombinase Flp1 which can correct a small number of plasmid copies, or the presence of the expressed *KAN* gene which could alter the plasmid's topology and attachment mechanism.

We observed a difference between the wild-type strain and the H4 5toA mutant (average proportion of cells having the plasmid, in WT ~ 40% vs mutant H4 5toA ~ 31%, *p*-value < 0.001, Chi squared test, see Methods). We also calculated this proportion along 3 time-point kinetics in non-selective media at 6 h, 12 h and 24 h after the overnight culture. The H4 5toA mutant resulted in a decrease of the 2μ reporter plasmid. This decrease was not as dramatic as the known effect of Rep1 and STB mutations, but remained significant (Fig. 3I). We also detect a slight but significant change in the average number of plasmids per plasmid-possessing cell

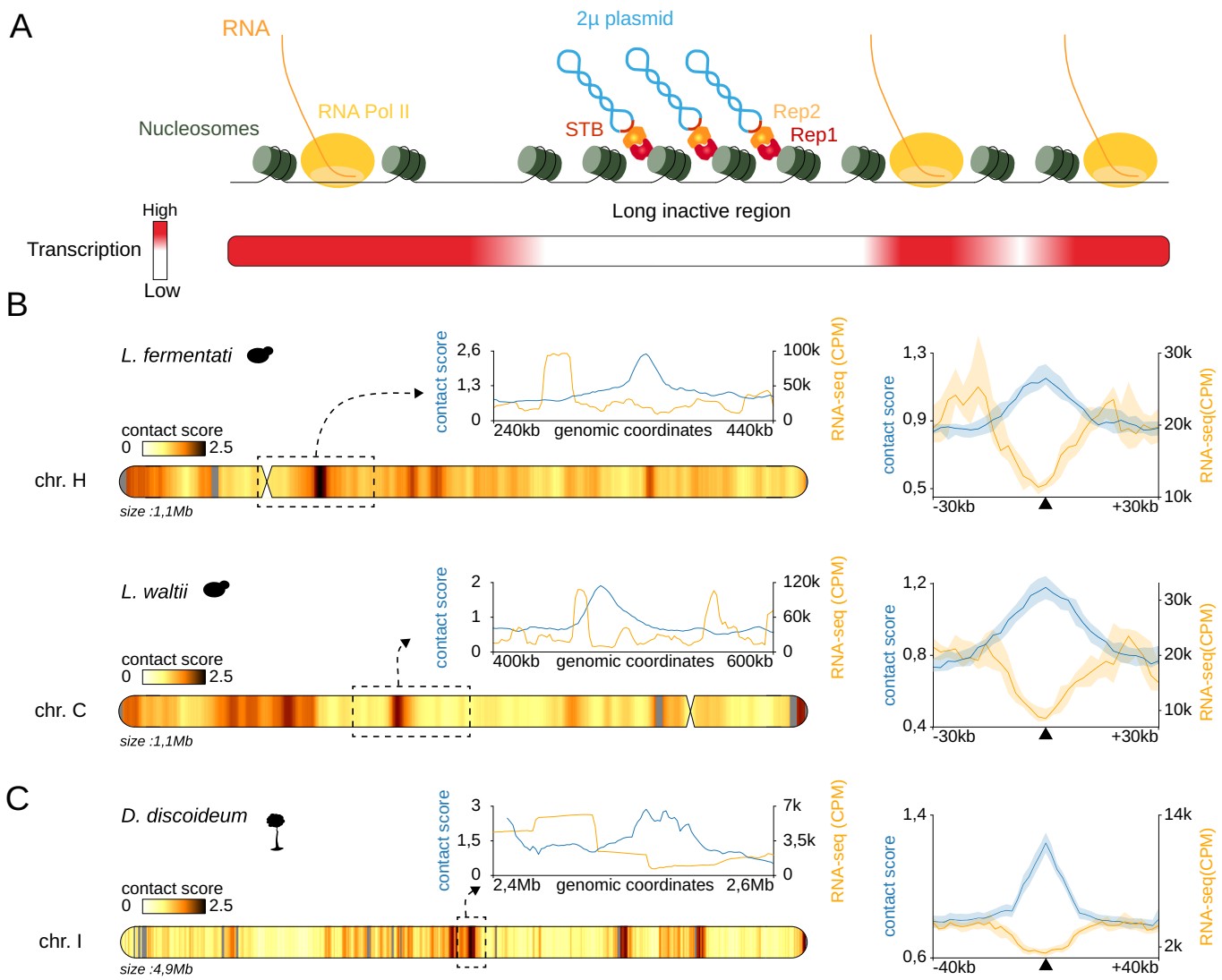

**Figure 4. Contact specificity is also present in parasitic plasmids of other eukaryotes.**

(**A**) Proposed model involving plasmid proteins Rep1/Rep2 and nucleosome signal to ensure attachment between plasmid 2μ and specific loci on host chromosomes (**B**) Contact profiles of natural plasmids pSM1 and pKW1 with host chromosomes of the yeasts *Lachancea fermentati* and *Lachancea waltii*. Averaged contact and transcription signals at the loci detected as peaks of contact for each natural plasmid (on the right). (**C**) Contact profiles of natural plasmid Ddp5 with host chromosomes of the social amoeba *Dictyostelium discoideum*. Averaged contact and transcription signals at the loci detected as peaks of contact for each natural plasmid (on the right).

(Fig. EV4F). These observations may reflect perturbed segregation of the pKan plasmid in the H4 5toA mutant. The drop in stability of the pKan plasmid in the H4 5toA host mutant is significant, but does not reach the dramatic effect of both the rep1Δ and ΔSTB plasmid mutations, two conditions where a key plasmid component of the tethering system is abolished. One interpretation could be that, although contact specificity is lost in the H4 5toA mutant, non-specific interactions or tethering to host chromosomes can persist, which could help maintain some segregation ability.

Taken together, these results point to a model in which Rep1/Rep2 proteins localise to long regions that both are transcriptionally inactive and display a nucleosome signal carried by the H4 tail (Fig. 4A). Note that these two features could be two faces of the same coin, since transcription disturbs nucleosome distributions and could affect the histone signal specificity.

## Other eukaryotic plasmids tethers to inactive chromatin

To assess whether this mechanism of binding to inactive sequences might concern other eukaryotic plasmids with "selfish" appearance, we analysed contact profiles of *Lachancea fermentati* and *Lachancea waltii* which also host natural plasmids related to the 2μ plasmid. In these 2 organisms, we can also detect foci of contacts distributed across all the host chromosomes, far from the centromeres, and with also a bias towards long genes (Figs. 4B and EV5A,B). Automatic detection of contact peaks identified 52 peaks for *L. fermentati* and 55 peaks for *L. waltii*. For *L. fermentati*, of the 18 long genes (size > 7 kb), 12 are detected as plasmid contact hotspots (*p*-value < 0.001, Methods) and for *L. waltii*, of the 19 long genes (size > 7 kb), 8 are detected as plasmid contact hotspots (*p*-value < 0.001, Methods). We also calculated the average transcription signal profile using RNA-seq, revealing a

depletion in transcription at the contact hotspots of these natural plasmids (Fig. 4B, right).

We also quantified the contacts made by the Ddp5 plasmid with the chromosomes of its host, the amoeba *D. discoideum* (Rieben et al, 1998). We performed Hi-C experiment on *D. discoideum* cells in vegetative state and measured the trans contacts between Ddp5 plasmid and its host chromosomes similarly to the 2 μ plasmid experiment (Figs. 4C and EV5C,D). A hundred hotspots or so were detected on the contact profile (Methods). Of the 76 long genes (size >10 kb) annotated along this genome, 20 were characterized as plasmid contact hotspots (*p*-value < 0.001, Methods). Using *D. discoideum* vegetative state RNA-seq datasets from (Wang et al., 2021), we computed the average transcription profile over windows containing these Ddp5 contact hotspots (Wang et al, 2021). In this species too, the regions showing preferential contacts with the natural plasmid display reduced transcription compared to the rest of the genome (Fig. 4C), reminiscent of the 2μ plasmid hotspots.

## Discussion

The origin of this plasmid is not well known. Several clues could point at bacteriophages. The specific Flp1 recombinase is a tyrosine recombinase like the Cre recombinase of bacteriophage P1. It has also been shown that bacteria can naturally transform *S. cerevisiae* yeast by conjugation (Heinemann and Sprague, 1989) making possible a transfer between different species of genetic material that evolved into the 2μ plasmid we know today.

In this work, we exploit various genomic datasets to confirm that the 2μ plasmid chromatization is very similar to its host, potentially contributing to its long co-existence within its host's nucleus (Lieberman, 2006). We investigated using chromosome conformation capture data the physical contacts between the 2μ plasmid and its host's chromosomes, revealing that the 2μ plasmid interacts to discrete loci along their arms. Most contact hotspots consist of long, poorly transcribed regions depleted of transcription machinery proteins or known DNA-binding complexes, and may be associated with nucleosome signals as they depend on the tail of histone H4.

Previous reports have pointed out that the STB region on the 2μ plasmid recruits several yeast factors necessary for its segregation into the two daughter cells, including Cse4 (Hajra et al, 2006), the microtubule-associated proteins Bim1 and Bik1, and the motor protein Kip1 (Prajapati et al, 2017). The disruption of microtubules using nocodazole leds to the depletion of these proteins at STB region and generates plasmid missegregation (Prajapati et al, 2017), suggesting that those factors are important for the proper repartition between mother and daughter cells. However, Hi-C contact maps in presence of nocodazole treatment show that the plasmid remains bound to host chromosomes, indicating that these factors are not needed to maintain attachment. Moreover, condensin (Kumar et al, 2021) and cohesin (Mehta et al, 2002) were also proposed to be necessary for the plasmid propagation, but our data show that chromosome binding is independent of both complexes. It cannot be ruled out that these proteins are useful to the 2μ plasmid but for other processes.

Observations obtained during heat shock experiments show that the kinetics of this system are of the order of a few minutes. These rapid kinetics are hardly compatible with processes for writing or erasing epigenetic marks. For example, it has been shown using optogenetic control of Set2 that the histone mark H3K36me3 has a writing and erasing time of around 30 min (Lerner et al, 2020). More generally, these experiments show that the attachment of the 2 μ plasmid is dynamic and non-specific to a DNA sequence, suggesting a high adaptability that can quickly adjust to the host biological state.

Our experiments point to a model where the 2μ plasmid recognises through the Rep1/Rep2 proteins complex a structural signal involving several nucleosomes in the least active regions of its host chromosomes. The nature of this signal involves the basic tail of histone H4 but remains to be characterised. Since the 2μ plasmid binds to relatively long inactive regions, a possibility is that chromatin geometry plays a role in the attachments. For instance, the 2μ plasmid system may recognise specific chromatin fibre patterns, such as the alpha or beta nucleosome motifs proposed in Hi-CO experiments (Ohno et al, 2019). Positioning along silent regions would also allow it to avoid interfering with the biological processes of its host, thus maintaining a certain neutrality that may account for its low fitness costs and its long cohabitation with *S. cerevisiae*.

The present results presented here are also molecular experimental evidence in favour of the hitchhiking model (Sau et al, 2019) which proposes that the 2μ plasmid physically attaches to the chromosomes of its host in order to take advantage of all the machinery and chromosome movements during segregation. The probability of contact between two different molecules (inter-chromosomal contacts) is very low from a thermodynamic point of view (Nicodemi and Prisco, 2009). The fact that we observe robust contact enrichment between the 2μ plasmid and the identified host positions is a strong indication that the plasmid must be physically attached to these positions.

Interestingly, we observed very similar behaviour in other natural plasmids present in the nuclei of other eukaryotes: notably in the yeasts *L. waltii* and *L. fermentati*, as well as in the amoeba *D. discoideum*. The contact hotspots between the Ddp5 plasmid and *D. discoideum* chromosomes display very similar characteristics compared to those we have shown in *S. cerevisiae* (i.e. long and non-transcribed regions). This suggests a similar plasmid - chromosome tethering mechanism in these two distant species. Whereas the CRISPR-cas9 system is based on a recognition mechanism that relies on a precise DNA sequence, the host-parasite system studied here seems to reveal a specificity mechanism based on a structural signal involving several nucleosomes. We envision that other biological processes depend on information encoded in nucleosomal availability and/or chromatin folding, notably chromosome attachment of certain DNA viral episomes like Epstein Barr virus (Kim et al, 2020) or Kaposi's sarcoma-associated herpesvirus.

## Methods

**Reagents and tools table**

| Reagent/ Resource | Reference or Source | Identifier or Catalog Number |
| --- | --- | --- |
| **Experimental models** | | |
| See Appendix Table S1 | | |
| **Recombinant DNA** | | |
| See Appendix Table S2 | | |

| Reagent/ Resource | Reference or Source | Identifier or Catalog Number |
|---|---|---|
| **Antibodies** | | |
| REP1 antibody | Melanie Dobson laboratory | Sengupta et al, 2001 https://doi.org/10.1128/jb.183.7.2306-2315.2001 |
| **Oligonucleotides and other sequence-based reagents** | | |
| PCR primer | This study | FG92 TTTCTCGGGCAATCTTCCTA |
| PCR primer | This study | FG24 GTATGCGCAATCCACATCGG |
| **Chemicals, Enzymes and other reagents** | | |
| RNase A | Fisher Life Science | Cat# 10753721 |
| Formaldehyde | Sigma-Aldrich | Cat# F8775 |
| Glycine | Sigma-Aldrich | Cat# G7126 |
| Mini-protean TGX stain-Free Gels | Bio-Rad | Cat# 2553068 |
| ARIMA-HiC | Arima Genomics | |
| Collibri | INVITROGEN | Cat# A38614096W |
| DTT | Sigma-Aldrich | Cat# D0632 |
| G dynabeads | INVITROGEN | Cat# 10765583 |
| Proteinase K | euroBio | Cat# GEXPRK01 |
| Precellys VK05 | OZYME | Cat# P000913 |
| microTUBE AFA Fiber 1 mL | Covaris | Cat# 520130 |
| microTUBE AFA Fiber 130 μL | Covaris | Cat# 520045 |
| Zymolyase | Amsbio | Cat# 120493 |
| Geneticin | Thermofisher | Cat# 11811031 |
| **Software** | | |
| HiCstuff 3.2.4 | www.github.com/koszullab/hicstuff | https://doi.org/10.5281/zenodo.8322591 |
| Chromosight 1.6.3 | www.github.com/koszullab/chromosight | https://doi.org/10.5281/zenodo.7150538 |
| Samtools 1.9 | http://www.htslib.org/download/ | https://doi.org/10.1093/bioinformatics/btp352 |
| Cooler 0.8.7–0.8.11 | https://cooler.readthedocs.io/en/latest/ | https://doi.org/10.1093/bioinformatics/btz540 |
| Bowtie2 2.3.4.1 | http://bowtie-bio.sourceforge.net/bowtie2/ | https://doi.org/10.1186/gb-2009-10-3-r25 |
| Bedtools 2.29.1 | https://github.co/arq5x/bedtools2 | https://doi.org/10.1093/bioinformatics/btq033 |
| deepTools 3.5.5 | https://github.com/deeptools/deepTools. | https://doi.org/10.1093/nar/gku365 |
| tinyMapper 0.9.1 | https://github.com/js2264/tinyMapper | https://doi.org/10.1093/bioinformatics/btae487 |
| Rideogram | https://github.com/TickingClock1992/RIdeogram | https://doi.org/10.7717/peerj-cs.251 |
| seqkit 2.1.0 | https://bioinf.shenwei.me/seqkit/ | |

| Reagent/ Resource | Reference or Source | Identifier or Catalog Number |
|---|---|---|
| dnaglider 0.0.5 | https://github.com/cmdoret/dnaglider | |
| pyBigWig 0.3.22 | https://github.com/deeptools/pyBigWig | |
| SRA Toolkit fastq-dump 2.11.3 | https://github.com/ncbi/sra-tools/wiki/01.-Downloading-SRA-Toolkit | |
| HTseq – count 1.99.2 | https://github.com/htseq/htseq | https://doi.org/10.1093/bioinformatics/btu638 |
| **Other** | | |
| NextSeq 500/550 v2.5 High Output Kit (75 Cycles) | Illumina | Cat# 20024906 |
| TruSeq DNA CD Indexes (96 Indexes) | Illumina | Cat# 20015949 |
| ScreenTape D5000 | AGILENT TECHNOLOGIES | Cat# 5067-5582 |
| Qiaquick PCR purification kit | Qiagen | Cat# 28104 |
| Covaris S220 | Covaris | Cat# 500217 |
| Agencourt AMPure XP beads | Beckman Coulter | Cat# A63881 |

## Strains and medium culture conditions

The genotype and background of strains used in this study are listed in the strain table Appendix Table S1. Plasmids are listed in Appendix Table S2.

The jhd2::KANMX4, set2::KANMX4, dot1::KANMX4, rsc1::KANMX4, rsc2::KANMX4, hst2::KANMX4 strains were made using PCR amplified regions of strain from EUROSCARF collection. The diploid strain containing Mycoplasma chromosomes were made by crossing a strain containing a CRISPR linearized version of Mycoplasma chromosomes with BY4742. Culture media Liquid YPD media (1% Yeast extract, 2% peptone, 2% Glucose), containing 200 μg.mL$^{-1}$ of Geneticin (Thermofisher CAT11811031) or not, and SD-HIS (0.17% Yeast Nitrogen Base, 0.5% Ammonium Sulfate, 0.2% artificial dropout lacking histidine and 2% Glucose) were prepared according to standard protocols. *D. discoideum* cells were cultured in 20 ml autoclaved SM medium (per L: 10 g glucose, 10 g proteose peptone, 1 g yeast extract, 1 g MgSO$_4$*7H$_2$O, 1.9 g KH$_2$PO$_4$, 0.6 g K$_2$HPO$_4$) with dead *Klebsellia pneumoniae* at 20 °C and 130 rpm. After 4 days of growth, cells were centrifuged at 300 rpm during 10 min before performing Hi-C procedure.

## Heat-shock experiment

Fresh YPD media was inoculated with overnight culture of cir+ BY4741 cells and grown at 25 °C. When the culture reached 10$^7$ cells.mL$^{-1}$, heat shock was applied by adding warm (65 °C)

fresh YPD media to shift media temperature from 25 °C to 37 °C. Cells were grown at 37 °C and cells were extracted at different time points for Hi-C.

## ChIP-seq procedure

ChIP was performed as described previously (Hu et al, 2015) without calibration strain. 15 OD$_{600}$ units of *S. cerevisiae* (~1.5 × 10$^8$ cells) were harvested from an exponentially growing culture. Fresh YPD media was added qsp 150 mL and fixation was performed by adding qsp 3% of formaldehyde (SIGMA, F8775) for 30 min at RT on mild agitation in a 500 mL erlenmeyer flask. Formaldehyde was quenched by the addition of Glycine (Fisher Scientific, 10061073) 300 mM for 20 min at RT under mild agitation. The cells were pelleted and stored at −80 °C until further processing. Cell lysis was performed in ChIP lysis solution (50 mM HEPES KOH pH 8, 140 mM NaCl, 1 mM EDTA, 1% Triton X100, 0.1% Na Deoxycholate, 1 mM PMSF, 1X protease inhibitor SIGMA, Complete Mini), cells were resuspended in 300 µL of ChIP lysis solution and then transferred in a VK05 2 mL tube (OZYME, P000913-LYSK0-A.0). The Tubes were shaken using a Precellys (OZYME) equipped with the cryolis module set at 4 °C at 7800 RPM (10 times the following sequence: 30 s ON, 20 s OFF). Cell lysate was then transferred to a 1 mL AFA Covaris tube (COVARIS, 520130). Sonication was performed on a Covaris S220 sonicator using the following parameters: peak power 240 W, duty factor 20%, cycle/burst 200 for 10 min. Cell lysate was then clarified by centrifugation (17,000 rcf, 10 min, 4 °C). Supernatant was recovered. 80 µL (Whole Cell Extract, WCE) was stored at −20 °C and will be used for the input; the rest was mixed with Rep1-antibody. The antibody has been gifted by Melanie Dobson, its production has been previously described (Sengupta et al, 2001). The tubes were put on a rotating wheel overnight at 4 °C. After being washed with ChIP lysis solution, 50 µL of protein G dynabeads (INVITROGEN, 10765583) were added to the tube and incubated at 4 °C for 2 h on a rotating wheel. Beads were then washed: twice in ChIP lysis solution, three times in ChIP lysis Higher salt solution (ChIP lysis solution but Nacl is at 500 mM), twice in ChIP washing solution (10 mM Tris-HCl pH 8, 250 mM LiCl, 0,5% NP-40, 0,5% NaDeoxycholate, 1 mM EDTA, 1 mM PMSF) and finally in TE 1X. Beads were resuspended in 120 µL of TES buffer (TE 1X, 1% SDS). 40 µL of TES3 (TE1X, 3% SDS) was added in the WCE tube. IP and WCE tubes were incubated at 65 °C overnight. 2 µL of 10 mg/mL RNAse-A was added to each tube and incubated at 37 °C for 1 h. 10 µL of Proteinase-K (euroBio, GEXPRK01-B5) was added to each tubes and incubated at 65 °C for 2 h. DNA was purified using Phenol-chloroform-isoamyl alcohol solution (pH 7, 25:24:1). Purified DNA was prepared for sequencing using Collibri™ ES DNA Library Prep (Invitrogen, A38605024). Paired-end DNA sequencing was performed on an Illumina NextSeq 500.

## FISH

FISH (Fluorescence In Situ Hybridization) experiments were performed as in (Gotta et al, 1996) with some modifications. The *M. mycoides* probe was obtained by direct labelling of the bacterial DNA (1.5 µg) using the Nick Translation kit from Jena Bioscience (Atto488 NT Labelling Kit). For the 2 µ plasmid labelling, a 5-kb PCR fragment was amplified from the 2 µ plasmid DNA using primer pair FG92 (TTTCTCGGGCAATCTTCCTA)/FG24 (GTAT

GCGCAATCCACATCGG). This PCR product (1.5 µg) was then labelled using the Nick Translation kit from Jena Bioscience (AF555 NT Labelling Kit). For the *M. mycoides* probe and the 2 µ plasmid probe, the labelling reaction was performed at 15 °C for 90 min and 30 min, respectively. The labelled DNA was purified using the Qiaquick PCR purification kit from Qiagen, eluted in 30 µl of water. The purified probe was then diluted in the probe mix buffer (50% formamide, 10% dextran sulfate, 2× SSC final). 20 OD of cells (1 OD corresponding to 10$^7$ cells) were grown to mid-logarithmic phase (1–2 × 10$^7$ cells/ml) and harvested at 1200 × *g* for 5 min at RT. Cells were fixed in 20 ml of 4% paraformaldehyde for 20 min at RT, washed twice in water, and resuspended in 2 ml of 0.1 M EDTA-KOH pH 8.0, 10 mM DTT for 10 min at 30 °C with gentle agitation. Cells were then collected at 800 × *g*, and the pellet was carefully resuspended in 2 ml YPD—1.2 M sorbitol. Next, cells were spheroplasted at 30 °C for 10 min with Zymolyase (60 µg/ml Zymolyase-100T to 1 ml YPD-sorbitol cell suspension). Sphero-plasting was stopped by the addition of 40 ml YPD—1.2 M sorbitol. Cells were washed twice in YPD—1.2 M sorbitol, and the pellet was resuspended in 1 ml YPD. Cells were put on diagnostic microscope slides and superficially air dried for 2 min. The slides were plunged in methanol at −20 °C for 6 min, transferred to acetone at −20 °C for 30 s, and air dried for 3 min. After an overnight incubation at RT in 4× SSC, 0.1% Tween, and 20 µg/ml RNase, the slides were washed in H$_2$O and dehydrated in ethanol 70%, 80%, 90%, and 100% consecutively at −20 °C for 1 min in each bath. Slides were air dried, and a solution of 2× SSC and 70% formamide was added for 5 min at 72 °C. After a second step of dehydration, the denatured probes were added to the slides for 10 min at 72 °C followed by a 37 °C incubation for 24 h in a humid chamber. The slides were then washed twice in 0.05× SSC at 40 °C for 5 min and incubated twice in BT buffer (0.15 M NaHCO$_3$, 0.1% Tween, 0.05% BSA) for 30 min at 37 °C. For the DAPI staining, the slides were incubated in a DAPI solution (1 µg/ml in 1× PBS) for 5 min and then washed twice in 1× PBS without DAPI.

## Microscopy and image analysis

For all fluorescent images, the axial (z) step is 200 nm and images shown are a maximum intensity projection of z-stack images. Images were acquired on a wide-field microscopy system based on an inverted microscope (TE2000; Nikon) equipped with a 100/1.4 NA immersion objective, a C-mos camera and a Spectra X light engine lamp (Lumencor, Inc) for illumination. The microscope is driven by the MetaMorph software (Molecular Devices). Images were not processed after acquisition. Images shown are maximum intensity projection of Z-stack acquisition.

## Hi-C procedure and sequencing

Cell fixation was performed with 3% Formaldehyde (Sigma-Aldrich cat no. F8775) and performed as described previously (Dauban et al, 2020). Quenching of formaldehyde was done by adding 300 mM of glycine at room temperature for 20 min. All the Hi-C were done using Arima Hi-C kit (Arima Genomics; restriction enzymes: DpnII, HinfI). Sequencing preparation was done using a Collibri ES DNA Library Prep kit for Illumina Systems (A38606024) and then sequenced on Illumina NextSeq500.

## Stability measurement of plasmid

Stability measurements were carried out as described in (McQuaid et al, 2019b). Yeast cells were cultured overnight (O/N) at 30 °C in YPD (Yeast Extract Peptone Dextrose) medium supplemented with G418 to select for plasmid-containing cells. The medium was prepared in a single batch to ensure consistency across all cultures.

Each strain was grown in 5 replicates for Fig. 3H and 6 replicates for Fig. 3I. For Fig. 3H, calculation of the proportion of plasmid-containing cells after O/N culture was carried out as follows: cells were spread on Petri dishes without selection medium and then replicated using a velour on dishes with selection medium. Statistical tests to compare the proportion of cells with the plasmid are the chi-square test directly on the counts of colonies grown on medium with and without selection (using the code stability_chi-test.py). For Fig. 3I, after the O/N incubation, a 1 mL aliquot from each culture was collected to determine the average plasmid copy number per cell and subsequent shotgun sequencing. For the kinetic measurements, the overnight cultures were diluted 1:1000 into fresh YPD medium. The diluted cultures were incubated at 30 °C, and samples were collected at three time points: 6, 12, and 24 h post-dilution. At each time point, aliquots were taken for plating to determine colony numbers in selective and non-selective media. Plates were incubated at 30 °C for 2 days. For DNA extraction and sequencing (Fig. EV4G), genomic DNA was extracted from the collected cells using a protocol involving zymolyase treatment to degrade the cell wall, followed by Proteinase K digestion to remove proteins. DNA was further purified using phenol extraction. The purified DNA was then prepared for shotgun sequencing using the Collibri Library Preparation Kit, according to the manufacturer's instructions. The average number of plasmids per cell (Fig. EV4G) is calculated as follows: (Proportion of read from plasmid/Expected proportion) divided by the proportion of cells with plasmids. Proportion of read from plasmid = number of reads from plasmid/total number of reads from shotgun sequencing, Expected proportion = size of the plasmid/size of the total genome (we took: 6 kb/12 Mb = 0.0005) and proportion of cells with plasmids computed from the values. The test used for comparison of the average number of plasmids per cell is the T test with default parameters (scipy.stats.ttest_ind function). All statistical tests and visualisation plots were performed with python scripts available on github.

## Data analysis

### Contact data processing (Hi-C, Micro-C)

Hi-C and MicroC processing was performed using hicstuff package (Matthey-Doret et al, 2022). Briefly, the paired-end reads were aligned to the S288C reference genome (GCA-000146045.2, R64-1-1) and the 2μ plasmid sequence (GenBank accession number: CM007980) as well as with *M. mycoides* (GCA-006265075). For the two experiments containing the lacO site array, reads were aligned to genomes based on strain W303 containing the lacO site sequences (for details on the constructions, see (Guérin et al, 2019)). A threshold of 1 for mapping quality was used for these 2 experiments. Genome for *L. waltii* was CBS6430 and X56553.1 sequence was used for pKW1 plasmid. Genome for *L. fermentati* was CBS 6772 and M18275.1 sequence was used for the plasmid pSM1. For *D. discoideum*, AX4 reference genome sequence GCA_000004695.1 dicty_2.7 was used and NC_001889.1

sequence was used for Ddp5 plasmid. We used bowtie2 in its very sensitive local mode. Unique mapped paired reads are then assigned to restriction fragments and non-informative contacts are filtered (Cournac et al, 2012; Matthey-Doret et al, 2022). PCR/optical duplicates are discarded (i.e. paired reads mapping at the same genomic positions). Contact signals were binned at 2 kb resolution except where noted (e.g., for the 2μ plasmid contact map or the averaged contact signal at long genes, a resolution of 200 bp was used).

### Computation of contact signal of 2μ plasmid

To compute the contact signal of the 2μ plasmid with each bin of the host genome, we used the normalised following score:

$$S_i = \frac{cp_{ij}}{\sum_{j=1}^{N_{genome}} c_{ij}} \Big/ \frac{n_{plasmid}}{N_{total}}$$

$S_i$ represents the contact score between 2μ plasmid and a bin $i$ in the host genome. It corresponds to the proportion of contacts made with the 2μ plasmid for a bin $i$ of the host genome normalised by the percentage of presence of 2μ plasmid in the library. $cp_{ij}$ is the number of contacts detected between 2μ plasmid and host bin $i$. $\sum_{j=1}^{N_{Genome}} c_{ij}$ is the total number of contacts detected for the host bin $i$. $N_{genome}$ is the total number of genomic bins (in general 2 kb bins). $n_{plasmid}$ is the number of reads involving the plasmid. $N_{total}$ is the total number of reads of the library. For ideogram visualization, the contact signal Si is represented using R-ideogram package (Hao et al, 2020).

### ChIP-Seq, Mnase-seq, ATAC-seq and RNA-seq processing

ChIP-Seq, Mnase-seq, ATAC-seq and RNA-seq processing was performed using TinyMapper (https://github.com/js2264/tinyMapper). Paired end reads were aligned using the very sensitive and local mode of bowtie2 (Langmead and Salzberg, 2012), against S288C reference genome and the 2μ plasmid (CM007980) sequences. Only concordant pairs were retained and reads with a mapping quality larger than 0 were kept. PCR duplicates were removed using samtools. When available, the input was similarly processed. Coverage for each genomic position was computed using deeptools bamcoverage function, then normalised by Count Per Million (CPM) method. ChIP signal was then computed by dividing immuno-precipitated normalised coverage by input coverage. Signals were then visualised with homemade python using pyBigWig package. RNA-seq signal was log represented.

To quantify the level of gene transcription of 2μ plasmid taking into account the DNA content per cell of 2μ plasmid, we computed, for two independent experiments, the number of reads overlapping each gene in the RNA Pol II ChIP-seq signal, divided by the number of reads overlapping each gene in the input signal (using htseq-count with default parameters (Anders et al, 2015). We computed the distribution of levels of transcription for the genes of *S. cerevisiae* using home python code.

### Automatic detection of peaks of contact with 2μ plasmid

To detect contact peaks with the 2μ plasmid, we used the *find_peaks* function of the scipy package (with the following parameters: height = 0.8, distance = 2). Before detection, the contact signal was interpolated using the *interp1d* function of the scipy package. In most of the average 2μ plasmid contact profiles shown, the set of contact peaks used is the one detected under normal log-phase culture conditions from Micro-C data (Swygert et al, 2019)

corresponding to the 73 genomic positions given in Appendix Table S6.

To determine significant enrichment of contact peaks of the 2μ plasmid with a given type of genomic region, e.g. long gene or telomeric regions. We used the code overlap_with-group.py, which generates 1000 realisations of genomic positions of the same size as the contacts detected. The proportion of peaks that overlap a genomic position group is calculated for the 2μ plasmid contacts and for each of the 1000 random realisations. For subtelomeric regions, we have taken 30 kb at the beginning and end of each chromosome. The p-value is given by the number of random configurations with a number greater than or equal to the number detected in the experimental data divided by the total number of realisations. The same strategy was used to assess significance for the group of long genes in the different species.

### Averaged contact profile of the 2μ plasmid around loci

To plot the average contact profile around loci of interest, we extracted the contact signals at windows $+/-40$ kb centred at positions of interest using a 2 kb binned signal. In cases where the limits of the window exceed a chromosome, NaN values were used. The confidence interval is represented around the mean value.

### Genomic features of regions contacted by the 2μ plasmid

To plot the aggregated profile of genomic signals, we convert contact data (cool file at 200 bp resolution) into a bw file using homemade python code. We then used the functions *computeMatrix* and *plotProfile* from deepTools suite (Ramírez et al, 2016) to compute and plot the heatmaps. The same approach was used to plot the averaged contact signal inside long genes. Gene boxes at regions contacted by the 2μ plasmid were generated using homemade python code and using the coordinates of genes of SGD database (http://sgd-archive.yeastgenome.org/sequence/S288C_reference/orf_dna/). To plot the pileup plots around the pairs of genomic positions of peaks of contact with the 2 μ plasmid, we used the *quantify* mode of Chromosight (Matthey-Doret et al, 2020) with the following parameters --perc-undetected = 100 --perc-zero = 100. All possible pairs of peaks of contact in intra or inter configurations were generated using homemade python code and the function *combinations* from itertools package.

The computational screen uses a dataset of Chip-exo libraries for about 800 different proteins and genomic signals (Rossi et al, 2021). For each aligned library (bowtie2 alignment with very sensitive mode and mapping quality >0), Chip-exo signal was computed with homemade python code with a binning of 2 kb. An enrichment score was computed by taking the average of the ChIP-exo signal $+/-2$ kb signal around the bins of the positions of contact peaks detected in log phase (Appendix Table S6). The different libraries were sorted according to the different categories identified in the UMAP analysis (Rossi et al, 2021).

## Data availability

The accession number for the sequencing reads reported in this study are accessible on GEO with the accession number GSE246637. All other data supporting the findings of this study are available from the corresponding author on reasonable request. Source data are provided with this paper. Data associated with this study are publicly available, and their reference numbers are listed in Appendix Tables S4, S5. The custom-made code of the analysis is available online at https://github.com/axelcournac/2micron_project. Open-access versions of the programmes and pipeline used (Hicstuff) are available online on the github account of the Koszul lab Hicstuff (www.github.com/koszullab/hicstuff) version 3.0.1, Bowtie2 (version 2.3.4.1 available online at http://bowtie-bio.sourceforge.net/bowtie2/), SAMtools (version 1.9 available online at http://www.htslib.org/download/http://www.htslib.org/download/), Bedtools86 (version 2.29.1 available online at https://bedtools.readthedocs.io/en/latest/content/installation.html) and Cooler (versions 0.8.7–0.8.11 available online at https://cooler.readthedocs.io/en/latest/).

The source data of this paper are collected in the following database record: biostudies:S-SCDT-10_1038-S44318-025-00389-1.

## Peer review information

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

## Acknowledgements

The authors are extremely grateful to M. Dobson for sharing the 2μ plasmid mutant strains, for the gift of the Rep1 antibody, and constructive feedback on this manuscript. We thank G. Liti and J. Schacherer for sharing strains and suggestions. We thank S. G. Swygert and T. Tsukiyama for sharing strains, Y. Barral, L. Baudry, G. Millot, P. Moreau, A. Piazza, C. Chapard, M. Delouis, C. Matthey-Doret, J. Sérizay for fruitful discussions and advice. Biomics Platform, C2RT, Institut Pasteur, Paris, France, is supported by France Génomique (ANR-10-INBS-09) and IBISA. This work used the computational and storage services (maestro cluster) provided by the IT department at Institut Pasteur, Paris. FG is supported by an ENS Paris Saclay fellowship. This work has received support under the programme Investissements d'Avenir launched by the French Government and implemented by ANR with the references ANR-10-LABX-54 MEMOLIFE and ANR-10-IDEX-0001-02 PSL Université Paris, Q-life ANR-17-CONV-6150005 (SA). The authors greatly acknowledge the PICT-IBiSa@Pasteur imaging facility of the Institut Curie, member of the France Bioimaging National Infrastructure (ANR-10-INBS-04). This research was funded, in whole or in part, by the European Research Council under the Horizon 2020 Program (ERC grant agreement 771813) to RK, the Agence Nationale pour la Recherche ANR-22-CE12-0013-01 to RK and AT and the Agence Nationale de Recherche sur le Sida et les hépatites virales grant agreement 24461 to RK and ANR-19-CE45-0003-01 to AC.

## Author contributions

**Fabien Girard**: Conceptualization; Data curation; Formal analysis; Investigation; Methodology; Writing—original draft. **Antoine Even**: Investigation; Methodology. **Agnes Thierry**: Investigation; Methodology. **Myriam Ruault**: Investigation; Methodology. **Léa Meneu**: Investigation; Methodology. **Pauline Larrous**: Investigation; Methodology. **Mickaël Garnier**: Investigation; Methodology.

**Sandrine Adiba**: Investigation; Methodology. **Angela Taddei**: Investigation; Writing—review and editing. **Romain Koszul**: Conceptualization; Supervision; Funding acquisition; Investigation; Writing—original draft; Writing—review and editing. **Axel Cournac**: Conceptualization; Data curation; Formal analysis; Supervision; Funding acquisition; Investigation; Methodology; Writing—original draft; Writing—review and editing.

Source data underlying figure panels in this paper may have individual authorship assigned. Where available, figure panel/source data authorship is listed in the following database record: biostudies:S-SCDT-10_1038-S44318-025-00389-1.

## Disclosure and competing interests statement

The authors declare no competing interests.

# Expanded View Figures

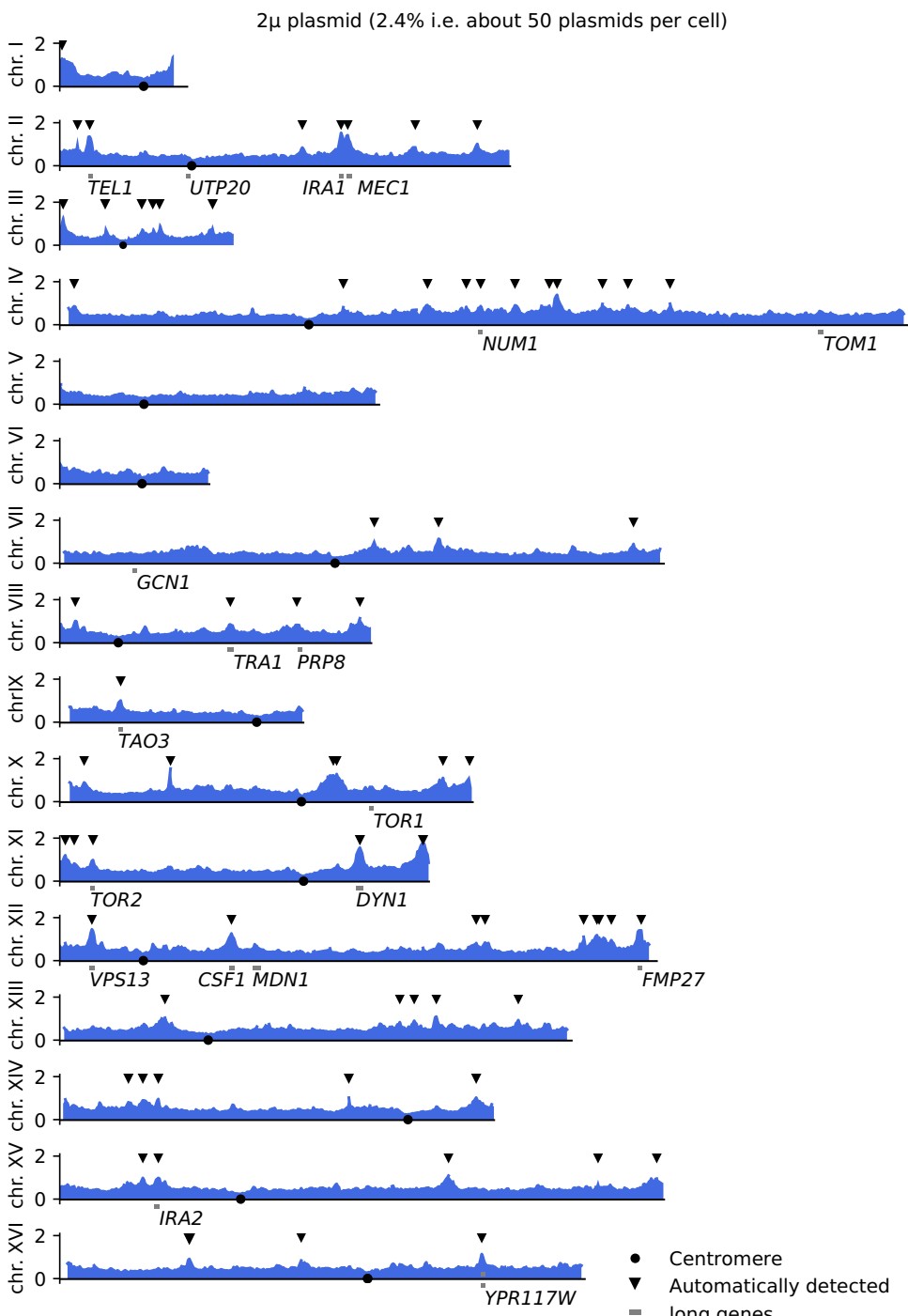

**Figure EV1.  Contact signal of the 2μ plasmid along the 16 chromosomes of *S. cerevisiae*.**

The contact signal is binned at 2 kb, genes with size >7 kb are annotated with grey rectangles and their names (MicroC data from Swygert et al (2019)). Automatically detected peaks of contact (73 genomic positions) were annotated with black triangles.

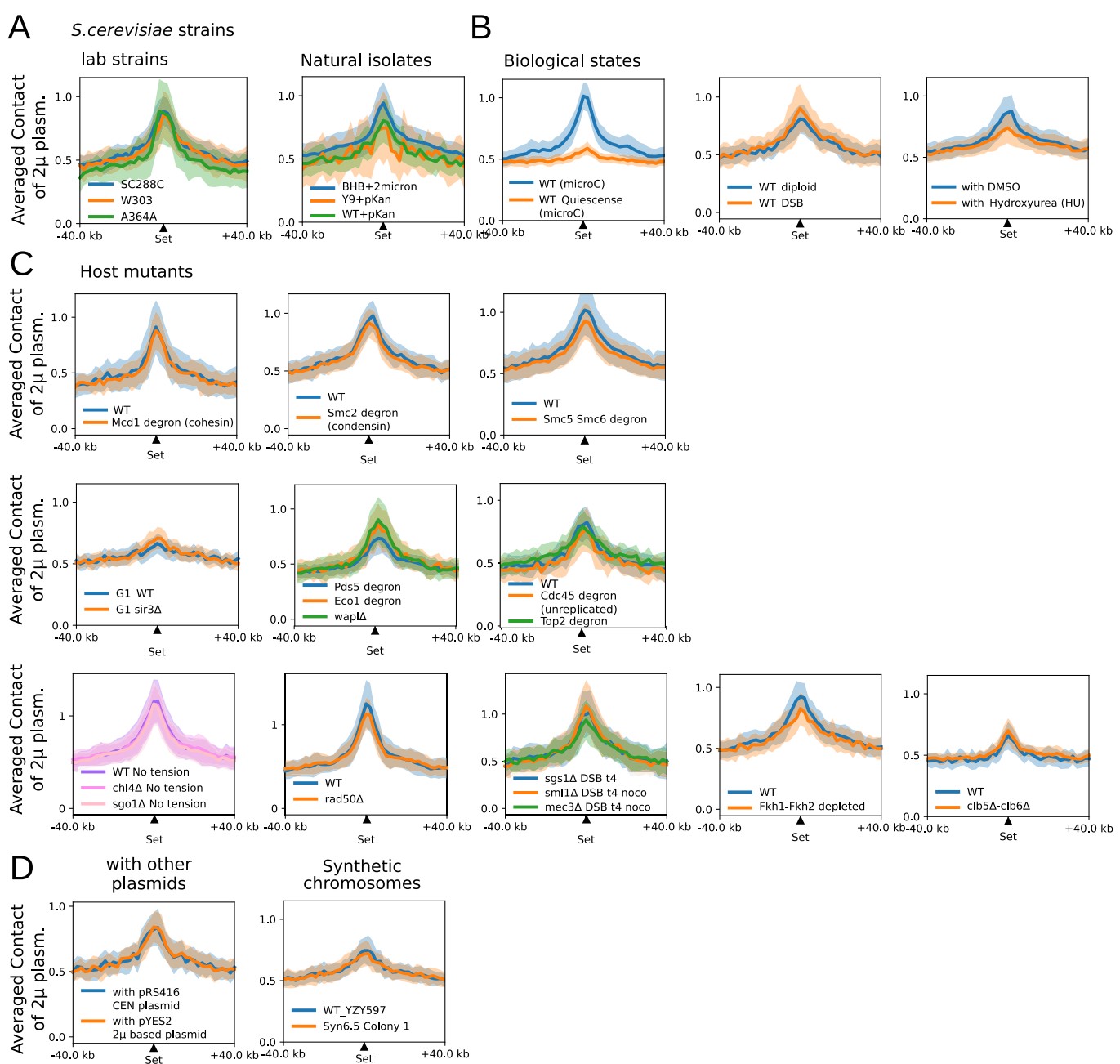

**Figure EV2. The specific positioning of the 2μ plasmid is conserved under a wide variety of biological conditions and mutants.**

(A) Averaged 2μ plasmid contact signal over the hotspots of contact identified in WT, log phase in different lab strains of *S. cerevisiae*: SC288C, W303 (Dauban et al 2020), A364A (Costantino et al, 2020) and strains from natural isolates (Peter et al, 2018). pKan version of the plasmid was used for the Y9 strain to ensure stability as well the corresponding control. pKan contains the KAN resistance gene and *FLP1* gene is disrupted. (B) Averaged 2μ plasmid contact signal over the hotspots of contact identified in WT, log phase in different biological states: in quiescence (Swygert et al, 2019) (same as Fig. 1C), in diploid stage, with double-strand break (DSB) of DNA (Piazza et al, 2021), with DMSO or HU treatment (Jeppsson et al, 2022). (C) Averaged 2μ plasmid contact signal over the hotspots of contact identified in WT, log phase in different mutants of *S. cerevisiae*: Mcd1 degron mutant (subunit of cohesin, AID system) (Costantino et al, 2020), Smc2 degron mutant (subunit of condensin, AID system) (Guérin et al, 2019), Smc5-Smc6 degron mutant (AID system) (Jeppsson et al, 2024). sir3Δ (Ruault et al, 2021), Pds5, Eco1 degron mutants (AID system), wpl1Δ mutant, Cdc45 degron mutant (stopped replication, AID system) (Dauban et al, 2020), Top2 degron mutant (topoisomerase II, AID system) (Lazar-Stefanita et al, 2017), in condition with no tension of microtubules i.e. with nocodazole treatment (noco), chl4Δ, sgo1Δ (Paldi et al, 2020), rad50Δ (Forey et al, 2021). sgs1Δ, sml1Δ, mec3Δ mutants at 4 h after induction of HO endonuclease-mediated site-specific DSB (Piazza et al, 2021), in Fkh1-Fkh2 depleted mutant (GAL1pr-FKH1 fkh2Δ mutant) (Eser et al, 2017), in clb5Δ-clb6Δ mutant (Barton et al, 2022). (D) Averaged 2μ plasmid contact signal over the hotspots of contact identified in WT, log phase in presence of other plasmids (centromeric and 2μ based) and with synthetic chromosomes (Zhao et al, 2021).

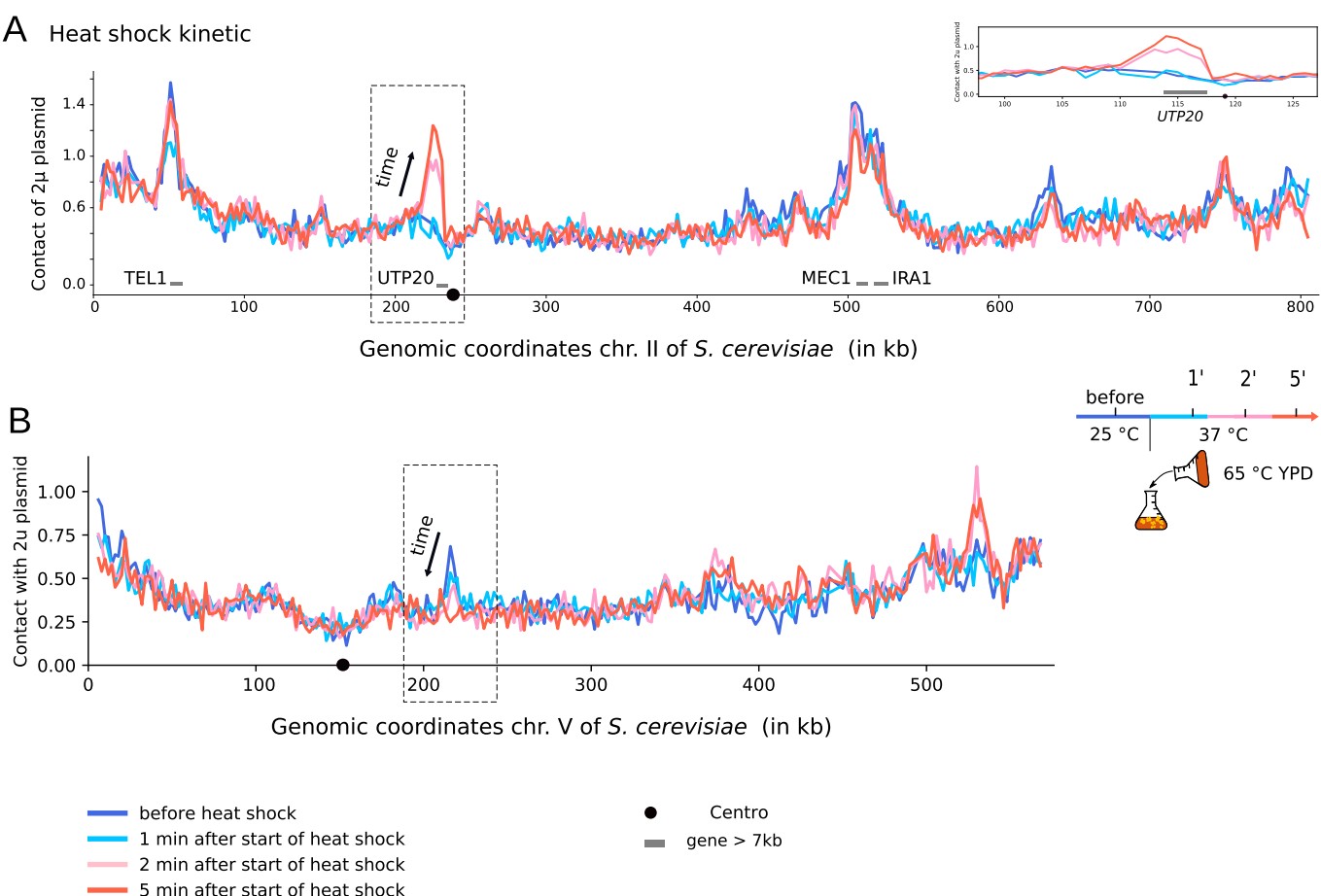

**Figure EV3. Contact signal of 2 μ plasmid during a heat shock.**

(A) Contact signal of 2 μ plasmid along the chromosome II of *S. cerevisiae* for 4 time points: before heat shock, 1 min, 2 min and 5 min after heat shock. (B) Contact signal of 2 μ plasmid along the chromosome V of *S. cerevisiae* for 4 time points: before heat shock, 1 min, 2 min and 5 min after heat shock. Binning for contact signals is 2 kb.

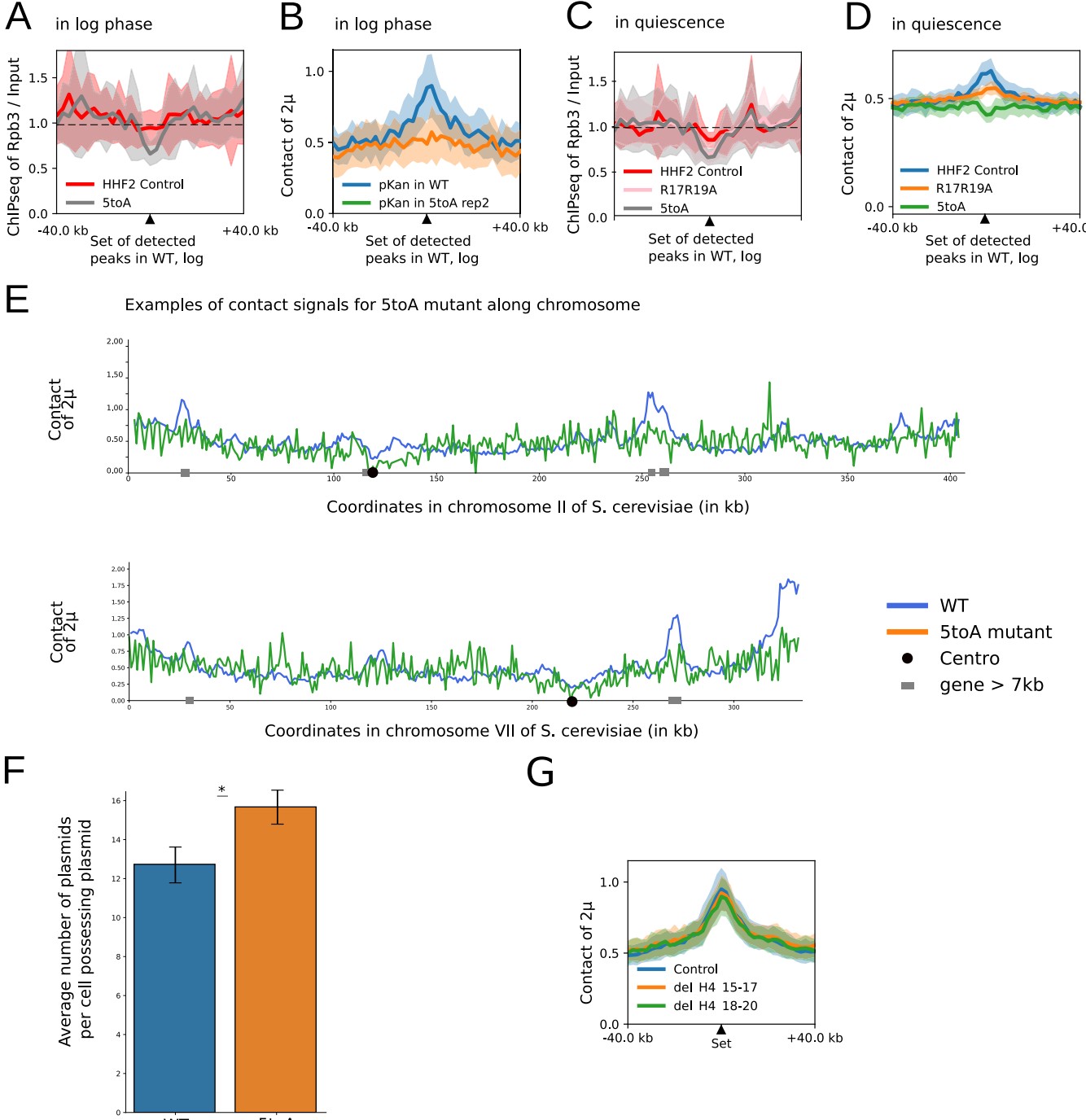

**Figure EV4. Contact signal of 2μ plasmid in the H4 5toA mutant.**

(A) Averaged transcription signal measured by Rpb3 (Pol II subunit) ChIP-seq (Swygert et al, 2021) over the set of identified loci contacted by 2μ plasmid in WT, log phase condition for the HHF2 control and H4 5toA mutant in log phase. In the HHF2 control, the endogenous H3 and H4 loci were deleted and complemented by a wild-type copy of H3 and H4 genes at an ectopic locus (TRP1). (B) Averaged 2μ plasmid contact signal over the set of identified loci contacted by 2μ plasmid in WT, log phase condition for the WT and H4 5toA mutant in log phase, replicate 2. pKan version of the 2μ plasmid was used to ensure plasmid stability (which contains the *KAN* resistance gene and whose *FLP1* gene is inactivated). (C) Averaged transcription signal measured by Rpb3 (Pol II sub-unit) ChIP-seq (Swygert et al, 2021) over the set of identified loci contacted by 2μ plasmid in WT, log phase condition for the HHF2 control, H4 R17R19A and H4 5toA mutants in quiescence phase (Swygert et al, 2021). (D) Averaged 2μ plasmid contact signal over the set of identified loci contacted by 2μ plasmid in WT, log phase condition for the HHF2 control, H4 R17R19A and H4 5toA mutants in quiescence phase (Swygert et al, 2021). (E) Examples of contact signals of 2μ plasmid (pKan version) in WT and H4 5toA mutant. (F) Average number of plasmids per cell computed taking into the proportion of cells having plasmids and reads proportion from plasmid sequence with shotgun sequencing. Results represent the average (± s.d.) from assaying 3 transformants for each condition, respectively, and correspond to overnight culture shown in Fig. 3I ($p$-value = 0.0308). (G), Averaged 2μ plasmid contact signal over the set of identified loci contacted by 2μ plasmid in WT, log phase condition for the control, H4 mutant where the amid acids 15 to 17 have been deleted (del H4 15–17) and H4 mutant where the amid acids 18 to 20 have been deleted (del H4 18–20). Source data are available online for this figure.

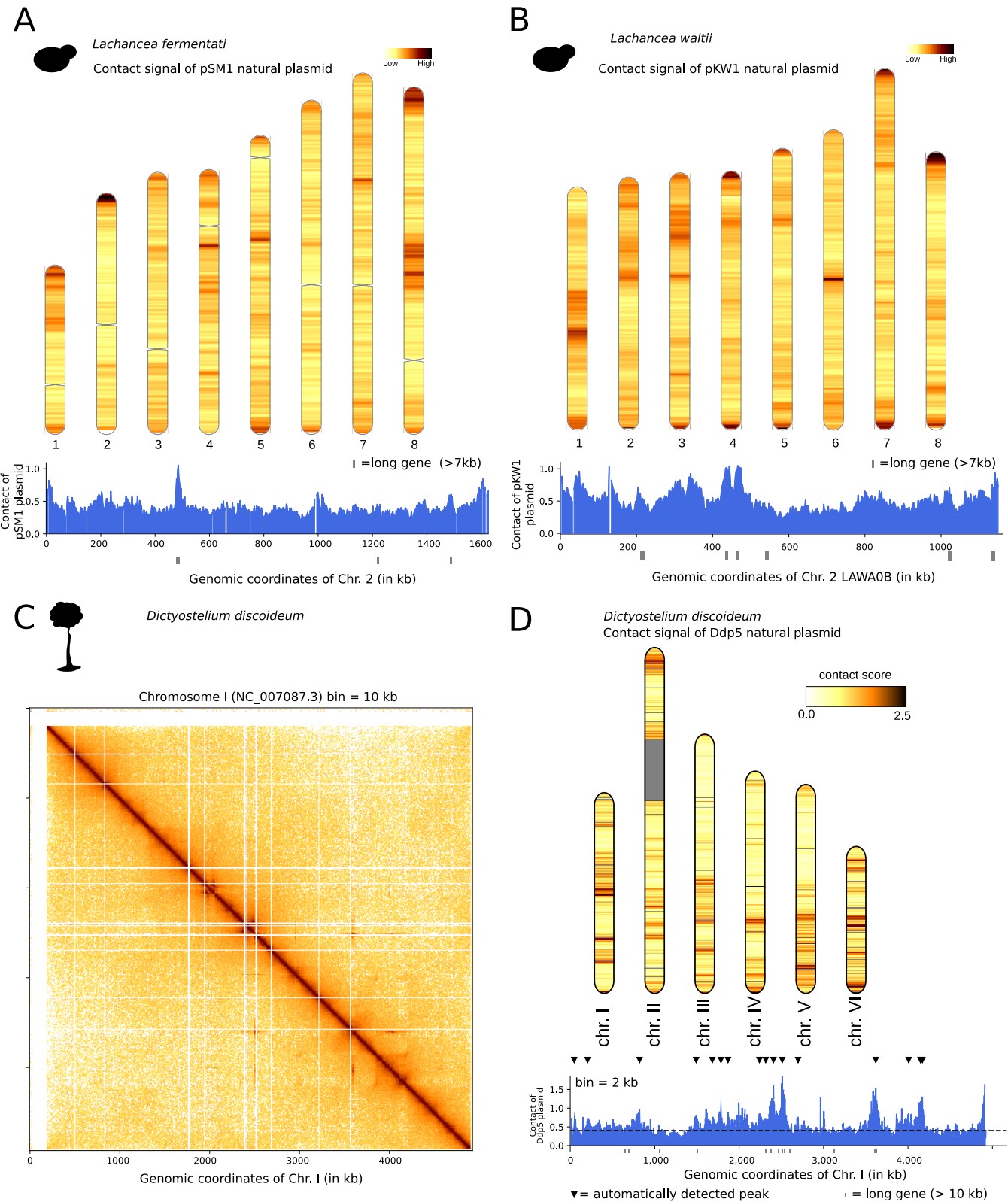

◄ **Figure EV5. Contact signals of natural plasmids in other eukaryotes.**

(A) Contact signal of the pSM1 natural plasmid with the chromosomes of *Lachancea fermentati* yeast. Contact profile for the chromosome 2, long genes are annotated as grey boxes. (B) Contact signal of the pKW1 natural plasmid with the chromosomes of *Lachancea waltii* yeast. Contact profile for the chromosome 2, long genes are annotated as grey boxes. (C) Contact map of chromosome I of *Dictyostelium discoideum* at 10 kb resolution. (D) Contact signal of the Ddp5 natural plasmid with the chromosomes of *Dictyostelium discoideum* and contact profile for chromosome 1. Black triangles indicate automatically detected peaks, and long genes are annotated as grey boxes.

