## [Peer Review File · The EMBO Journal]

Plasmids are anchored to inactive regions of eukaryotic chromosomes through a nucleosome-encoded signal

Fabien Girard, Antoine Even, Agnes Thierry, Myriam Ruault, Léa Meneu, Pauline Larrous, Mickaël Garnier, Sandrine Adiba, Angela Taddei, Romain Koszul, and Axel Cournac

Corresponding author(s): Axel Cournac (axel.cournac@pasteur.fr) , Romain Koszul (romain.koszul@pasteur.fr)

Review Timeline:

Submission Date:	2nd Feb 24
Editorial Decision:	7th May 24
Revision Received:	21st Oct 24
Editorial Decision:	10th Jan 25
Revision Received:	21st Jan 25
Accepted:	31st Jan 25

Editor: *Cornelius Schneider*

Transaction Report:

Dear Dr Courmac,

Thank you again for submitting your manuscript to the EMBO Journal and for sharing your preliminary point-by-point reply. We find your arguments convincing and the proposed experiments suitable and would therefore like to invite you to revise your manuscript based on your preliminary point-by-point reply.

I should also add that it is The EMBO Journal policy to allow only a single major round of revision and that it is therefore important to resolve the main concerns at this stage.

We generally allow three months as standard revision time, which can be extended to 6 months in case of major revisions, such as the experiments required here. As a matter of policy, competing manuscripts published during this period will not negatively impact on our assessment of the conceptual advance presented by your study. However, we request that you contact the editor as soon as possible upon publication of any related work, to discuss how to proceed. Should you foresee a problem in meeting the deadline, please let us know in advance and we may be able to grant an extension.

Thank you for the opportunity to consider your work for publication. I look forward to your revision.

Yours sincerely,

Cornelius Schneider

Cornelius Schneider, PhD
Editor
The EMBO Journal
c.schneider@embojournal.org

Please remember: Digital image enhancement is acceptable practice, as long as it accurately represents the original data and

conforms to community standards. If a figure has been subjected to significant electronic manipulation, this must be noted in the figure legend or in the 'Materials and Methods' section. The editors reserve the right to request original versions of figures and the original images that were used to assemble the figure.

We realize that it is difficult to revise to a specific deadline. In the interest of protecting the conceptual advance provided by the work, we recommend a revision within 3 months (5th Aug 2024). Please discuss the revision progress ahead of this time with the editor if you require more time to complete the revisions. Use the link below to submit your revision:

Referee #1:

This paper is concerned with the localization of the yeast 2μ plasmid with potential relevance for its maintenance mechanisms. These have been the subject of extensive study and analysis, since 2μ is an intriguing example of a eukaryotic genetic element in the form of a circular plasmid that appears to act as a relatively benign "molecular parasite". 2μ plasmid uses an otherwise unprecedented mechanism involving a site-specific recombinase (FLP) to regulate its copy number by inverting the direction of migration of replication forks (Futcher, 1986; *J. Theor. Biol.* 119, 197-204). A second gene (RAF) is involved in activating FLP transcriptional regulation through blocking the regulatory action of the final two genes REP1 and REP2. However the major function of Rep1 and Rep2 is acting on a cis-plasmid locus known as STB to confer plasmid stability through effective partitioning during mitotic and meiotic divisions. The work of multiple labs has demonstrated that the partitioning/ segregation mechanism of 2μ plasmid involves interaction of plasmid components with yeast chromosomes.

The current manuscript is very well written and clear, and provides a significant amount of data and analysis. Using proximity ligation techniques (Hi-C and Micro-C), the authors map contacts between 2μ plasmid and yeast chromosomes. The authors report that the plasmid links preferentially to specific regions of low transcriptional activity. The localization of plasmid DNA sequence is supported by REP1 ChIP analysis, suggesting REP1 protein is also localized to these regions as expected from its function. However this localization does not appear to depend on a number of other factors including the SMC complex previously reported as important for 2μ plasmid segregation. The authors conclude that 2μ contacts with these sequences depend on a nucleosomal signal associated with depletion of RNA polymerase II. The authors further show that similar patterns of association apply to the Dictyostelium plasmid and suggest it may be a more widespread mechanism.

The work represents a comprehensive application of proximity mapping techniques to the association of the yeast 2μ plasmid with yeast genomic sequences, and as such is a significant update on previous studies that lacked the resolution of Hi-C type techniques. Almost half a century ago, Livingston and Hahne (1979; <https://pubmed.ncbi.nlm.nih.gov/386345/>) and Nelson and Fangman (1979) (<https://pubmed.ncbi.nlm.nih.gov/392520/>) confirmed that the plasmid has a conventional nucleosome structure, and evidence for an association of 2μ with specific nuclear structures originated with work by Wu et al (1987; <https://pubmed.ncbi.nlm.nih.gov/3542994/>). Scott-Drew et al (2002; <https://pubmed.ncbi.nlm.nih.gov/12095225/>) used imaging approaches to conclude that the STB locus of 2μ (together with the plasmid proteins Rep1 and Rep2) localizes to discrete chromatin sites, and consistent with the current report showed that these sites are distinct from telomeres. Additional more recent studies are cited in the manuscript.

Functional analysis showed the involvement of cohesin in the maintenance mechanism of 2μ (<https://pubmed.ncbi.nlm.nih.gov/17670945/>; <https://pubmed.ncbi.nlm.nih.gov/15169893/>) as well as the involvement of the "remodels the structure of chromatin" (RSC) chromatin remodeling complex, the nuclear motor Kip1, and histone H3 variant Cse4 (as discussed in the current manuscript). However a full understanding of how these components contribute to the 2μ partitioning system remains elusive.

Interestingly, the current work shows no evidence of effect of these various factors on plasmid association with the identified genomic sequences, despite clear evidence that they are required for plasmid maintenance. This may suggest that the specific plasmid DNA localization documented is not necessarily required for effective plasmid segregation. In other words, some or all of the plasmid may be preferentially associated with particular sites, but evidence that this has a functional significance does not appear to be clear. It would be useful to know what proportion of the total plasmid is associated with the chromatin, or is some plasmid "free" in the nucleoplasm?

The manuscript makes reference to plasmid copy number, here determined by average read counts (e.g. page 6: " 2μ plasmid mutants lacking either REP1 or the STB loci ... These mutant plasmids are unstable (Kikuchi, 1983; McQuaid et al., 2019) and carry a marker gene to be retained in the cell. This instability is reflected by a relatively low copy number per cell, which can be assessed from the proportion of reads from 2μ in each library".) However this is not a useful measure of copy number. What is being measured is the average copy number relative to total extracted DNA, not average copy number per plasmid-containing

cell, which is the relevant measure. As analyzed in detail in the Kikuchi (1983) paper, unstable plasmids still replicate once per cell cycle; they are unstable due to a defective partition system and hence accumulate to high copy numbers in a small proportion of cells in the population. Hence it is necessary to measure both the proportion of plasmid-containing cells (using a selective plating assay) and the average copy number and hence calculate the plasmid copy number per plasmid containing cell. Without direct determination of the proportion of plasmid-containing cells, it is not possible to draw conclusions regarding plasmid segregation or the copy number of plasmids in cells that contain them.

This is also relevant on page 10, where it is stated that: "RSC2 was shown to be essential for the 2 μ plasmid maintenance (Wong et al., 2002). We observed indeed a significant drop in the number of plasmids per cell in this mutant (Extended Data Fig. 5). However, its attachment to host chromosomes remain unchanged". Again this statement is not correctly phrased- what is affected in the *rsc2* mutant is the segregation of the plasmid and it therefore accumulates in a subset of cells to a higher than normal copy number and the average copy number across all cells in the population falls due to the large number of cells that have lost the plasmid and still continue to divide for a while. Nevertheless the conclusion that Rsc2p is not needed for normal plasmid association with identified chromatin regions is supported by the data.

The identified sites of 2 μ association are shown here to be associated with low transcriptional activity, particularly containing long genes. In these regions there is enrichment of H3K36me3, and in the *Set2* mutant (*Set2p* methylates H3K36me3) the level of transcription of long genes is slightly increased and contact specificity for the 2 μ plasmid is substantially reduced (but still detectable). Does this indicate causative association of 2 μ with low transcribed regions/ long genes or correlation with wider changes induced by loss of H3K36me3?

In considering the factor(s) by which the 2 μ associates with these regions identified as depleted in *PoIII* activity, the authors identify the tail of histone H4 as required to confer localization based on a five-alanine substitution variant. A key question this raises is whether plasmid segregation is affected in this mutant H4 mutant, which would be predicted if the localization of plasmid contributes to its partition mechanism. Identifying the nature and proteins involved with the H4 tail interaction could also provide more functional insight.

Overall the manuscript provides compelling data for the preferential localization of 2 μ plasmid with specific regions of chromatin with higher levels of transcriptional inactivity. Given the relocation to alternative regions with low transcriptional activity in *set2* mutants, and failure to associate in histone H4 tail mutants, does this reflect a general association preference of 2 μ with less transcriptionally active regions but without particular functional significance? Testing the segregation of plasmids in these mutant backgrounds could provide supportive evidence and further our understanding of 2 μ partition mechanism.

Minor points:

"long genes have their contact signal with the 2 μ plasmid greatly reduced while the level of transcription is slightly increased (Supplementary Fig. 13c)." should read "Supplementary Fig. 14c".

Methods do not always provide sufficient detail to allow experimental repeat- eg REP1 ChIP experiment.

Referee #2:

The manuscript entitled "Anchoring of parasitic plasmids to inactive regions of eukaryotic chromosomes through nucleosome signals" describes the interactions that parasitic plasmids establish with host genomes, using available and newly generated Hi-C and Micro-C data. These studies focus primarily on the 2-micron of *Saccharomyces cerevisiae* and demonstrate that it associates with long, lowly expressed genes. The contacts of the plasmid with these target genes depend on the plasmid-encoded protein Rep1 and its binding site, the STB sequence, on the plasmid. Since Rep1 and STB are required for the proper maintenance of the plasmid in the population, these data suggest that association of the plasmid with its target loci on the chromosome contribute to its mitotic stability. The interaction of the plasmid with the target chromosomal loci depended on the basic patch on the tail of histone H4. Analysis of the interaction of other parasitic plasmids with the chromosomes of their hosts, such as plasmids related to the 2-micron in other fungi and the Ddp5 plasmid in *Dictyostelium discoideum*, indicate that they also contact long, poorly expressed genes. Together, these data support the idea that these parasitic plasmids propagate between sister-chromatids by hitchhiking on chromosomes and indicate that for that they use low expressed genes as binding sites. This may help keep their fitness-cost very low for the host.

Together, this is a very thorough and interesting study that addresses convincingly and fairly conclusively for the first time a long-standing question. The data are likely to be relevant for a broad variety of parasitic genomes, including viruses that remain episomal during their latency phase and hitchhike on chromosomes during cell division, such as the Epstein-Barr virus (EBV) and the Hepatitis B virus (HBV). The data are convincing, well presented and well documented. However, a few simple tests would significantly enhance the relevance of the work and some statements probably need revision. Provided that these points are addressed, this reviewer fully support the publication of this important and beautiful piece of work.

Main points:

1- The authors convincingly show that mutating the basic patch of H4 tail strongly impair the interaction of the 2-micron with its target loci. Furthermore, the authors conclude from the fact that there are less reads for the 2-micron in the H4-5toA mutant cells that it must be less stable in these cells. However, other mechanisms could also reduce the number of copies without affecting mitotic propagation. The authors need to characterize the mitotic stability of the 2-micron plasmid in the H4 mutant cells to demonstrate that it is less stable indeed. This is an essential test for supporting their conclusions. It is also a simple test. The easiest assay is to measure the number of cells that have lost the plasmid after growth on non-selective medium, taking wild type as positive control and the mutant rep1- or stb- plasmids as negative controls. If the stability is not down to the mutant level, this would indicate that hitchhiking is not the only parameter ensuring the mitotic stability of the plasmid. If it is even still as stable as in the wild type, this would indicate that association with chromosomes is not relevant for partition but for some other process.

2- The authors show that removing the Rsc2 protein does not impair interaction of the 2-micron with its target genes, although it strongly affects its maintenance. Discussion is needed.

3- The authors state at the end of their discussion that they envision that eccDNAs, excised from the genome could also rely on the same principle (implicating, for their segregation). However, such eccDNAs do not encode either of Rep1 and Rep2 and do not contain an STB site. They are also very unstable. The authors need to be more explicit about what they want to say here...

Minor points:

- Page 12: At the beginning of the paragraph the two *Lachancea* species are introduced in the abbreviated form of their name, which is spelled out only later in the same paragraph. It should be the other way around.

- Nomenclature: At several places (but not systematically), it is unclear whether the authors mention genes or their products. Genes should be italicised (marked // here) and protein names start with a capital letter followed by two small letters and a number. /RSC2/ is the gene, Rsc2 is its product but RSC2 is ambiguous.

- Page 14: Unlike yeast and metazoans, *D. discoideum*, is not an opisthokont but an Amoebozoa. Yeast and humans separated around 500 Mio years later from each other than they did from *Dictyostelium*. These three species cannot be said to be equidistant. It is even more interesting that such highly divergent species and their parasitic plasmids show similar features.

Referee #3:

Overall, Girard et al. perform compelling analyses to demonstrate that the budding yeast 2u plasmid - a rare eukaryotic innovation - tethers to specific regions of the genome. These binding sites correlate with reduced transcription and, intriguingly, may involve the recognition of emergent chromatin features. Additionally, this strategy appears to be employed by related and unrelated eukaryotic plasmids, in divergent yeast lineages and the amoeba *Dictyostelium discoideum*, suggesting that tethering to inactive chromosomal regions may be a clever means by which extrachromosomal DNA ensures its propagation and survival. The findings of the paper are well-supported and of broad interest given their generality. Ultimately, I believe this would merit publication in a journal such as EMBO. However, several major and minor issues exist that should be addressed prior to publication (see below). In particular, the "chromosome tethering" hypothesis requires further testing, both with pre-existing data and data from new experiments. Additionally, more evidence needs to be presented in support of the conserved nature of plasmid-chromosome contact sites. Finally, a more precise and careful presentation of the data would greatly improve the ability of this paper to be understood by those outside of the field. At least some of these revisions would need to be made before I can recommend the publication of the manuscript.

Necessary (major) concerns

1. The authors provide compelling support for the "chromosome hitchhiking" model of parasitic plasmid inheritance by demonstrating that the 2u plasmid makes various contacts with host chromosomes (Figure 1C). They also identify that the H4 5toA mutant is able to disrupt many of these contacts (Figure 3F). This presents an important opportunity to ask the question: does disrupting chromosome contacts disrupt plasmid propagation?
 - a. The authors indicate that pKan exhibits a reduced copy number in the H4 5toA mutant compared to wildtype (Supplemental Figure 5). However, this effect is too minor to be conclusive and the interpretation should be removed.
 - b. The effect of chromosome contacts could be addressed by re-analysis of existing data (e.g., what proportion of the reads map to the endogenous 2-micron plasmid in H4 5toA mutants?). Alternatively, performing qPCR for 2-micron plasmid levels could allow assessment of population level effects of the mutation on 2-micron stability.
 - c. Additional experiments using pre-existing methodology (e.g., FISH) could be employed to determine if the proportion of cells that carry the 2-micron plasmid is reduced in the H4 5toA mutant.
2. Given the importance of the claim that parasitic plasmids in different eukaryotes utilize similar persistence strategies, additional analysis should be performed to support the conclusion that the natural plasmids in *L. fermentati* and *L. waltii* exhibit similar contact site profiles to *S. cerevisiae*.
 - a. (1) Contact sites should be detected with a peak caller to support the existence of robust plasmid-chromosome contacts.
 - b. (2) It should be confirmed that the peaks detected correspond with longer genes (as in figure 2E), to support the interpretation that contact sites are similar in nature to those in *Saccharomyces cerevisiae*. The same analysis should be applied for *Dictyostelium discoideum*.
 - c. (3) This may be beyond the scope of the current paper, but RNAseq to demonstrate that the sites the plasmid binds to are

transcriptionally inactive would greatly strengthen the conclusion that this is a conserved strategy.

3. Throughout the paper, major issues with annotation exist in the figures, often to the point of precluding or interfering with proper interpretation. Axes or figure legends should, where appropriate, include information on units (e.g., for RNA-seq or ATAC-seq). Where possible, statistical analyses should be applied to confirm that observations made are not simply subjective (e.g., comparing peak height for different mutants). Application of the peak-calling algorithm more uniformly could also improve confidence in interpretation. Informally presented descriptions of genotypes or experimental set-up (e.g., almost all genotypes in Supplementary Figures 7 and 14) should be corrected, according to general yeast nomenclature guidelines (see below).

Minor concerns

NOTE: New experiment or analysis suggestions are denoted with a double asterisk (**)

1. General comments

a. Standard yeast nomenclature should be followed: genes (or derived alleles) are indicated in capital letters and italicized (e.g., RSC2 or RSC2-AID), proteins are indicated with the first letter capitalized and unitalicized (e.g., Rsc2), and deletions should be indicated in lower case and italics with the delta following (e.g., *rsc2*).

b. All abbreviations should be defined the first time used (e.g., 2u, SPB, Mb).

c. A careful readthrough should be performed to make sure that various typos and grammatical errors were not introduced during manuscript editing.

d. Make sure that figure and table references are as desired.

e. It may be useful to assign a unique term to the plasmid-chromosome contact sites that you continue to reference throughout the manuscript.

2. Abstract

a. The wording implies all SMC complexes were tested, when only cohesin and condensin (not Smc5/6) were tested. Reword appropriately.

3. Introduction

a. The speculation about the origin of the 2-micron plasmid from bacteriophages is more appropriate for the discussion than the introduction.

b. Additional references (including current references) should be used in discussing the mechanisms employed by 2-micron inheritance (e.g., a reference for Flp1).

c. The last paragraph of the introduction is repetitive with the first paragraph.

4. Results

a. Chromatinization and 3D folding of the 2u plasmid

i. ** The claim that "the level of transcription of 2u plasmid is 10 times less than the average for its host" would benefit from quantification.

ii. I think this section is meant to be split at the fragment "Discrete contacts between the 2u plasmid and host chromosomes"

iii. ** Stats should be used to support the idea that more peaks were at subtelomeres than randomly expected

iv. The text should clarify that plasmid binding directly to telomeres could not be tested due to their repetitive nature.

v. ** The peak-calling algorithm should also be applied to the ARS plasmid in order to support its "relatively even" binding.

b. REP1 and STB sequences are necessary for the specific attachment and stability

i. The nature of the pKan plasmid should be clearly introduced in the text (e.g., disruption of the FLP1 gene in all constructs). Additionally, the use of this acronym should be defined clearly in the Extended Data figure legends.

c. Plasmid/chromosomes contact regions are stable under a range of conditions

i. The Y9 strain does not seem to have "nearly identical contact hotspots profiles." Also, it should be made clear in the supplementary figure legend (or text) that it does not have the endogenous 2u plasmid, and so the pKan plasmid was used instead.

ii. The changes in 2u-chromosome contacts occur during the meiotic divisions, not during late meiotic prophase, as specified in the text.

iii. Interpretation of the *sir3* mutant is precluded by the inability to detect normal contact sites in the control for this dataset

d. The 2u preferentially tethers to inactive regions along the host's chromosomes

i. It may be more appropriate to refer to the long genes as "lowly expressed" as opposed to "non-expressed"

ii. ** For proper interpretation, a comparison of longer gene contact scores to shorter gene contact scores would be useful.

e. Plasmid tethering is rapidly reversible

i. ** The authors demonstrate that contact sites are rapidly "reversible" upon heat shock in Figure 2G. The interpretation of this data would be enhanced by using either RNAseq or qPCR to report on the gene expression changes of the genes featuring these changes (e.g., UTP20 and FIR1 + RZG8).

f. 2u plasmid tether to exogenous artificial inactive chromosome

i. ** By eye, the LacO array data does not clearly support the conclusions made. Is the peak at the LacO array called with a peak-caller in the absence of LacI? Does this peak disappear once LacI binds? Perhaps a different representation would make changes more apparent.

g. Plasmid anchoring depends on the H4 basic tail

i. ** It would be nice to improve quantification of how many peaks are called in the H4 5toA mutant. Are all 73 lost? Of course, the relevant wildtype comparison is to a strain with the pKAN plasmid. Are all 73 peaks detected for the pKAN plasmid?

ii. ** The authors indicate that 2u-chromosome contacts are reduced for the H4 5toA mutant, but only show this for the exogenous pKan constructs in log phase. This interpretation would benefit from analysis of the native 2u-chromosome plasmid

contacts during log phase in the H4 5toA mutant.

h. Other eukaryotic plasmids tethers to inactive chromatin

i. The entire paragraph is duplicated

5. Discussion

a. "Condensing" should be "condensin"

b. A citation should support the notion that the probability of inter-chromosomal contacts is "very low from a thermodynamic point of view"

6. Methods

a. Make sure that the tables are properly referenced

7. Tables

a. As with throughout, annotations of the genotypes used should be improved in the Tables 1 and 2.

b. An additional table with information about the data sets generated for this paper (e.g., Figure, Identifier, Strains) would be useful.

c. For Table 3

i. One "RSGY" is written as "RSGT"

ii. 2u-containing strains should be indicated [cir+] and 2u-lacking strains should be written as [cir-]. Indicate the status of the 2-micron plasmid in all strains.

iii. Indicate strain background for RSGY 1217

d. Table 4

i. Define pKAN first and make it clear that KanMX disrupts FLP1 in this strain. Make it clear that the other pKAN plasmids are derivatives of the original plasmid.

8. Figures

a. Figure 1

i. Currently, no units are present for the ATAC-seq, RNA-seq, and H3 chemical cleavage axes.

ii. 1A: Unclear what the cartoon is depicting. Indicate in the legend that "IR" stands for "Inverted Repeat," which is why its reads are unmappable. Indicate whether pink or blue represent the forward or reverse strands.

iii. 1C: According to the text, triangles should be annotating the dark bands in the chromosomal heat maps below.

iv. 1F: Given the deeper orange present across the quiescent chromosome, not clear why the contact score isn't generally higher in "quiescence."

b. Figure 2

i. Currently, no units are present for the RNA-seq data.

ii. 2F: Confused by what exactly is being displayed - given that signal/expected is lower than 1, aren't trans contacts between contact sites dis-enriched?

c. Figure 3

i. Currently, no units for CHIP H3, ATAC-seq, CHIP-exo signal, CHIP REP1.

ii. 3B: The color of yellow used does not print well.

iii. For 3A and 3B, a dotted line indicating the genomic average (and statistics) would support the interpretation that H3 occupancy and chromatin accessibility differ at 2u peaks.

iv. 3C: Currently difficult to interpret. May be useful to (1) label additional points of interest (e.g., proteins associated with active transcription) and (2) include dotted lines to indicate when changes observed are significant.

v. 3G: Make the dotted circle a rectangle that matches the inset.

d. Figure 4

i. Consider included a yeast silhouette as in the supplement for *L. fermentati* and *L. waltii*

9. Supplementary Figures

a. Supplementary Figure 1

i. No units for any plot

ii. Make it clear that H3 chemical cleavage, RNA-seq, and ATAC-seq are reproduced from 1A. Make data representation (e.g., strand portrayal and color choices) match. Interpretation would be facilitated by depicting Cse4 enrichment at a centromere.

b. Supplementary Figure 3

i. Consider indicating the contact sites with black triangles as in Supplementary Figure 2.

ii. 3A: Indicate what CHIP-seq REP1 signal represents (e.g., consider including an axis).

c. Supplementary Figure 4

i. Improve image quality (currently pixelated) and make sure that gene labels aren't cut off.

ii. Long gene gray boxes aren't visible

d. Supplementary Figure 5

i. Indicate references where the data was taken from

ii. Improve annotation of genotypes

iii. It may be worth providing this information for all mutants analyzed

e. Supplementary Figure 6

i. Relabel pKAN-STB-P as pKAN and make 6C 6A to indicate the other plasmids are derived from it.

ii. Not clear what "STB-P" is

f. Supplementary Figure 7

- i. Edit almost all labels to improve clarity re: experimental condition and genotype. A few of the issues (not exhaustive) are indicated below.
 - 1. No_tension = nocodazole
 - 2. What is "t4"?
 - 3. How was Fkh depleted? What genes are "Fkh" and "topo2" referring to? How were Clb5 and Clb6 disrupted?
- ii. The appropriate comparison for Y9 + pKan is BY + pKan
- iii. Indicate that the WT (microC) vs. quiescence (microC) data is a reproduction of Figure 1C.
- g. Supplementary Figure 9
 - i. Unclear whether longer genes just tend to be more lowly expressed, or if gene length and low expression both independently contribute to contact site formation
 - ii. 9F: Labels are illegible
- h. Supplementary Figure 10
 - i. As with Figure 2F, confused by what exactly is being displayed - given that signal/expected is lower than 1, aren't trans contacts between contact sites dis-enriched?
- i. Supplementary Figure 11
 - i. Centromere depiction doesn't match the label
 - ii. The red used for the "2 hour" timepoint is difficult to see
- j. Supplementary Figure 12
 - i. ** Co-localization quantification is missing and would help support the point made in Figure 2J
 - ii. Not clear what the clause "distribution of the number of 2u plasmid foci" is referring to
- k. Supplementary Figure 14
 - i. Improve annotation of all genotypes
 - ii. Seems as though dot1 , set2 , rsc2 , and hst2 data are all reproduced from the main figure
 - iii. For some of the strains (e.g., lno80, Fun2, and lsw2), the peaks in the control are so low that the experiments aren't interpretable
 - iv. Unclear whether the difference observed for SWR1-AID is significant. By eye, effect size seems similar to what is observed for set2 , albeit in the opposite direction
- l. Supplementary Figure 15
 - i. ** Unclear why the WT in Figure S15A and S15C is labeled HHF2. Also, is the dip observed for Rbp3 binding in the H4 5toA mutant not significant? A line depicting the genomic average and statistics could improve interpretation.
 - ii. S15D: Misleading to compare the wildtype 2u contact peaks during log phase to the mutant contact peaks during quiescence - this graph should be removed (or the WT quiescence should be used as the control). Additionally, not clear what the HHF2 control is.
 - iii. S15E: The centromeres are not visible and the long genes are not consistently depicted (e.g., height is not identical)
 - iv. Figure legend indicates that "pKAN version of the 2u plasmid was used to ensure plasmid stability" for S15B - not clear that there is impaired plasmid stability in the H4 5toA mutant. If so, why was this not necessary for S15D?
- m. Supplementary Figure 16
 - i. S16A and B: Chromosome labels can't be read
 - ii. S16B: A contact plot with long genes annotated should be included for at least one chromosome.
 - iii. S16D: Centromeres should be depicted (if known). Long gene annotation difficult to see.

Point-by-point response to reviewers' comments

We thank the reviewers for their thorough reading of our manuscript and the detailed reviews that helped us to improve the work.

Referee #1 (Report for Author)

This paper is concerned with the localization of the yeast 2μ plasmid with potential relevance for its maintenance mechanisms. These have been the subject of extensive study and analysis, since 2μ is an intriguing example of a eukaryotic genetic element in the form of a circular plasmid that appears to act as a relatively benign "molecular parasite". 2μ plasmid uses an otherwise unprecedented mechanism involving a site-specific recombinase (FLP) to regulate its copy number by inverting the direction of migration of replication forks (Futcher, 1986; *J. Theor. Biol.* 119, 197-204). A second gene (RAF) is involved in activating FLP transcriptional regulation through blocking the regulatory action of the final two genes REP1 and REP2. However the major function of Rep1 and Rep2 is acting on a cis-plasmid locus known as STB to confer plasmid stability through effective partitioning during mitotic and meiotic divisions. The work of multiple labs has demonstrated that the partitioning/segregation mechanism of 2μ plasmid involves interaction of plasmid components with yeast chromosomes.

The current manuscript is very well written and clear, and provides a significant amount of data and analysis. Using proximity ligation techniques (Hi-C and Micro-C), the authors map contacts between 2μ plasmid and yeast chromosomes. The authors report that the plasmid links preferentially to specific regions of low transcriptional activity. The localization of plasmid DNA sequence is supported by REP1 ChIP analysis, suggesting REP1 protein is also localized to these regions as expected from its function. However this localization does not appear to depend on a number of other factors including the SMC complex previously reported as important for 2μ plasmid segregation. The authors conclude that 2μ contacts with these sequences depend on a nucleosomal signal associated with depletion of RNA polymerase II. The authors further show that similar patterns of association apply to the *Dictyostelium* plasmid and suggest it may be a more widespread mechanism.

The work represents a comprehensive application of proximity mapping techniques to the association of the yeast 2μ plasmid with yeast genomic sequences, and as such is a significant update on previous studies that lacked the resolution of Hi-C type techniques. Almost half a century ago, Livingston and Hahne (1979; <https://pubmed.ncbi.nlm.nih.gov/386345/>) and Nelson and Fangman (1979) (<https://pubmed.ncbi.nlm.nih.gov/392520/>) confirmed that the plasmid has a conventional nucleosome structure, and evidence for an association of 2μ with specific nuclear structures originated with work by Wu et al (1987; <https://pubmed.ncbi.nlm.nih.gov/3542994/>). Scott-Drew et al (2002; <https://pubmed.ncbi.nlm.nih.gov/12095225/>) used imaging approaches to conclude that the STB locus of 2μ (together with the plasmid proteins Rep1 and Rep2) localizes to discrete chromatin sites, and consistent with the current report showed that these sites are distinct from telomeres. Additional more recent studies are cited in the manuscript.

We would like to thank referee 1 for their comments and perspective on the work. We have taken into account these references and provide a more complete introduction on the work done on the 2 μ over the last 50 years.

Functional analysis showed the involvement of cohesin in the maintenance mechanism of 2 μ (<https://pubmed.ncbi.nlm.nih.gov/17670945/> ; <https://pubmed.ncbi.nlm.nih.gov/15169893/>) as well as the involvement of the "remodels the structure of chromatin" (RSC) chromatin remodeling complex, the nuclear motor Kip1, and histone H3 variant Cse4 (as discussed in the current manuscript). However a full understanding of how these components contribute to the 2 μ partitioning system remains elusive.

Interestingly, the current work shows no evidence of effect of these various factors on plasmid association with the identified genomic sequences, despite clear evidence that they are required for plasmid maintenance. This may suggest that the specific plasmid DNA localization documented is not necessarily required for effective plasmid segregation. In other words, some or all of the plasmid may be preferentially associated with particular sites, but evidence that this has a functional significance does not appear to be clear. It would be useful to know what proportion of the total plasmid is associated with the chromatin, or is some plasmid "free" in the nucleoplasm?

This question is difficult to answer. Measuring free and immobile molecules in the small yeast nucleus, for instance using HaloTag and SNAP-Tag fusions, requires state-of-the-art microscopy approaches and techniques. We agree with the reviewer that this work is worth trying, but we also think that it represents an independent study that goes beyond the scope of this work. We nevertheless agree that our discussion should point at the possibility raised here.

Note that we processed several cohesin and condensin ChIP-seq data and did not observe any significant enrichment of SMCs at the STB site when normalised with the input. It is unclear why there is such discrepancy with the ChIP analysis. We mention this observation in the text and put now the ChIP seq analysis in **Appendix Fig. S1**.

We added the following sentence to the text:

"We also did not detect any enrichment of cohesin and condensin complexes at the STB site."

The manuscript makes reference to plasmid copy number, here determined by average read counts (e.g. page 6: "2 μ plasmid mutants lacking either REP1 or the STB loci ... These mutant plasmids are unstable (Kikuchi, 1983; McQuaid et al., 2019) and carry a marker gene to be retained in the cell. This instability is reflected by a relatively low copy number per cell, which can be assessed from the proportion of reads from 2 μ in each library".) However this is not a useful measure of copy number. What is being measured is the average copy number relative to total extracted DNA, not average copy number per plasmid-containing cell, which is the relevant measure. As analyzed in detail in the Kikuchi (1983) paper, unstable plasmids still replicate once per cell cycle; they are unstable due to a defective partition system and hence accumulate to high copy numbers in a small proportion of cells in the population. Hence it is necessary to measure both the proportion of plasmid-containing cells (using a selective plating assay) and the average copy number and hence calculate the plasmid copy number per plasmid containing cell. Without direct determination of the proportion of plasmid-containing cells, it is not possible to draw conclusions regarding plasmid segregation or the copy number of plasmids in cells that contain them.

This is a valid point and we agree with the referee that the proportion of reads from the 2 μ plasmid in a library does not give a complete picture of stability, in particular how the segregation process is affected. See below for a better assessment of the impact of the 5toA tail on plasmid stability and the proportion of cells having the plasmid.

This is also relevant on page 10, where it is stated that: "RSC2 was shown to be essential for the 2 μ plasmid maintenance (Wong et al., 2002). We observed indeed a significant drop in the number of plasmids per cell in this mutant (Extended Data Fig. 5). However, its attachment to host chromosomes remain unchanged".

Again this statement is not correctly phrased- what is affected in the rsc2 mutant is the **segregation of the plasmid** and it therefore accumulates in a subset of cells to a higher than normal copy number and the average copy number across all cells in the population falls due to the large number of cells that have lost the plasmid and still continue to divide for a while. Nevertheless the conclusion that Rsc2p is not needed for normal plasmid association with identified chromatin regions is supported by the data.

We agree with the referee to make this important distinction and we rephrase the sentences accordingly:

"Rsc2 was shown to be essential for the 2 μ plasmid segregation and to overcome maternal inheritance bias (Wong et al., 2002)."

The identified sites of 2 μ association are shown here to be associated with low transcriptional activity, particularly containing long genes. In these regions there is enrichment of H3K36me3, and in the Set2 mutant (Set2p methylates H3K36me3) the level of transcription of long genes is slightly increased and contact specificity for the 2 μ plasmid is substantially reduced (but still detectable). Does this indicate causative association of 2 μ with low transcribed regions/ long genes or correlation with wider changes induced by loss of H3K36me3?

We interpret this experiment as follows: deletion of the SET2 gene results in the absence of the H3K36me3 epigenetic mark and derepression of long genes (as shown in <https://genesdev.cshlp.org/content/21/11/1422.full>). We do checked that the level of transcription at the level of the long genes is indeed increased in a *set2Δ* mutant using RNA-seq data from <https://www.ncbi.nlm.nih.gov/pmc/articles/PMC5770421/>) (**Appendix Figure S12B**). We propose that this increased level of transcription modifies the chromatin state and leads to the fact that long genes are no longer preferential contact sites for the 2 μ .

Certain regions remain contact hotspots for the 2 μ plasmid: for instance, the subtelomeric regions or regions inside chromosome arms (**Appendix Figure S12B**, chrVIII). The fact that in absence of methyltransferase Set2 and the associated H3K36me3 mark, contact hotspots remain detectable suggests that this epigenetic mark is not directly necessary for specific contacts of the 2 μ plasmid. The decrease we observe on the agglomerated plots (Extended Figure 15A and 15C) results probably from changes in local transcription levels and in chromatin state. We clarify this point and explain our interpretation in the reviewed version of the work and we also added an example of hotspot of contact of 2 μ plasmid in the *set2Δ* mutant that shows that the histone mark associated to Set2 is not essential to create specific hotspots of contact.

In considering the factor(s) by which the 2 μ associates with these regions identified as depleted in PolII activity, the authors identify the tail of histone H4 as required to confer localization based on a five-alanine substitution variant. A key question this raises is whether plasmid segregation is affected in this mutant H4 mutant, which would be predicted if the

localization of plasmid contributes to its partition mechanism. Identifying the nature and proteins involved with the H4 tail interaction could also provide more functional insight.

This is an important experiment, also suggested by other reviewers. We have therefore tested for the stability of the plasmid in the H4 5toA mutant using a pKAN plasmid system (inspired from McQuaid, Polvi, et Dobson 2019 and Kikuchi 1983). The plasmid segregation was affected in this mutant as compared to the WT. We have included this result in the main text, **Fig. 3H**.

H, Stability measurements showing inheritance of the 2 μ m-based (pKan) plasmids in a *cir*⁰ yeast strain determined by plating assays for WT, 5toA, rep1 Δ and Δ STB mutants. Results represent the average (\pm s.d.) from assaying three transformants for each plasmid, respectively. Asterisks indicate significant differences determined by a chi square test (* P < 0.05, ** P < 0.005, *** P < 0.0005).

We added the following sentences in the main text:

“To test the stability of the plasmid in the 5toA mutant, we measured the proportion of cells containing the artificial pKAN plasmid after overnight culture in a selective medium (**Fig. 3H**, **Fig. EV4F**, Methods, as described in (McQuaid, Polvi, et Dobson 2019)).

We observed a difference between the wild-type strain and the 5toA mutant (average proportion of cells having the plasmid, in WT~29% vs mutant 5toA~24.5%, p-value<0.001, Chi squared test, see Methods). We also calculated this proportion along 3 time-point kinetics at 6h, 12h and 24h after the overnight culture in non-selective media. The difference

is significant along the 3 time points (**Fig. 3H**). We also detect a slight but significant change in the average number of plasmids per plasmid-possessing cell (**Fig. EV4G**). These observations may reflect perturbed segregation of the pKAN plasmid in the 5toA mutant.”

We performed another serie of measurements that we added in **Fig. EV4F** as well as the average number of plasmids per cell using shotgun sequencing and the values of proportions of cells having plasmids:

F, Stability measurements showing inheritance of the 2 μ m-based (pKan) plasmids in a *cir^D* yeast strain determined by plating assays for WT and 5toA mutant (replicate 2). Results represent the average (\pm s.d.) from assaying 5 transformants for each plasmid respectively. Asterisks indicate significant differences determined by a chi square test (* $P < 0.05$, ** $P < 0.005$, *** $P < 0.0005$). **G**, Average number of plasmids per cell computed taking into the proportion of cells having plasmids and reads proportion from plasmid sequence with shotgun sequencing. Results represent the average (\pm s.d.) from assaying 3 transformants for each condition respectively and correspond to overnight culture shown in **Fig. 3H**.

For the question regarding proteins involved with the H4 tail interaction, it should be noted that in the region of the tail of histone H4 on which the amino acid substitutions were made, 2 histone marks are present: H4K16Ac and H4K20Ac.

To test for other potential molecular partners, in particular those linked to these 2 histone marks, we generated novel Hi-C data and computational analyses of the following strains:

- H4 mutant with deletion of amid acids from position 15 to 17, to test whether acetylation on lysine 16 (H4K16Ac) is necessary for plasmid attachment.
- H4 mutant with deletion of amid acids from position 18 to 20, to test whether acetylation on lysine 20 (H4K20Ac) is necessary for plasmid attachment.

Hotspots contacts are retained in both strain and are undistinguishable from WT (see Figure below) which have been added to **Fig. EV4H**.

- Furthermore, in addition to the ~800 proteins that we tested for occupancy enrichment on the contact hot spots of 2 micron plasmid (Fig. 3C), we present enrichment analyses for 3 other genomic signals giving a signature of depletion or enrichment at the regions contacted by 2 μ :

Appendix Figure S11: Averaged plot around contacted regions by 2 μ for 3 genomic signals.

GapR ChIP-seq signal giving the presence of positive supercoiling is depleted around regions contacted by the 2 μ . yH2A ChIP-seq signal giving the presence of the histone mark yH2A which is associated with yeast heterochromatin is depleted around regions contacted by the 2 μ . Dinoflagellate-viral-nucleoproteins (DVNPs) expressed in yeast show enrichment in regions contacted by 2 μ .

We added the following test in the main text:

“We explored many other different genomic signals, most of which do not show an enrichment signature (data not shown). Some, however, show a specific average signal: the GapR ChIP-seq signal, which indicates the presence of positive supercoiling (Guo et al. 2021), is depleted in the regions contacted by the 2 μ (**Appendix Figure S11A**). Also, unexpectedly, dinoflagellate-viral-nucleoproteins (DVNPs) expressed in yeast show enrichment in regions contacted by 2 μ (**Appendix Figure S11A**). These proteins are supposed to localise to histone binding sites (Irwin et al. 2018).“

Overall the manuscript provides compelling data for the preferential localization of 2 μ plasmid with specific regions of chromatin with higher levels of transcriptional inactivity. Given the relocation to alternative regions with low transcriptional activity in set2 mutants, and failure to associate in histone H4 tail mutants, does this reflect a general association preference of 2 μ with less transcriptionally active regions but without particular functional significance? Testing the segregation of plasmids in these mutant backgrounds could provide supportive evidence and further our understanding of 2 μ partition mechanism.

We agree with this comment and the stability measurements have been carried out in the mutant 5toA, rep1 Δ and Δ STB mutants (see above).

Minor points:

"long genes have their contact signal with the 2 μ plasmid greatly reduced while the level of transcription is slightly increased (Supplementary Fig. 13c)." should read "Supplementary Fig. 14c".

This has been corrected.

Methods do not always provide sufficient detail to allow experimental repeat- eg REP1 ChIP experiment.

We now provide more details on the experimental procedure notably the Rep1 ChIP-seq and added the Methods section:

Referee #2 (Report for Author)

The manuscript entitled "Anchoring of parasitic plasmids to inactive regions of eukaryotic chromosomes through nucleosome signals" describes the interactions that parasitic plasmids establish with host genomes, using available and newly generated Hi-C and Micro-C data. These studies focus primarily on the 2-micron of *Saccharomyces cerevisiae* and demonstrate that it associates with long, lowly expressed genes. The contacts of the plasmid with these target genes depend on the plasmid-encoded protein Rep1 and its binding site, the STB sequence, on the plasmid. Since Rep1 and STB are required for the proper maintenance of the plasmid in the population, these data suggest that association of the plasmid with its target loci on the chromosome contribute to its mitotic stability. The interaction of the plasmid with the target chromosomal loci depended on the basic patch on the tail of histone H4. Analysis of the interaction of other parasitic plasmids with the chromosomes of their hosts, such as plasmids related to the 2-micron in other fungi and the Ddp5 plasmid in *Dictyostelium discoideum*, indicate that they also contact long, poorly expressed genes. Together, these data support the idea that these parasitic plasmids propagate between sister-chromatids by hitchhiking on chromosomes and indicate that for that they use low expressed genes as binding sites. This may help keep their fitness-cost very low for the host.

Together, this is a very thorough and interesting study that addresses convincingly and fairly conclusively for the first time a long-standing question. The data are likely to be relevant for a broad variety of parasitic genomes, including viruses that remain episomal during their latency phase and hitchhike on chromosomes during cell division, such as the Epstein-Barr virus (EBV) and the Hepatitis B virus (HBV). The data are convincing, well presented and well documented. However, a few simple tests would significantly enhance the relevance of the work and some statements probably need revision. Provided that these points are addressed, this reviewer fully support the publication of this important and beautiful piece of work.

We thank referee 2 for their comments and provide below a point by point response to their concern and suggestions.

Main points:

1- The authors convincingly show that mutating the basic patch of H4 tail strongly impair the interaction of the 2-micron with its target loci. Furthermore, the authors conclude from the fact that there are less reads for the 2-micron in the H4-5toA mutant cells that it must be less stable in these cells. However, other mechanisms could also reduce the number of copies without affecting mitotic propagation. The authors need to characterize the mitotic stability of the 2-micron plasmid in the H4 mutant cells to demonstrate that it is less stable indeed. This is an essential test for supporting their conclusions. It is also a simple test.

The easiest assay is to measure the number of cells that have lost the plasmid after growth on non-selective medium, taking wild type as positive control and the mutant rep1- or stb-plasmids as negative controls. If the stability is not down to the mutant level, this would indicate that hitchhiking is not the only parameter ensuring the mitotic stability of the plasmid. If it is even still as stable as in the wild type, this would indicate that association with chromosomes is not relevant for partition but for some other process.

This is a key experiment also suggested by other reviewers, and we have now done this experiment. Our result points at an increased instability of the 2 μ plasmid in the 5toA H4 tail mutant, suggesting a functional role of this peptide in the segregation of the 2 μ . The results are now described in the **Fig. 3H**, **Fig. EV4G**, **Fig. EV4H** (see above).

2- The authors show that removing the Rsc2 protein does not impair interaction of the 2-micron with its target genes, although it strongly affects its maintenance. Discussion is needed.

The impact of Rsc2 protein deletion has been shown previously by Wong et al. 2001 through a genetic screen (<https://pubmed.ncbi.nlm.nih.gov/12024034/>). Rsc2 protein is a subunit of the RSC chromatin remodelling complex and can affect transcription regulation. We propose that Rsc2p has an effect on the stability of the 2 μ plasmid that is independent of the direct contacts we described here since the contact signal is not affected. For example, this could be by disrupting the expression of genes essential for the maintenance of 2 μ .

We added the following sentence in the main text taking also into account remark from referee2:

“The Rsc2 protein is part of the RSC chromatin remodeling complex and can affect the regulation of transcription of numerous genes. It is possible that the reported effect of Rsc2p on 2 μ plasmid segregation can be explained by an indirect effect such as impact on the expression of genes required for proper 2 μ plasmid maintenance.”

3- The authors state at the end of their discussion that they envision that eccDNAs, excised from the genome could also rely on the same principle (implicating, for their segregation). However, such eccDNAs do not encode either of Rep1 and Rep2 and do not contain an STB site. They are also very unstable. The authors need to be more explicit about what they want to say here...

We agree with the reviewer's implicit suggestion that this comment is too speculative. We decided to remove this sentence.

Minor points:

- Page 12: At the beginning of the paragraph the two *Lachancea* species are introduced in the abbreviated form of their name, which is spelled out only later in the same paragraph. It should be the other way around.

The modification has been made.

- Nomenclature: At several places (but not systematically), it is unclear whether the authors mention genes or their products.

Genes should be italicised (marked // here) and protein names start with a capital letter followed by two small letter and a number. */RSC2/* is the gene, Rsc2 is its product but RSC2 is ambiguous.

We apologise for these mistakes. All text, figures and supplementary have been checked for correct nomenclatures.

- Page 14: Unlike yeast and metazoans, *D. discoideum*, is not an opisthokont but an Amoebozoa. Yeast and humans separated around 500 Mio years later from each other than they did from Dictyostelium. These three species cannot be said to be equidistant. It is even more interesting that such highly divergent species and their parasitic plasmids show similar features.

We thank the referee for pointing out this mistake and changed the text at the end of the introduction:

“Since Amoeba belongs to Amoebozoa, a group that preceded divergence with Opisthokont that encompasses animals and fungi, these results suggest that this property may be a widespread strategy for eukaryotic plasmids to ensure their correct segregation during cell division in a way that does not disrupt host regulation.”

Referee #3 (Report for Author)

Overall, Girard et al. perform compelling analyses to demonstrate that the budding yeast 2 μ plasmid - a rare eukaryotic innovation - tethers to specific regions of the genome. These binding sites correlate with reduced transcription and, intriguingly, may involve the recognition of emergent chromatin features. Additionally, this strategy appears to be employed by related and unrelated eukaryotic plasmids, in divergent yeast lineages and the amoeba *Dictyostelium discoideum*, suggesting that tethering to inactive chromosomal regions may be a clever means by which extrachromosomal DNA ensures its propagation and survival.

The findings of the paper are well-supported and of broad interest given their generality. Ultimately, I believe this would merit publication in a journal such as EMBO. However, several major and minor issues exist that should be addressed prior to publication (see below). In particular, the "chromosome tethering" hypothesis requires further testing, both with pre-existing data and data from new experiments. Additionally, more evidence needs to be presented in support of the conserved nature of plasmid-chromosome contact sites. Finally, a more precise and careful presentation of the data would greatly improve the ability of this paper to be understood by those outside of the field. At least some of these revisions would need to be made before I can recommend the publication of the manuscript.

We thank this reviewer for their comments and appreciate the care given to the presentation of the results and figures presented in our article. We do apologise for the typos or omissions pointed out.

Necessary (major) concerns

1. The authors provide compelling support for the "chromosome hitchhiking" model of parasitic plasmid inheritance by demonstrating that the 2 μ plasmid makes various contacts with host chromosomes (Figure 1C). They also identify that the H4 5toA mutant is able to disrupt many of these contacts (Figure 3F). This presents an important opportunity to ask the question: does disrupting chromosome contacts disrupt plasmid propagation?

This point was raised by the other reviewers as well. We have now addressed the stability of the plasmid in Fig. 3H with several measurements, showing a decreased stability of the 2 μ in the 5toA H4 tail mutant compared to the WT (see above).

a. The authors indicate that pKan exhibits a reduced copy number in the H4 5toA mutant compared to wildtype (Supplemental Figure 5). However, this effect is too minor to be conclusive and the interpretation should be removed.

We agree with the referee that the copy number information is not robust enough. We now performed a stability experiment to confirm this observation and we removed the associated sentence. We also computed the number of plasmid per cell with shotgun sequencing and taking into account the proportion of cells containing plasmid (see above).

b. The effect of chromosome contacts could be addressed by re-analysis of existing data (e.g., what proportion of the reads map to the endogenous 2-micron plasmid in H4 5toA

mutants?). Alternatively, performing qPCR for 2-micron plasmid levels could allow assessment of population level effects of the mutation on 2-micron stability.

The heterogeneity of 2 μ presence in individual cells suggest that qPCR and contacts are not the best approach to assess for the maintenance and stability of the plasmid in a population. A stability measurement we now provide gives more direct evidence of stability change in the 5toA mutant.

c. Additional experiments using pre-existing methodology (e.g., FISH) could be employed to determine if the proportion of cells that carry the 2-micron plasmid is reduced in the H4 5toA mutant.

We appreciate the suggestion. Given the size of the plasmid and former experiments using FISH, quantification would be tricky (studies have drawn contradicting conclusions through microscopy imaging of this repeated element in the past <https://pubmed.ncbi.nlm.nih.gov/34270553/> <https://pubmed.ncbi.nlm.nih.gov/11266454/>). As discussed above, we have decided to assess the stability using genetics, as it appeared as a faster and quantitative alternative, and reserved imaging for a future work. We hope the reviewer will be convinced this was indeed a good solution to that question.

2. Given the importance of the claim that parasitic plasmids in different eukaryotes utilize similar persistence strategies, additional analysis should be performed to support the conclusion that the natural plasmids in *L. fermentati* and *L. waltii* exhibit similar contact site profiles to *S. cerevisiae*.

a. (1) Contact sites should be detected with a peak caller to support the existence of robust plasmid-chromosome contacts.

Indeed. We generated new Hi-C data for these 2 species and applied automatic peak detection, reproducing results similar (e.g. contacts hotspots) to those obtained for *Saccharomyces cerevisiae*. We added the following sentence:
“Automatic detection of contact peaks identified 52 peaks for *L. fermentati* and 55 peaks for *L. waltii*.”

b. (2) It should be confirmed that the peaks detected correspond with longer genes (as in figure 2E), to support the interpretation that contact sites are similar in nature to those in *Saccharomyces cerevisiae*. The same analysis should be applied for *Dictyostelium discoideum*.

We have now added statistical tests to confirm the over-representation of long, silent genes in the contact hotspots regions detected in the other organisms.
We added the following sentence:

“For *L. fermentati*, of the 18 long genes (size > 7kb), 12 are detected as plasmid contact hotspots (p-value < 0.001, Methods) and for *L. waltii*, of the 19 long genes (size > 7kb), 8 are detected as plasmid contact hotspots (p-value < 0.001, Methods). ”

“For *Dictyostelium discoideum*, of the 76 long genes (size > 10kb), 20 are detected as plasmid contact hotspots (p-value < 0.001, Methods). “

c. (3) This may be beyond the scope of the current paper, but RNAseq to demonstrate that the sites the plasmid binds to are transcriptionally inactive would greatly strengthen the conclusion that this is a conserved strategy.

We generated RNA-seq datasets of the two *L. waltii* and *L. fermentati* strains analyzed using Hi-C, to indeed demonstrate that the plasmids in these species do contact, as in *S. cerevisiae*, transcriptionally inactive regions. The results are now presented in **Fig. 4B**.

We added the following sentence:

“We also calculated the average transcription signal profile with RNA-seq and it shows transcription depletion at the contact hotspots of natural plasmids (**Fig. 4B, right**).”

3. Throughout the paper, major issues with annotation exist in the figures, often to the point of precluding or interfering with proper interpretation. Axes or figure legends should, where appropriate, include information on units (e.g., for RNA-seq or ATAC-seq). Where possible, statistical analyses should be applied to confirm that observations made are not simply subjective (e.g., comparing peak height for different mutants). Application of the peak-calling algorithm more uniformly could also improve confidence in interpretation. Informally presented descriptions of genotypes or experimental set-up (e.g., almost all genotypes in Supplementary Figures 7 and 14) should be corrected, according to general yeast nomenclature guidelines (see below).

We apologise for this lack of rigor, and have reworked and paid extra care to the various annotations, statistical tests, nomenclatures and else present in the main text and the additional data.

Minor concerns

NOTE: New experiment or analysis suggestions are denoted with a double asterisk (**)

1. General comments

a. Standard yeast nomenclature should be followed: genes (or derived alleles) are indicated in capital letters and italicized (e.g., RSC2 or RSC2-AID), proteins are indicated with the first letter capitalized and unitalicized (e.g., Rsc2), and deletions should be indicated in lower case and italics with the delta following (e.g., rsc2).

We went through the entire manuscript again, modifying all nomenclatures where necessary.

b. All abbreviations should be defined the first time used (e.g., 2 μ , SPB, Mb).

This has been checked and corrected.

c. A careful readthrough should be performed to make sure that various typos and grammatical errors were not introduced during manuscript editing.

d. Make sure that figure and table references are as desired.

e. It may be useful to assign a unique term to the plasmid-chromosome contact sites that you continue to reference throughout the manuscript.

We have reworked in detail the various annotations, nomenclatures in the main text and the additional data.

2. Abstract

a. The wording implies all SMC complexes were tested, when only cohesin and condensin (not Smc5/6) were tested. Reword appropriately.

We apologised, this was a mistake. We actually had tested Smc5/6 but the data were not shown in the article. This is relevant and important since Smc5/6 has been proposed to be involved in the 2 μ plasmid stability in a recent work of the Malik lab (<https://pubmed.ncbi.nlm.nih.gov/33063663/>).

Hi-C data has also recently been generated for mutants with depletion of the Smc5/Smc6 complex by our lab and by the Björkegren laboratory (<https://www.ncbi.nlm.nih.gov/bioproject/PRJNA986466>).

We therefore analysed these data and show that the contacts made by the natural 2 μ plasmid are similar in both WT and Smc5/Smc6 depleted mutants. We now have added in the supplementary data these analyses concerning the mutants linked to the Smc5/Smc6 complex (**Fig. EV2C**), and confirm that the contacts between the 2 μ and chromosomal hotspots occur independently of all three SMC complexes.

3. Introduction

a. The speculation about the origin of the 2-micron plasmid from bacteriophages is more appropriate for the discussion than the introduction.

We agree that this point is speculative and is now mentioned in the discussion section.

b. Additional references (including current references) should be used in discussing the mechanisms employed by 2-micron inheritance (e.g., a reference for Flp1).

We added the following references to this discussion:

<https://pubmed.ncbi.nlm.nih.gov/3939256/>

Veit BE, Fangman WL. Chromatin organization of the *Saccharomyces cerevisiae* 2 microns plasmid depends on plasmid-encoded products. *Mol Cell Biol.* 1985 Sep;5(9):2190-6. doi: 10.1128/mcb.5.9.2190-2196.1985. PMID: 3939256; PMCID: PMC366943

<https://pubmed.ncbi.nlm.nih.gov/9625741/>

Scott-Drew S, Murray JA. Localisation and interaction of the protein components of the yeast 2 μ circle plasmid partitioning system suggest a mechanism for plasmid inheritance. *J Cell Sci.* 1998 Jul;111 (Pt 13):1779-89. doi: 10.1242/jcs.111.13.1779. PMID: 9625741.

c. The last paragraph of the introduction is repetitive with the first paragraph.

We corrected the redundancy.

4. Results

a. Chromatinization and 3D folding of the 2 μ plasmid

i. ** The claim that "the level of transcription of 2 μ plasmid is 10 times less than the average for its host" would benefit from quantification.

We thank the referee for this suggestion and quantified the level of transcription for each gene of the 2 μ plasmid, taking into account the number of copies of the plasmid present in the cells. We generated ChIP seq data for RNA Pol II in asynchronous WT cells. The number of copies of each gene of the plasmid was normalised using the input signal.

We computed, for two independent experiments, the number of reads overlapping each gene in the RNA Pol II ChIP-seq signal, divided by the number of reads overlapping each gene in the input signal (using htseq-count with default parameters; Anders et al., *Bioinformatics*, 2015 PMID: 25260700). This analysis showed that the level of transcription of the 2 μ plasmid genes is only twice smaller than the average host genes transcription and therefore does not significantly differ from it. This result does not modify the interpretation but we corrected the original text and added the analysis in the supplementary data. We thank the reviewer for suggesting this control analysis.

We thus added in the main text:

“Taking into account the number of copies per cell of the 2 μ plasmid, the level of transcription of the 4 genes of 2 μ plasmid is 2 times less than the median expression of genes for its host (**Appendix Figure S1B**).”

and in the Methods section:

“To quantify the level of gene transcription of 2 μ plasmid taking into account the number of copies per cell, we computed, for two independent experiments, the number of reads overlapping each gene in the RNA Pol II ChIP-seq signal, divided by the number of reads overlapping each gene in the input signal (using htseq-count with default parameters; Anders et al., Bioinformatics, 2015 PMID: [25260700](https://pubmed.ncbi.nlm.nih.gov/25260700/)). We computed the distribution of levels of transcription for the genes of *S. cerevisiae* using home python code. ”

ii. I think this section is meant to be split at the fragment "Discrete contacts between the 2 μ plasmid and host chromosomes"

That's correct and this has been corrected.

iii. ** Stats should be used to support the idea that more peaks were at subtelomeres than randomly expected

We counted the number of contact hotspots of 2 μ that overlap one of the 32 sub-telomeric regions (defined as the 30 kb first at the start and end of each of the 16 chromosomes). We detected 14 hotspots overlapping a sub-telomeric region. We computed a null model to test for the expectation (Methods). Briefly, the null model was made using 1,000 groups of random positions of similar size. In the random group, one instance only 14 hotspots ended up in these regions, corresponding to a p-value of 0.001. We also used this bootstrap strategy to assess significance for the group of long genes in the different species.

We added the following sentence in the text:

“We detected 14 contact hotspots overlapping a sub-telomeric region (30 kb at the start and end of each of the 16 chromosomes), a significant enrichment compared to random group realisations (p-value= 0.001, Methods).”

And we also added the following paragraph in the Methods section:

“To determine the potential significance of contact hotspots enrichment of the 2 μ plasmid in subtelomeric regions, we computed a null model as follows. We used the code *overlap_with_group.py*, which generates 1,000 realisations of genomic positions of the same size as the contacts detected. The proportion of peaks that overlap a genomic position group is calculated for the 2 μ plasmid contacts and for each of the 1,000 random realisations. For subtelomeric regions, we took 30 kb at chromosome extremities. The p-value is given by the number of random configurations with a number greater than or equal to the number detected on the experimental data divided by the total number of realisations. Same strategy was used to assess significance for the group of long genes in the different species.”

iv. The text should clarify that plasmid binding directly to telomeres could not be tested due to their repetitive nature.

We agree. It is true that the repetitive nature of telomeric regions can prevent them to be directly mapped so we added the following sentence in the main text:

“It should be noted that exact telomeric regions in the yeast *S. cerevisiae* are not perfectly mappable due to the presence of repeated sequences. The number of contact peaks overlapping with telomeric regions could therefore be difficult to establish.”

v. ** The peak-calling algorithm should also be applied to the ARS plasmid in order to support its "relatively even" binding.

We added a peak calling analysis for this plasmid and added the analysis in the Appendix Figs. S3B,C.

- b. REP1 and STB sequences are necessary for the specific attachment and stability
 - i. The nature of the pKan plasmid should be clearly introduced in the text (e.g., disruption of the FLP1 gene in all constructs). Additionally, the use of this acronym should be defined clearly in the Extended Data figure legends.

We give the information in the main text and in the legends of the Supplementary Figures.

We added notably the following sentence:

“These mutant plasmids are unstable (McQuaid et al., 2019) so a pKAN plasmid version was used which contains the *kanMX4* gene cassette that disrupts FLP1 gene to maintain the plasmids in the cells.”

- c. Plasmid/chromosomes contact regions are stable under a range of conditions

- i. The Y9 strain does not seem to have "nearly identical contact hotspots profiles." Also, it should be made clear in the supplementary figure legend (or text) that it does not have the endogenous 2u plasmid, and so the pKan plasmid was used instead.

We agree with the referee that the signal is weaker, but the specificity is clearly visible when the contact signals are applied chromosome by chromosome. We added in the legend that the plasmid used for this mutant was the pKAN version of the 2μ. In addition, we've added a control for this mutant, the WT with plasmid pKAN, which shows that the average signals are close.

- ii. The changes in 2u-chromosome contacts occur during the meiotic divisions, not during late meiotic prophase, as specified in the text.

We checked this point and the changes in 2 μ -chromosome contacts occur during the meiotic divisions, not during late meiotic prophase. We have corrected the text accordingly.

“A small general increase in contact variability was observed at the later stages of meiotic divisions (**Appendix Fig. S6**), which could reflect increased compaction of chromosomes or change in chromatin state at this step.”

iii. Interpretation of the sir3 mutant is precluded by the inability to detect normal contact sites in the control for this dataset

We agree with this comment. It should be noted that the signal between different chromosomes, or more generally between 2 different molecules, is more difficult to capture in contact technologies than the contact signal within the same chromosome. The inter-molecule contact signal is therefore more subject to fluctuations and experimental variations. The Hi-C libraries analysed in our study come from several different laboratories and years. The quality of the contact signal can therefore vary widely.

For the contact signal in the sir3 Δ mutant, if we carefully expect the signal along the chromosomes, we can see that contact peaks are still present in this condition, for example on chromosome 11 at the peak on the *DYN1* gene:

Another argument in favor of Sir3 not being involved in 2 μ contact specificity is that it is not enriched on the 2 μ contact regions identified (Fig. 3C analysis, values: Sir3 replicate 1: 0.76, replicate 2 Sir3: 0.877).

We therefore decided to keep the analysis for the sir3 Δ mutant.

d. The 2 μ preferentially tethers to inactive regions along the host's chromosomes

i. It may be more appropriate to refer to the long genes as "lowly expressed" as opposed to "non-expressed"

We agree with this remark, most are lowly expressed, and we made the change over the text.

ii. ** For proper interpretation, a comparison of longer gene contact scores to shorter gene contact scores would be useful.

We have generated the plot of contact rate versus gene size, and averaged for two groups of genes (i.e. above and below 5 kb) and added it to **Appendix Fig. S7G**. Mann Whitney test was applied (p -value <0.001).

e. Plasmid tethering is rapidly reversible

i. ** The authors demonstrate that contact sites are rapidly "reversible" upon heat shock in Figure 2G. The interpretation of this data would be enhanced by using either RNAseq or qPCR to report on the gene expression changes of the genes featuring these changes (e.g., UTP20 and FIR1 + RZG8).

We performed such analyses with available published data from Frank Pugh lab (using time points belonging to different replicates to have enough coverage). For the genes *UTP20* and *RPA190*, we observed a decrease in the transcription level. We also computed an averaged transcription signal for genes contacted after heat shock for 3 different time points and it also showed a decrease of the transcription level of these genes.

Since there are no replicates of these Chip-exo experiments with sufficient coverage, we have decided not to include this analysis in the supplementary data.

f. 2u plasmid tether to exogenous artificial inactive chromosome

i. ** By eye, the LacO array data does not clearly support the conclusions made. Is the peak at the LacO array called with a peak-caller in the absence of LacI? Does this peak disappear once LacI binds? Perhaps a different representation would make changes more apparent.

We agree with the referee that the original visualisation plot is not clear enough. We replaced this figure and now provide in the **Appendix Figure S9** a new representation (below) that displays the contact signal as a 1D track, for the 2 conditions. A de novo peak calling analysis was applied with the same parameters. If the region is not annotated by automatic detection with default parameters, contact enrichment is clearly visible in the absence of LacI in the array of 200 lacO sites. It should be noted that automatic detection of peaks depends on many parameters that can be played with. We have, however, kept the same parameters for all types of experiment.

We removed the initial figure and put this new figure at **Appendix Figure S9** with the following legend:

“a, Contact signal of the 2μ plasmid with an array of 200 LacO binding sites without and with expression of LacI. The array of 200 LacO binding sites is represented by an orange box. Automatically detected peaks are represented by black triangles. Contact enrichment on the LacO binding site array is visible only in the condition without LacI expression.”

g. Plasmid anchoring depends on the H4 basic tail

i. ** It would be nice to improve quantification of how many peaks are called in the H4 5toA mutant. Are all 73 lost? Of course, the relevant wildtype comparison is to a strain with the pKAN plasmid. Are all 73 peaks detected for the pKAN plasmid?

Such analyses are now provided and we added the following sentence:

“If we compare the contact peaks detected for the pKan plasmid in WT background and the pKan in the 5toA background, the 5toA has only 24 detected peaks and only one common peak with the 36 peaks detected for the pKan plasmid in the WT background.”

ii. ** The authors indicate that 2u-chromosome contacts are reduced for the H4 5toA mutant, but only show this for the exogenous pKan constructs in log phase. This interpretation would benefit from analysis of the native 2u-chromosome plasmid contacts during log phase in the H4 5toA mutant.

We agree with this comment. We supplied the native 2 μ plasmid in 5toA background in quiescence (with published data from the Tsukiyama laboratory, where the 2 μ plasmid is still present in their strain) and the specificity is no longer present (**Fig. EV4D**).

Experimenting with the WT plasmid in the 5toA mutant requires retransformation of the plasmid in this mutant without a selection marker. This would require crossing experiments. We preferred not to engage in this new cycle of experiments. We now provide a second biological replicate of the pKAN plasmid in the 5toA mutant, as shown in **Fig. EV4B**.

h. Other eukaryotic plasmids tethers to inactive chromatin

i. The entire paragraph is duplicated

We removed the duplicated paragraph.

5. Discussion

a. "Condensing" should be "condensin"

This has been corrected in the new version.

b. A citation should support the notion that the probability of inter-chromosomal contacts is "very low from a thermodynamic point of view"

We will added the following reference:

Nicodemi M, Prisco A. Thermodynamic pathways to genome spatial organization in the cell nucleus. *Biophys J*. 2009 Mar 18;96(6):2168-77. doi: 10.1016/j.bpj.2008.12.3919. PMID: 19289043; PMCID: PMC2717292.

<https://www.ncbi.nlm.nih.gov/pmc/articles/PMC2717292/>

From thereon, and unless notified, all suggestions / corrections have been implemented in the final text.

6. Methods

a. Make sure that the tables are properly referenced

7. Tables

a. As with throughout, annotations of the genotypes used should be improved in the Tables 1 and 2.

Corrections have been made.

b. An additional table with information about the data sets generated for this paper (e.g., Figure, Identifier, Strains) would be useful.

A new table has been added with the following information: Library Id, GEO accession number, Description, Associated Figure.

c. For Table 3

i. One "RSGY" is written as "RSGT"

The correction has been made.

ii. 2u-containing strains should be indicated [cir+] and 2u-lacking strains should be written as [cir-]. Indicate the status of the 2-micron plasmid in all strains.

This information is now added.

iii. Indicate strain background for RSGY 1217

This information is now added.

d. Table 4

i. Define pKAN first and make it clear that KanMX disrupts FLP1 in this strain. Make it clear that the other pKAN plasmids are derivatives of the original plasmid.

We added the following sentence in the main text:

"For Y9 strain that does not naturally contain 2 μ plasmid, a pKAN plasmid was used which contains the resistance gene KanMX that disrupts *FLP1*."

We also added in Appendix Fig. S5 the information that other pKAN plasmids were derivative of the original plasmid.

"Plasmids pKAN- Δ REP1 and pKAN- Δ STB are derived from the initial plasmid pKAN whose STB region contains only the STB-P part. STB-P stands for STB-proximal, so named for its positioning relative to the single origin of replication (ORI) on the 2 μ plasmid [9]."

8. Figures

a. Figure 1

i. Currently, no units are present for the ATAC-seq, RNA-seq, and H3 chemical cleavage axes.

The units of these signal is Count Per Million reads (CPM) which has been added in the legend.

ii. 1A: Unclear what the cartoon is depicting.

Indicate in the legend that "IR" stands for "Inverted Repeat," which is why its reads are unmappable.

This information is now added in the legend of Figure 1.

Indicate whether pink or blue represent the forward or reverse strands.

The information of the direction has been added.

iii. 1C: According to the text, triangles should be annotating the dark bands in the chromosomal heat maps below.

That's correct.

iv. 1F: Given the deeper orange present across the quiescent chromosome, not clear why the contact score isn't generally higher in "quiescence."

That's a true remark. A signal smoothing process was included in these representations. As the quiescent signal was noisier, this resulted in a darker chromosomal heatmap diagram. We decided to represent the signal without smoothing for all heatmap diagrams. This display mode is consistent with the fact that the contact score in the quiescence state is not higher than in the exponential phase.

b. Figure 2

i. Currently, no units are present for the RNA-seq data.

The unit has been added in the figure: it is count per million of reads (CPM).

ii. 2F: Confused by what exactly is being displayed - given that signal/expected is lower than 1, aren't trans contacts between contact sites dis-enriched?

This is correct. Contact sites are depleted around centromeres, where the trans contact signal in chromosomes is the strongest. The average contact signal between 2 μ plasmid contact sites is thus dis-enriched. We added the following sentence:

"It appears that the regions contacted by 2 μ make less inter-chromosome contact with each other than random groups. This may be explained by the fact that 2 μ plasmid contact sites at centromeres are rare. The contact signal between chromosomes is strongest between centromeres due to the Rab1 configuration (<https://pubmed.ncbi.nlm.nih.gov/20436457/>), so 2 μ plasmid contact sites make less contact with each other than average."

c. Figure 3

i. Currently, no units for CHIP H3, ATAC-seq, CHIP-exo signal, CHIP REP1.

The CHIP signal is normalised against the Input signal to generate an enrichment ratio. The result is a dimensionless ratio indicating the fold enrichment of the immunoprecipitated protein or modification over background (input).

When no input signal is used, as in ATAC-seq, the signal density of reads can be expressed in counts per million (CPM), we added this unit into the legend.

ii. 3B: The color of yellow used does not print well.

We've darkened the associated color.

iii. For 3A and 3B, a dotted line indicating the genomic average (and statistics) would support the interpretation that H3 occupancy and chromatin accessibility differ at 2u peaks.

We've added a dotted line averaging the signal over the whole genome.

iv. 3C: Currently difficult to interpret. May be useful to

(1) label additional points of interest (e.g., proteins associated with active transcription)

We now provide an alternative representation of this plot with more accurate categories in **Appendix Fig. S11B** (see below).

and (2) include dotted lines to indicate when changes observed are significant.

We also provide the distribution of ChIP-exo signal scores for the whole data set in **Appendix Fig. S11B**. This allows us to define a threshold (1.25) above which the p-value of the observed enrichment is < 0.01 and add a dotted line at the threshold.

v. 3G: Make the dotted circle a rectangle that matches the inset.

It has been done.

d. Figure 4

i. Consider included a yeast silhouette as in the supplement for *L. fermentati* and *L. waltii*

This has been done.

9. Supplementary Figures

a. Supplementary Figure 1

i. No units for any plot

The units are counts per million (CPM). The information has been added in the legend of the main text and in the Appendix Fig. S1.

ii. Make it clear that H3 chemical cleavage, RNA-seq, and ATAC-seq are reproduced from 1A.

The information has been added in the legend.

Make data representation (e.g., strand portrayal and color choices) match.

This has been corrected.

Interpretation would be facilitated by depicting Cse4 enrichment at a centromere.

We added a plot with the Cse4 ChIP-seq signal at the centromere of chromosomes II.

b. Supplementary Figure 3

i. Consider indicating the contact sites with black triangles as in Supplementary Figure 2.

We added them to Figure B.

ii. 3A: Indicate what ChIP-seq REP1 signal represents (e.g., consider including an axis).

Both signals were Z standardised i.e. $z = (x - \mu) / \sigma$, where x is the ChIP-seq signal value (ChIP/input) or contact score, μ is the mean and σ is the standard deviation. The Z score (or standard score) represents how many standard deviations a particular data point is from the mean of the dataset. It is a way of standardizing different data points to compare them on a common scale. We added the following information in the legend of Figure B:

“ Both signals were Z standardised i.e. $z = (x - \mu) / \sigma$, where x is the ChIP-seq signal value (ChIP/input) or contact score, μ is the mean and σ is the standard deviation of each signal. “

c. Supplementary Figure 4

- i. Improve image quality (currently pixelated) and make sure that gene labels aren't cut off.

The quality and resolution of the image has been corrected and improved as well as for all images in the article.

- ii. Long gene gray boxes aren't visible

The long genes annotation has been checked and added where necessary.

d. Supplementary Figure 5

- i. Indicate references where the data was taken from

This information has been added.

- ii. Improve annotation of genotypes

This has been done.

- iii. It may be worth providing this information for all mutants analyzed

In order not to overload this figure, which contains the most important mutants and conditions, we have added a column in Appendix Table S3 for this information.

e. Supplementary Figure 6

- i. Relabel pKAN-STB-P as pKAN and make 6C 6A to indicate the other plasmids are derived from it.

We relabeled pKAN-STB-P as pKAN and we've specified in the legend that the other plasmids are derived from the pKAN plasmid.

- ii. Not clear what "STB-P" is

The information is now added with the corresponding reference in the legend:

"Plasmids pKAN- Δ REP1 and pKAN- Δ STB are derived from the initial plasmid pKAN whose STB region contains only the STB-P part. STB-P stands for *STB-proximal*, so named for its positioning relative to the single origin of replication (*ORI*) on the 2 μ plasmid [9]."

f. Supplementary Figure 7

- i. Edit almost all labels to improve clarity re: experimental condition and genotype. A few of the issues (not exhaustive) are indicated below.

1. No_tension = nocodazole

Yes that's correct.

2. What is "t4"?

This corresponds to 4 h after induction of HO endonuclease-mediated site-specific double strand break (DSB). This information has been added.

How was Fkh depleted? What genes are "Fkh" and "topo2" referring to?

Fkh1-Fkh2 depleted mutant was done using the conditional construction GAL1pr-FKH1 fkh2 Δ mutant.

Fkh correspond to FKH1 and FKH2 genes.
topo2 corresponds to Top2 protein which is a topoisomerase II.

How were Clb5 and Clb6 disrupted?

The constructions *clb5Δ::KanMX6* *clb6Δ::TRP1* were used. (for details see supplementary data of <https://www.ncbi.nlm.nih.gov/pmc/articles/PMC8856730/#supp1>).

The Fig. EV2 has been corrected so that gene and mutant nomenclatures are correct.

ii. The appropriate comparison for Y9 + pKan is BY + pKan
That's correct: we added the Xcontrol BY + pKAN to the Fig. EV2A.

iii. Indicate that the WT (microC) vs. quiescence (microC) data is a reproduction of Figure 1C.

The information has been added.

g. Supplementary Figure 9

i. Unclear whether longer genes just tend to be more lowly expressed, or if gene length and low expression both independently contribute to contact site formation

We are conscious of this point. One of the analyses that responds to this remark is Appendix Figure S7A. We draw attention to the fact that in the scatter plot, only the points in the top left-hand zone show excess contact with the 2μ .

If it were only a low transcript level, the whole left part of the scatter plot would be enriched in contact, and if it were only gene size, the whole upper part of the plot would be enriched in contact.

We then carried out several statistical analyses, varying just one of the 2 parameters (size and transcript level), which correspond to Appendix Figs. S7 E,F. Although more difficult to read, these figures support the effects of the 2 variables. In these analyses, we set one of the parameters, such as gene size, and look at the correlation between contact rate and

transcript level. This shows that when size remains constant, the correlation with transcript level is still detectable. The same applies to the reciprocal analysis when the transcription level is fixed.

We believe that together these analyses are sufficient to show that the 2 parameters; long and low transcript level are important in creating a contact hotspot.

ii. 9F: Labels are illegible

We corrected the labels giving only those that are necessary. The number of elements is present to indicate that the loss of significance is due to a small number of elements for the computation of the Pearson correlation (we added this information in the legend).

h. Supplementary Figure 10

i. As with Figure 2F, confused by what exactly is being displayed - given that signal/expected is lower than 1, aren't trans contacts between contact sites dis-enriched?

This is correct, regions contacted by the 2μ make less contact with each other than expected. Contact sites are depleted around centromeres (**Fig. 1E**), where the trans contact signal in chromosomes is the strongest. It follows that the average contact signal between 2μ plasmid contact sites is thus dis-enriched. We added the following sentence:

"It appears that the regions contacted by 2μ make less inter-chromosome contact with each other than random groups. This may be explained by the fact that 2μ plasmid contact sites at centromeres are rare. The contact signal between chromosomes is strongest between centromeres due to the Rab1 configuration (<https://pubmed.ncbi.nlm.nih.gov/20436457/>), so 2μ plasmid contact sites make less contact with each other than average."

i. Supplementary Figure 11

i. Centromere depiction doesn't match the label

This has been corrected and enlarged.

ii. The red used for the "2 hour" time point is difficult to see

We change the colours for the 2 min and 1 min time points. We also added an inset to zoom specifically on the UTP20 gene that becomes a hotspot of contact.

j. Supplementary Figure 12

i. ** Co-localization quantification is missing and would help support the point made in Figure 2J

We agree with this comment and now added a new quantification analysis. The distribution of distances in μm between Mmyco supplementary chromosome and 2μ plasmid (red line) was computed and compared with a null model consisting in a random distribution of positions in the nucleus (green line, MC-CI 5%, Monte Carlo approach with confidence interval at 5%). The figure was added in the **Appendix Fig. S10D**.

ii. Not clear what the clause "distribution of the number of 2μ plasmid foci" is referring to

This corresponds to cell number distribution as a function of the number of 2μ spots identified per nucleus. We rephrased the legend which was not clear:
"c, Distribution of the number of 2μ plasmid foci per cell. "

k. Supplementary Figure 14

i. Improve annotation of all genotypes
It has been done in the figure and legend.

ii. Seems as though dot1, set2, rsc2, and hst2 data are all reproduced from the main figure
Yes, that's correct. We added the precision in the legend: "(same as figure 3)".

iii. For some of the strains (e.g., Ino80, Fun2, and Isw2), the peaks in the control are so low that the experiments aren't interpretable

We agree that the enrichment on the agglomerated plot shown for several libraries is not very strong in the control and in the mutant. This can be explained by the fact that the quality of contact data is quite variable in many of the conditions tested as explained above for the *sir3Δ* mutant. However, if we look closely at the contact signals on a chromosome scale, we can see that the contact peaks detected are still visible. Here we show an example for chromosome 11 here for Ino80 depletion condition: although the contact signal is much weaker and flatter, enrichment of contact at long genes (grey boxes) is still visible:

iv. Unclear whether the difference observed for SWR1-AID is significant. By eye, effect size seems similar to what is observed for *set2*, albeit in the opposite direction

We agree that some differences between mutant and control are difficult to quantify and would require further work to confirm them. Here we only show that contact specificity is still visible in the mutant and that the protein tested is not essential to maintain contact specificity.

I. Supplementary Figure 15

i. ** Unclear why the WT in Figure S15A and S15C is labeled HHF2.

Now Fig. EV4.

In this experiment, the control is HHF2 and comes from the work of Tsukiyama Lab (<https://elifesciences.org/articles/72062>).

In this strain, the endogenous H3 and H4 loci were deleted and complemented by a wild-type copy of H3 and H4 genes at an ectopic locus (TRP1). This control strain grow and enter quiescence very similarly to true WT strains with two copies of H3 and H4 genes.

Here the information concerning the strain that we added in the **Appendix Table S4**:

yTT7177 HHF2	MATa RAD5+ ura3-1 hht1-hhf1::Nat hht2-hhf2::Hyg trp1-1::pRS 404-HHT2-HHF2	W303	Swygert et al 2021
-----------------	---	------	--------------------------

We changed the annotations on the figure and added in the legend:

“In the HHF2, control the endogenous H3 and H4 loci were deleted and complemented by a wild-type copy of H3 and H4 genes at an ectopic locus (TRP1). ”

Also, is the dip observed for Rbp3 binding in the H4 5toA mutant not significant? A line depicting the genomic average and statistics could improve interpretation.

The dip observed in Rbp3 ChIP-seq that can be seen in the agglomerated plot suggests that the 2 μ contact rate would be stronger in this mutant at the identified regions, since we have shown that low transcriptional activity is favorable to the establishment of 2 μ plasmid contacts. The fact that plasmid 2 μ no longer has contact enrichment at these less transcribed regions, is therefore a strong sign that contact specificity has been lost in this mutant.

We have now added the line for the ChIP-seq signal, IP/input for the 2 agglomerated plots in log and quiescence phases.

ii. S15D: Misleading to compare the wildtype 2 μ contact peaks during log phase to the mutant contact peaks during quiescence - this graph should be removed (or the WT quiescence should be used as the control). Additionally, not clear what the HHF2 control is.

We agree with the referee that the legend can be confusing. HHF2 is the control carried in quiescence but the agglomerated plot was realised on the same set of reference peaks used throughout the analysis, i.e. the peaks detected under standard conditions: in an asynchronous population for WT, log phase, strain BY4741. This analysis represents a control in another biological state, the quiescence of the loss of 2 micron contact specificity in the 5toA mutant. We have clarified the legend.

iii. S15E: The centromeres are not visible and the long genes are not consistently depicted (e.g., height is not identical)

Centromere symbols have been enlarged and homogenised, as well as long genes.

iv. Figure legend indicates that "pKAN version of the 2 μ plasmid was used to ensure plasmid stability" for S15B - not clear that there is impaired plasmid stability in the H4 5toA mutant. If so, why was this not necessary for S15D?

We agree with this remark. In the quiescent contact data from Tsuliyama's laboratory, 2 μ was still present. When we used the strain sent by this laboratory (reanalysed and shown in **Fig. EV4D**), the 2 μ was no longer present, so we retransformed with the pKAN plasmid. This plasmid, which contains a resistance gene, made it easy to carry out the transformation and also to force it to remain present in the mutants if its stability was affected.

m. Supplementary Figure 16

i. S16A and B: Chromosome labels can't be read

This has been corrected.

ii. S16B: A contact plot with long genes annotated should be included for at least one chromosome.

This analysis has now been carried out with new Hi-C data, showing the LAWA0B chromosome of *Lachancea Waltii* for which the greatest number (6) of large genes (>7 kb) are found.

We also redo the same analysis for *Lachancea fermentati* with new Hi-C data:

iii. S16D: Centromeres should be depicted (if known). Long gene annotation difficult to see.

The centromeres are not well annotated in this organism. They could be found using Hi-C, but that's beyond the scope of this article. The resolution of the figure has been improved to enlarge the figure and show the long genes. Also we now provide a statistical test with bootstrap strategy to show that the proportion of long genes in detected peaks is stronger than expected and added in the main text:

“Of the 76 long genes (size > 10kb), 20 are detected as plasmid contact hotspots (p-value < 0.001, Methods)”.

Dear Dr Cournac,

Thank you for submitting a revised version of your manuscript. Your study has now been seen by all original referees, who find that their previous concerns have been addressed and now recommend publication of the manuscript. Please also find the additional comment of referee #2 to your second point-by point response below. There remain only a few mainly editorial points that have to be addressed before I can extend formal acceptance of the manuscript:

- Please remove the main figures and track changes from the manuscript
- We are only able to accommodate up to 5 keywords.
- Please adjust the format of the reference list and of the in-text citations according to EMBO Journal format (alphabetical order, author name et al + year.../up to 10 author names in the reference list before et al / please refer to our Guide to Authors for additional information on EMBO J reference format).
- Please unite the "Data accessibility" and "Code Availability" section into a single "Data Availability" section.
- Please rename the Conflict of Interest section into "Disclosure and Competing Interests Statement", in accordance with our updated Guide to Authors (<https://www.embopress.org/competing-interests>)
- As we are switching from a free-text author contribution statement towards a more formal statement based on Contributor Role Taxonomy (CRediT) terms, please remove the present Author Contribution section and instead specify each author's contribution(s) directly in the Author Information page of our submission system during upload of the final manuscript. See <https://casrai.org/credit/> for more information.

There is a reference to "data not shown" on page 11, "We explored many other different genomic signals, most of which do not show an enrichment signature (data not shown)." section. According to our policy, which does not permit references to "data not shown", please include this information in the Appendix. Please see also <https://www.embopress.org/page/journal/14602075/authorguide#unpublisheddata>.

Please remove the coloured text from the APPENDIX 1 FILE WITH ToC and correct references to alphabetical, 10 authors + et al.

- Please provide the Reagent and Tools Table. For more information, please check <https://www.embopress.org/page/journal/14602075/authorguide#structuredmethods> and download the template for Reagent Table (attached for your convenience)
- Please provide suggestions for a short 'blurb' text prefacing and summing up the conceptual aspect of the study in two sentences (max. 250 characters), followed by 3-5 one-sentence 'bullet points' with brief factual statements of key results of the paper; they will form the basis of an editor-written 'Synopsis' accompanying the online version of the article. Please also provide an altered synopsis image, making sure that the aspect ratio conforms to our website's format - it should be exactly 550 pixels wide and between 300-600 pixels high.
- Please provide the specific URL for GSE246637 dataset in the data availability statement.
- Figure Legends (main + EV):
 1. Please define the annotated p values * as well as provide the exact p-values for the same in the legend of figure EV 4g; as appropriate.
 2. Please note that the exact p values are not provided in the legends of figures 3h; EV 4f.
 3. Please note that in figures 3h; EV 4f; there is a mismatch between the annotated p values in the figure legend and the annotated p values in the figure file that should be corrected.
 4. Please note that the box plots need to be defined in terms of minima, maxima, centre, bounds of box and whiskers, and percentile in the legends of figures 3h; EV 4f.
- Please adjust the order of the manuscript sections: Title page with complete author information, Abstract, Keywords, Introduction, Results, Discussion, Methods, Data Availability Section, Acknowledgements, Disclosure and Competing Interests

Statement, References, Main figure legends, Tables, Expanded Figure Legends.

With best regards,

Cornelius Schneider

Cornelius Schneider, PhD
Editor
The EMBO Journal
c.schneider@embojournal.org

We realize that it is difficult to revise to a specific deadline. In the interest of protecting the conceptual advance provided by the work, we recommend a revision within 3 months (10th Apr 2025). Please discuss the revision progress ahead of this time with the editor if you require more time to complete the revisions. Use the link below to submit your revision:

Referee #1:

I have been through the revised manuscript and responses of the authors to the comments raised by reviewers including myself.

I would like to thank the authors for the open-minded and very thorough consideration of all the points raised, and the considerable amount of extra work done, particularly the plasmid stability assays which were essential for the correct interpretation of the results. I am satisfied that all points have been well considered and dealt with from my perspective and believe this manuscript now represents a thorough analysis of the association of 2micron with chromosomes and a valuable addition to our understanding of the maintenance mechanism. I only note that it is remarkable we still do not completely understand this mechanism after more than 50 years of research, which illustrates the ongoing interest in this problem!

Referee #2:

I have went through the revised version of the manuscript entitled "Anchoring of parasitic plasmids..." by Girard et al. The authors have made commendable efforts to address the points raised by the three reviewers. However, I feel that there is still some work needed before this interesting paper is ready for publication.

Main point: The data added in Figure 3H are problematic in multiple ways.

1) The authors claim that the data represent the average and standard variation of three transformants, whereas the data is plotted in form of box and whiskers (representing the distribution of the data as the lowest and highest 25% - in the whiskers - and 50% rest of the data in the box, where the median is indicated). Not only this representation is not what is announced by the authors but it also requires substantially more than 3 measurements... This representation is not compatible with the claim that it is based on 3 data points at each time point... It is therefore very unclear what the actual data is...

2) From what one can understand from the rebuttal letter and the strain list, the author used a flp1- plasmid in a cir- strain (this should be made clear in the main text). It is unclear why they did not use a wild type plasmid or at least a context where it can replicate normally (cir+ strain). Fact is that this plasmid is very unstable on its own. Barely 30% of the cells keep it even during growth in selective medium. This is not substantially better than replicative, non-CEN plasmids, which have no segregation mechanism... There is at least need for discussion, if not redoing the experiment in a more physiological context.

3) More importantly, in the H4-5toA strain this plasmid is only slightly less stable than in wild type cells, whereas the rep1- and stb- plasmids are lost at a much higher rate. The H4-5toA, rep1- and stb- mutations all erase the interaction of the plasmid with the attachment loci on chromosomes as convincingly shown in Fig 3F for the 5toA mutation and in Fig S5A-B for the Δ REP1 and the Δ STB plasmids. This poses an important question: Why is the 5toA mutation not having the same effect as the rep1- and stb1- mutations? As it stands, the data shows that the effects of Rep1 and STB in plasmid stability are largely independent of the attachment to the sites discovered by the authors, which, accordingly to the 5toA data, is itself at best marginal...

Minor points:

1- The paper remains not optimally written. Having a native speaker edit the manuscript for proper language would make it much more convincing.

2- The authors mention the RZG8 gene but this gene does not exist in the SGD database, which is the reference database. This needs clarification.

3- Page 8: "A statistical analysis shows more generally that the 2micron contacts depend on the transcription levels..." To the knowledge of this reviewer, statistical analysis can measure the strength of a correlation but not the causality underlying it. The verb "depend" is not accurate.

4- Page 9: DYN1 is a very long but not a silent gene. Even if expressed at low level, it is needed for proper spindle positioning during each mitosis and its deletion has a measurable phenotype in exponentially growing cells.

Referee #3:

Overall, the authors did a thorough job addressing all of my concerns, and I am happy to recommend that this paper be accepted for publication.

Two additional comments/suggestions:

(1) I would add to the discussion speculation as to whether the H4-5toA mutant didn't have a more dramatic effect (e.g., a Rep1 or STB deletion mutant) -- this could suggest other mechanisms to ensure stability or that non-specific chromosome contacts are still sufficient to drive inheritance.

(2) A typo exists in the following sentence: "the 5toA has only 24 detected peaks and only one common peak with the 36 peaks detected for the pKan plasmid in the WT background." I think there are three common peaks.

Dear Dr. Schneider,

Thank you very much for your email and for giving us the opportunity to respond to Reviewer 2's comments. We appreciate their feedback. Overall, all the reviewers were very constructive.

We hope to be able to convince you that the work qualifies for publication in the EMBO Journal with the following changes to address the remaining concerns.

We feel that carrying out a new series of experiments is a stretch for us (the first author having defended his PhD and embarked on other life projects), but we have nevertheless included new extra data points regarding the stability assay. In addition, we have also reprocessed the data, reorganized the figures, and rewrote the text to take into account the different points raised by reviewer 2.

As you will see in the accompanying response to reviewer, we think we complied to all reviewers'2 comments, and hope you agree with this suggested revision plan. Alternatively, we could embark on more experiment, but we feel that the results presented here are sound and make already new and original contribution to a long-standing question, as pointed by two reviewers. We noticed a renewed interest in the field of chromosome biology of episome maintenance and biology, and think this work would be a timely addition.

We would be happy to discuss further with you if necessary.

We would be happy to discuss further with you if necessary.

Best regards,

Axel Cournac

for the authors

Referee #1 (Report for Author)

I have been through the revised manuscript and responses of the authors to the comments raised by reviewers including myself. I would like to thank the authors for the open-minded and very thorough consideration of all the points raised, and the considerable amount of extra work done, particularly the plasmid stability assays which were essential for the correct interpretation of the results. I am satisfied that all points have been well considered and dealt with from my perspective and believe this manuscript now represents a thorough analysis of the association of 2micron with chromosomes and a valuable addition to our understanding of the maintenance mechanism. I only note that it is remarkable we still do not completely understand this mechanism after more than 50 years of research, which illustrates the ongoing interest in this problem!

We thank the referee for their comment.

Referee #2 (Report for Author)

I have went through the revised version of the manuscript entitled "Anchoring of parasitic plasmids..." by Girard et al. The authors have made commendable efforts to address the points raised by the three reviewers. However, I feel that there is still some work needed before this interesting paper is ready for publication.

Thanks, we hope the following will address these remaining concerns.

Main point: The data added in Figure 3H are problematic in multiple ways.

1) The authors claim that the data represent the average and standard variation of three transformants, whereas the data is plotted in form of box and whiskers (representing the distribution of the data as the lowest and highest 25% - in the whiskers - and 50% rest of the data in the box, where the median is indicated). Not only this representation is not what is announced by the authors but it also requires substantially more than 3 measurements... This representation is not compatible with the claim that it is based on 3 data points at each time point... It is therefore very unclear what the actual data is...

We thank the referee for raising this point, and we apologize for the lack of clarity and pointing at a mistake in this figure. We have added and reorganized data, changed the visual representation, reorganized the figures and rewrote the text as follows.

In the original Fig. 3H, three replicates per strain were used for each time point. However, we recognize that the use of boxplot to visualize these data was not appropriate and misleading.

- We now first present an experiment showing stability measurements with 5 replicates for each strain. We modified the presentation of the plot : the graph displays each point corresponding to a different replicate, and the average \pm standard deviation for each strain is shown using a bar plot :

The associated statistical test remains the same (Chi square test). It should be noted that these measurements correspond to a series of measurements which were only presented as supplementary data (EV 4F) in the version of the resubmitted article.

- We next display the time course measurements, and added measures to have at least 5 replicates for each time point (the original 3 of the revised version plus a second series of measurements containing 3 new replicates, bringing the total number of replicates to 6 for each time point).
- We also adopted bar plot visualization, representing the average \pm standard deviation:

These 2 panels are now presented in **Fig. 3H** and **Fig. 3I**. The two panels are separated because the measurements are not carried out in exactly the same way (in the first panel, we used velvet replicas to count resistant colonies, Methods).

Overall, both approaches show that the 5toA mutation has a significant effect on the presence of the plasmid, albeit to a lower extent than the *rep1* or *STB* mutation (below).

2) From what one can understand from the rebuttal letter and the strain list, the author used a *flp1*- plasmid in a *cir*- strain (this should be made clear in the main text). It is unclear why they did not use a wild type plasmid or at least a context where it can replicate normally (*cir*+ strain). Fact is that this plasmid is very unstable on its own. Barely 30% of the cells keep it even during growth in selective medium. This is not substantially better than replicative, non-CEN plasmids, which have no segregation mechanism... There is at least need for discussion, if not redoing the experiment in a more physiological context.

We apologize for the lack of clarity.

We chose to work with this plasmid because the presence of the *KAN* marker enables direct stability measurements as described in the literature, and because the same plasmid was previously used by the Dobson lab to monitor the stability of the molecule in mutant context (McQuaid ME et al., *Nucleic Acids Res.* 2019 PMID: 30445476). It also makes it simple to transform a strain, as was the case with the H4 5toA mutant.

Compared to the native version of the 2 μ plasmid (cir+ strain), the pKan plasmid lacks the *FLP1* gene and contains the *KAN* marker. It contains all the other *REP1*, *REP2* and *RAF1* genes as well as the same ARS origin of replication as the natural plasmid. To underline these differences, we have included a map of the pKan plasmid in the main **Fig. 3E**.

We have also added in the text :

“We used an artificial version of the 2 μ plasmid, called pKan, which differs from the natural version in that it contains a KAN gene and does not contain the Flp1 recombinase gene (Fig. 3E).”

As pointed out by the reviewer, the stability of the pKan plasmid is lower than that of the native 2 μ in a host wt context. We now indicate that this is the case in the manuscript, and propose a couple of explanations to account for these differences.

One is that it could be due to replication. Indeed, the pKan plasmid contains the same origin of replication (ARS) as the natural version, enabling it to replicate like the natural 2 μ thanks to the host's replication machinery. But it lacks the Flp1-specific recombinase, which is expressed if the number of plasmids per cell falls below a certain threshold (Ghosh SK et al. *Annu Rev Biochem.* 2006; PMID: 16756491). Therefore, although the plasmid will replicate once per cycle at the start of S phase as the natural version of the 2 μ (Zakian VA et al., *Cell.* 1979 PMID: 385147), the absence of Flp1 may result in lower proportions of cells having the plasmid.

Another difference between the natural and artificial versions is the presence of the *KAN* gene in the pKan, which must be expressed to provide resistance to geneticin (G418). This transcription may interfere with its attachment/segregation mechanism, and increase its instability compared to the natural system in WT strains.

We added the following sentences to the text to clarify these differences with the wild-type 2 μ sequence.

“Note that only ~40% of the pKan plasmid was found in WT in the stability assay. This reduced stability of this artificial plasmid could have a different origin that remains unknown, such as the absence of the specific recombinase Flp1 which can correct a small number of plasmid copies, or the presence of the expressed KAN gene which could alter the plasmid's topology and attachment mechanism.”

and:

“The H4 5toA mutant resulted in a decrease of the 2 μ reporter plasmid. This decrease was not as dramatic as the known effect of Rep1 and STB mutations, but remained significant (Fig. 3I).”

Figure 3. Specific positioning may be associated with nucleosome signal.

A, Averaged signal at the hotspots of contact with 2 μ plasmid for nucleosome occupancy (ChIP-seq of H3 histone) and **B**, chromatin accessibility (ATAC-seq). **C**, Average value of signal at the hotspots of contact for 1251 ChIP-exo libraries sorted by general categories (Rossi et al., 2021). **C**, Average value of signal at the hotspots of contact for 1251 ChIP-exo libraries sorted by general categories (Rossi et al., 2021). **D**, Averaged contact signal of the 2 μ plasmid in mutants of epigenetic marks *dot1* Δ , *set2* Δ and mutant in chromatin remodeler *rsc2* Δ and in deacetylase mutant *hst2* Δ . **E**, Contact profile of the 2 μ plasmid (pKan version) with chromosomes of *S. cerevisiae* in H4 5toA mutant, WT and of the pARS plasmid in WT for chromosome XI. **F**, Averaged contact signal of the 2 μ plasmid in H4 5toA mutant and control. **G**, Averaged signal around TSS for nucleosomes and Rep1 occupancy signals. **H**,

Stability measurements showing inheritance of a 2 μ m-based pKan plasmid in a *cir*⁰ yeast strain determined by plating assays for WT and 5toA mutant after overnight culture (O/N)(Methods). Results represent the average (\pm s.d.) from assaying 5 replicates for each condition. **I**, Stability measurements showing inheritance of the 2 μ m-based (pKan) plasmids in a *cir*⁰ yeast strain determined by plating assays for WT, 5toA, rep1 Δ and Δ STB mutants. Results represent the average (\pm s.d.) from assaying 6 replicates for each condition at 3 different time points (6h, 12h, 24h) after O/N culture. Asterisks indicate significant differences determined by a Chi square test (* P < 0.05, ** P < 0.005, *** P < 0.0005).

3) More importantly, in the H4-5toA strain this plasmid is only slightly less stable than in wild type cells, whereas the rep1- and stb- plasmids are lost at a much higher rate. The H4-5toA, rep1- and stb- mutations all erase the interaction of the plasmid with the attachment loci on chromosomes as convincingly shown in Fig 3F for the 5toA mutation and in Fig S5A-B for the Δ REP1 and the Δ STB plasmids. This poses an important question: Why is the 5toA mutation not having the same effect as the rep1- and stb1- mutations? As it stands, the data shows that the effects of Rep1 and STB in plasmid stability are largely independent of the attachment to the sites discovered by the authors, which, accordingly to the 5toA data, is itself at best marginal...

We agree with the point raised and have decided to add clarifications in the text to provide a more accurate interpretation of the measurement results in the 5toA mutant.

Our favourite hypothesis is that, although contact specificity is lost in the 5toA mutant, a general degree of non-specific affinity for host chromosomes persists. The plasmid partitioning and segregation processes are not drastically affected in the 5toA mutant, hence the significant effect we measure in stability experiments. In the two rep1 Δ and Δ STB mutants of the plasmid, a key plasmid component of the tethering system is abolished, hence the much lower stabilities rates.

We propose to ponder our text and discuss more thoroughly this important point, as also suggested by reviewer 3.

“The drop in stability of the pKan plasmid in a 5toA host mutant is significant, but does not reach the dramatic effect of both the rep1 Δ and Δ STB plasmid mutations, two conditions where a key plasmid component of the tethering system is abolished. One interpretation could be that, although contact specificity is lost in the 5toA mutant, non-specific interactions or tethering to host chromosomes can persist, which could help maintain some segregation ability.”

Minor points:

1- The paper remains not optimally written. Having a native speaker edit the manuscript for proper language would make it much more convincing.

We have re-read the entire text and changed a number of wordings and punctuation marks.

2- The authors mention the RZG8 gene but this gene does not exist in the SGD database, which is the reference database. This needs clarification.

We apologize for this typo, which refers to the ZRG8 gene and not RZG8. This typo has now been corrected in the text and on Figure 2G. We thank the referee for pointing this out.

3- Page 8: "A statistical analysis shows more generally that the 2micron contacts depend on the transcription levels..." To the knowledge of this reviewer, statistical analysis can measure the strength of a correlation but not the causality underlying it. The verb "depend" is not accurate.

We agree with this comment and important distinction. We have therefore reworded the text as follows:

"A statistical analysis shows more generally that the 2 μ contacts are correlated with the transcription level and size of the gene (**Appendix Fig. S7**) (Methods)."

4- Page 9: DYN1 is a very long but not a silent gene. Even if expressed at low level, it is needed for proper spindle positioning during each mitosis and its deletion has a measurable phenotype in exponentially growing cells.

The text has now been corrected:

*"A magnification of contact distribution over the long inactive genes revealed a maximum enrichment in the middle of the weakly expressed gene (e.g., DYN1 gene in **Fig. 2A, Fig. 2E**)."*

Referee #3 (Report for Author)

Overall, the authors did a thorough job addressing all of my concerns, and I am happy to recommend that this paper be accepted for publication.

We thank the referee for the positive feedback.

Two additional comments/suggestions:

(1) I would add to the discussion speculation as to whether the H4-5toA mutant didn't have a more dramatic effect (e.g., a Rep1 or STB deletion mutant) -- this could suggest other mechanisms to ensure stability or that non-specific chromosome contacts are still sufficient to drive inheritance.

We fully agree with this comment and have rewritten part of the discussion to address it (see response to reviewer #2 point 3). Indeed, we think that although contact specificity is lost in the 5toA mutant, non-specific binding to host chromosomes may persist and explain the improved stability.

(2) A typo exists in the following sentence: "the 5toA has only 24 detected peaks and only one common peak with the 36 peaks detected for the pKan plasmid in the WT background." I think there are three common peaks.

We thank the referee for this typo, which has now been corrected.

Additional comment by referee #2:

I have now went through the response of the authors to my comments and the revised manuscript. I agree with their response. Their revised manuscript addresses my issues satisfactorily enough for now (more work will be needed in the future) and I have no objection to make against publication of this important study.

All editorial and formatting issues were resolved by the authors.

Dear Dr. Cournac,

I am pleased to inform you that your manuscript has been accepted for publication in the EMBO Journal.

Yours sincerely,

Cornelius Schneider, PhD
Editor
The EMBO Journal
c.schneider@embojournal.org
